# Coresets from Trajectories: Selecting Data via Correlation of Loss Differences

**Manish Nagaraj**                                                    *mnagara@purdue.edu*
*Electrical and Computer Engineering*
*Purdue University*

**Deepak Ravikumar**                                                  *dravikum@purdue.edu*
*Electrical and Computer Engineering*
*Purdue University*

**Kaushik Roy**                                                       *kaushik@purdue.edu*
*Electrical and Computer Engineering*
*Purdue University*

**Reviewed on OpenReview:** *https://openreview.net/forum?id=QYOpbZTWJ9*

## Abstract

Deep learning models achieve state-of-the-art performance across domains but face scalability challenges in real-time or resource-constrained scenarios. To address this, we propose *Correlation of Loss Differences* (CLD), a simple and scalable metric for coreset selection that identifies the most impactful training samples by measuring their alignment with the loss trajectories of a held-out validation set. CLD is highly efficient, requiring only per-sample loss values computed at training checkpoints, and avoiding the costly gradient and curvature computations used in many existing subset selection methods. We develop a general theoretical framework that establishes convergence guarantees for CLD-based coresets, demonstrating that the convergence error is upper-bounded by the alignment of the selected samples and the representativeness of the validation set. On CIFAR-100 and ImageNet-1k, CLD-based coresets typically outperform or closely match state-of-the-art methods across subset sizes, and remain within 1% of more computationally expensive baselines even when not leading. CLD transfers effectively across architectures (ResNet, VGG, DenseNet), enabling proxy-to-target selection with $< 1\%$ degradation. Moreover, CLD is stable when using only early checkpoints, incurring negligible accuracy loss. Finally, CLD exhibits inherent bias reduction via per-class validation alignment, obviating the need for additional stratified sampling. Together, these properties make CLD a principled, efficient, stable, and transferable tool for scalable dataset optimization. [1]

## 1 Introduction

Deep learning models rely on large and diverse datasets to achieve state-of-the-art performance across a wide range of tasks. However, training on such datasets is increasingly constrained by compute and memory budgets, especially in real-time or resource-limited settings. This raises a fundamental question: *Which subsets of data most effectively support generalization?* A natural answer is offered by *coresets*, compact, representative subsets of training data that retain full-dataset performance when used for training.

Coresets support a range of applications including active learning (Coleman et al., 2020), neural architecture search (Na et al., 2021; Shim et al., 2021), dataset distillation (Cazenavette et al., 2022), and continual learning (Aljundi et al., 2019; Borsos et al., 2020). However, most existing approaches are either based

---

[1]The code is available on GitHub.

on heuristic criteria unrelated to generalization (Toneva et al., 2018; Belouadah et al., 2020; Rebuffi et al., 2017), or expensive second-order or bilevel optimization (Killamsetty et al., 2021a; Pruthi et al., 2020; Garg & Roy, 2023; Killamsetty et al., 2021b; Xia et al., 2024b; Borsos et al., 2020), limiting scalability.

We propose a simple, scalable, and theoretically grounded alternative to coreset generation using a metric we define as the **Correlation of Loss Differences** (`CLD`). This metric quantifies how closely the loss trajectory of a training sample aligns with the average validation loss trajectory during training (Figure 1, left). Since the validation set reflects the test distribution, high positive `CLD` samples are likely to contribute positively to generalization. Selecting such samples yields compact coresets that preserve, or even improve, test accuracy by filtering out ambiguous or harmful examples (Figure 1, right).

Beyond simplicity, `CLD` provides strong theoretical guarantees. We prove that training on high-`CLD` samples achieves convergence in population risk with an error bound that closely matches full-data training, where the excess error is explicitly governed by the sample alignment parameter $\kappa$ and the validation representativeness $\delta$ (see Theorem 1). Our theory reveals that high `CLD` is not just sufficient but also necessary to minimize the convergence error-bound under coreset training.

We validate these findings across CIFAR-100 and ImageNet-1k, where `CLD`-selected coresets typically outperform or closely match state-of-the-art methods across a wide range of coreset sizes, and remain within 1% of more computationally expensive baselines even when not leading. Unlike these methods, `CLD` requires only per-sample loss values, allowing it to avoid the costly gradients, Jacobians, or feature embeddings used by many influence- and similarity-based selectors. Concretely, the only computational overhead beyond a standard training run on the full dataset of size $N$ is a single forward pass at each checkpoint over a small, held-out validation set of size $Q$. Since the validation set is typically much smaller than the training set ($Q \ll N$), this lightweight approach yields significant gains in both computational and storage efficiency, which we quantify in our full cost analysis summarized in Table 1.

An additional strength of `CLD` is its robustness; the metric remains stable when computed using sparsely sampled training checkpoints and is consistent across random seeds, making it practical for large-scale or budgeted deployments. Furthermore, `CLD` coresets transfer effectively across architectures. This is one of the key advantages of `CLD`. Coresets computed using small proxy models (e.g., ResNet-18) generalize to larger models (e.g., ResNet-50, DenseNet) with performance drops consistently under 1%. Hence we can compute coresets using a smaller proxy model (e.g., ResNet-18) and reuse it to train larger target models (e.g., ResNet-50/VGG/DenseNet) with minimal loss in accuracy, while substantially reducing selection cost.

**Summary of contributions:**

1. **Correlation of Loss Differences for Coreset Selection.** We introduce `CLD`, a simple and scalable metric for coreset construction based on the correlation between a training sample's loss differences and the average validation loss trajectory, serving as a proxy for generalization, in Section 4.

2. **Theoretical Guarantees.** We develop a general convergence framework showing that training on high-`CLD` samples yields population risk close to full-data training, with the suboptimality explicitly governed by sample alignment and validation representativeness, in Section 5.

3. **Experimental Validation.** We show that on CIFAR-100 and ImageNet-1k, `CLD`-selected coresets typically outperform or closely match state-of-the-art methods across a wide range of subset sizes, and remain within 1% of more expensive baselines when not leading, in Section 6.

4. **Efficiency, Transferability, and Stability.** `CLD` avoids gradient and curvature computations, incurs minimal compute and storage cost, transfers across architectures via proxy models, and remains stable under checkpoint subsampling and random seeds, making it highly practical for large-scale settings. We discuss these in Section 7 and Section 8.

## 2 Related Literature

The need for scalable coreset methods has led to a variety of approaches, which can be broadly grouped into *score-based*, *optimization-based*, and *training property-based* methods.

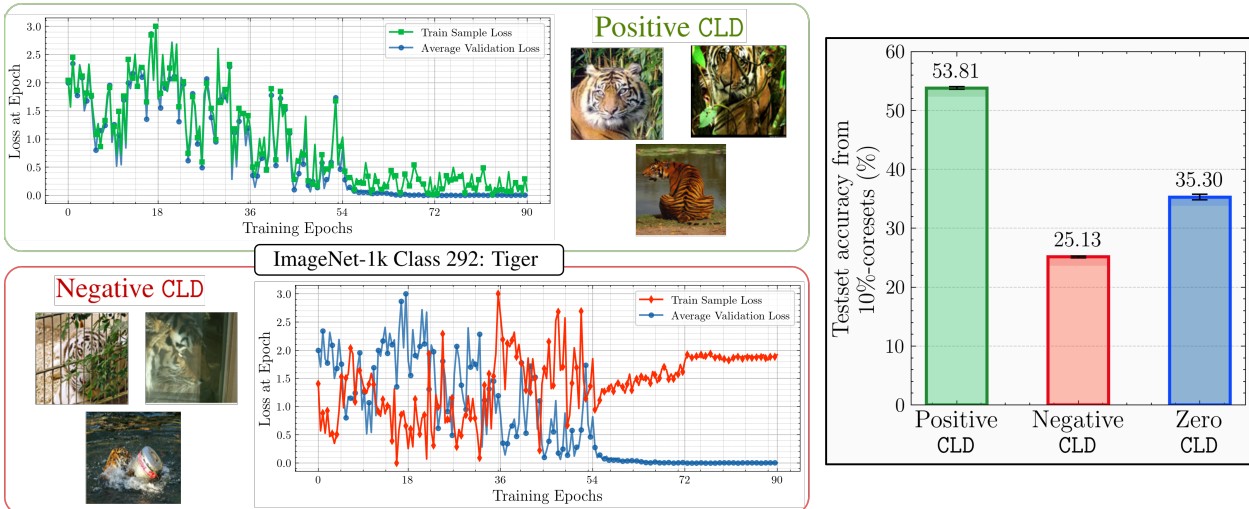

Figure 1: **Correlation of Loss Differences (CLD) at a glance. Left:** *ImageNet-1k* "Tiger" samples illustrating varying CLD scores. High-CLD samples (top row) closely track the validation loss trajectory, indicating informative and representative data. Low/negative-CLD samples (bottom row) significantly deviate, typically corresponding to atypical, ambiguous, or mislabeled examples. **Right:** Performance comparison of coresets formed by selecting equal-sized subsets of the highest 10% positive, lowest 10% negative, and 10% zero-valued CLD samples of ImageNet-1k on ResNet-18. Coresets with high-positive CLD samples achieve superior accuracy over various seeds.

***Score-based methods*** select training samples according to predefined metrics, often in the feature space or based on model prediction confidence. Some methods score a sample by its distance to a class center (Rebuffi et al., 2017; Castro et al., 2018; Belouadah et al., 2020), to the class feature median (e.g., Moderate (Xia et al., 2022)), or to other samples (Sener & Savarese, 2018). Cal (Margatina et al., 2021) identifies contrastive examples via the KL divergence between predictive distributions, while Herding (Chen et al., 2010) selects representative samples using a kernel-based approach in Hilbert space. Other methods rely on prediction probabilities from a (possibly proxy) model. For example, Forgetting (Toneva et al., 2018) counts how often a sample is misclassified after previously being correct, GraNd and EL2N (Paul et al., 2021) rank examples by their early-training loss gradient norm or $l_2$ error norm, respectively, and AUM (Pleiss et al., 2020) scores the *area under the margin* across training to flag potential label issues. While these approaches are simple and avoid expensive gradient or Hessian computation, they often inherit biases from their heuristic scoring rules and offer no convergence guarantees. To mitigate bias, methods such as CCS (Zheng et al., 2022) employ stratified sampling for diversity, while $\mathbb{D}^2$-Pruning (Maharana et al., 2024) combines difficulty (prediction variance) and diversity (feature density) in a graph-based framework. Nonetheless, such strategies often make restrictive assumptions to avoid noisy samples (e.g., CCS discards up to 30% of data) and still lack theoretical generalization guarantees.

***Optimization-based methods*** formulate coreset selection as an explicit optimization problem, often with provable convergence guarantees. CRAIG (Mirzasoleiman et al., 2020) and GradMatch (Killamsetty et al., 2021a) select samples that align with the full-data gradient direction, while Glister (Killamsetty et al., 2021b) maximizes held-out validation log-likelihood. Bilevel optimization (Borsos et al., 2020) has been used to leverage influence functions (Koh & Liang, 2017) for selecting samples with maximal generalization benefit, and GraphCut (Iyer et al., 2021) uses submodular information measures as the objective. Recently, BoundarySet-CCS (Yang et al., 2024) minimized decision boundary reconstruction error while ensuring class diversity. While theoretically grounded, these approaches often require repeated optimization loops, making them computationally expensive for large datasets.

***Training property-based methods*** exploit the dynamics of training to assess sample importance. SloCurv (Garg & Roy, 2023) uses second-order loss curvature statistics to identify samples with better generalization potential, and TracIn (Pruthi et al., 2020) tracks gradient alignment with a validation set.

`TDDS` (Zhang et al., 2024) extends this idea by projecting each sample's gradient onto the accumulated gradient to quantify its true contribution, and by monitoring this projection across multiple iterations to account for fluctuations in importance over time. Although effective, such methods depend on costly first-or second-order statistics, limiting scalability.

***Scalable training property-based methods*** reduce this overhead by measuring per-sample dynamics using only forward-pass signals. `Dyn-Unc` (He et al., 2024) summarizes the variability of the true-class probability over sliding windows and prunes by thresholding aggregated uncertainty after training. `DUAL` (Cho et al., 2025) combines this uncertainty score with a difficulty term and uses a pruning-ratio–adaptive *Beta sampling* schedule to reweight selection at high pruning ratios. In contrast, our method `CLD` ranks examples by how well their *loss-difference trajectories* align with the *class-wise validation loss trajectory*, providing an explicit generalization-alignment criterion. All three approaches admit identical minimal logging, one scalar per example per checkpoint (probability for `Dyn-Unc`/`DUAL`; loss for `CLD`), so compute and storage are directly comparable. Unlike uncertainty-only methods, however, `CLD`'s alignment objective supports a convergence guarantee (Section 5), and it avoids ratio-specific sampling schedules and their hyperparameters; we report robustness at extreme cases (e.g., coresets of size 5%) under identical logging (Section 6). *In short,* `CLD` *combines the practicality of score-based metrics (low compute/storage via scalar logging) with a validation-aligned training-dynamics signal that admits a convergence guarantee.*

## 3 Preliminaries and Problem Setup

We consider the supervised learning problem of learning a mapping from the input space to the output space, $\mathbf{X} \to \mathbf{Y}$, where $\mathbf{X} \subseteq \mathbb{R}^d$ and $\mathbf{Y} \subseteq \mathbb{R}$. The training dataset $\mathbf{S}$ consists of $N$ samples drawn from an unknown distribution $\mathbf{D} = \mathbf{X} \times \mathbf{Y}$, with each sample denoted as $\vec{z}_i = (\vec{x}_i, y_i)$. Thus, $\mathbf{S} = \{\vec{z}_1, \dots, \vec{z}_N\}$. Additionally, we assume access to a query (held-out validation) set $\mathbf{V} \sim \mathbf{D}^Q = \{\vec{q}_1, \dots, \vec{q}_Q\}$ containing $Q$ samples, which represents the true distribution $\mathbf{D}$.

A learning algorithm $\mathbf{A}$ (e.g., SGD) is used to train a model with parameters $\theta \in \mathbb{R}^p$ on the training set $\mathbf{S}$ over $T$ iterations. We denote the model parameters at iteration $t$ of training on $\mathbf{S}$ as $\theta_{\mathbf{S}}^t$, where $\theta_{\mathbf{S}}^0$ corresponds to the random initialization prior to the first update. The performance of the model at step $t$ on a sample $\vec{z}_m$ is evaluated using a loss function $\ell(\theta_{\mathbf{S}}^t, \vec{z}_m) : \mathbb{R}^p \times \mathbb{R}^d \to \mathbb{R}$, which quantifies the prediction error on $\vec{z}_m$ at that point in training.

The goal of training is to minimize the *population risk* $R_{\mathbf{D}}(\theta)$,

$$\arg\min_{\theta} R_{\mathbf{D}}(\theta) = \arg\min_{\theta} \mathbb{E}_{\vec{z} \sim \mathbf{D}} [\ell(\theta, \vec{z})]. \tag{1}$$

However, since $\mathbf{D}$ is unknown, we instead minimize the *empirical risk* $\hat{R}(\theta, \mathbf{S})$,

$$\arg\min_{\theta} \hat{R}(\theta, \mathcal{S}) = \arg\min_{\theta} \left( \frac{1}{N} \sum_{m=1}^N \ell(\theta, \vec{z}_m) \right). \tag{2}$$

The gradient of the loss with respect to the parameters $\theta$ at step $t$ for a sample $\vec{z}_i$ is denoted as $\nabla_\theta \ell(\theta_{\mathbf{S}}^t, \vec{z}_i)$. The average gradient over the validation set ($G_{\mathbf{V}}$) is,

$$G_{\mathbf{V}}(\theta_{\mathbf{S}}^t) = \frac{1}{Q} \sum_{j=1}^Q \nabla_\theta \ell(\theta_{\mathbf{S}}^t, \vec{q}_j) \tag{3}$$

## 4 Correlation of Loss Differences (CLD)

We now define the core quantity used in our method, the correlation of per-sample loss trajectories with the validation set.

**Loss Trajectories**   For every sample $\vec{z}$ we record the per-iteration change in loss during the model training run, and collect these $T$ increments in a *loss-difference trajectory*

$$\vec{\Delta}(\vec{z}) \;=\; \left( \ell(\theta_{\mathbf{S}}^1, \vec{z}) - \ell(\theta_{\mathbf{S}}^0, \vec{z}), \; \ldots, \; \ell(\theta_{\mathbf{S}}^T, \vec{z}) - \ell(\theta_{\mathbf{S}}^{T-1}, \vec{z}) \right) \in \mathbb{R}^T. \tag{4}$$

The *validation-average trajectory* is defined similarly as,

$$\vec{\Delta}'_{\mathbf{V}} \;=\; \left( \frac{1}{Q} \sum_{j=1}^Q \left[ \ell\left( \theta_{\mathbf{S}}^1, \vec{q}_j \right) - \ell\left( \theta_{\mathbf{S}}^0, \vec{q}_j \right) \right], \; \ldots, \; \frac{1}{Q} \sum_{j=1}^Q \left[ \ell\left( \theta_{\mathbf{S}}^T, \vec{q}_j \right) - \ell\left( \theta_{\mathbf{S}}^{T-1}, \vec{q}_j \right) \right] \right) \in \mathbb{R}^T. \tag{5}$$

**Definition 1** (Correlation of Loss Differences (CLD)). *The* CLD *score of a training sample $\vec{z}_m \in \mathbf{S}$ is the correlation between the sample's loss trajectory $\vec{\Delta}_m$ and the average loss trajectory of the validation set $\mathbf{V}$:*

$$\mathtt{CLD}(\vec{z}_m) \coloneqq \rho\left( \vec{\Delta}_m, \vec{\Delta}'_{\mathbf{V}} \right), \tag{6}$$

where $\rho$ is the correlation metric. In our experiments, we employ Pearson correlation (Pearson, 1895) due to its scale invariance and computational simplicity. Intuitively, the CLD metric quantifies how well a training sample's loss dynamics align with the aggregate loss trajectory of the validation set, which serves as a proxy for generalization behavior. Samples with higher CLD values are deemed more influential and can thus be prioritized for coreset construction. We investigate this hypothesis both theoretically and empirically in the following sections.

While Definition 1 defines CLD using a global validation trajectory, our practical implementation uses class-specific averages to ensure semantic alignment; see Section 4.1.

## 4.1   Coreset Selection Procedure

We first train a source model $\theta_{\mathbf{S}}$ on the full dataset $\mathbf{S}$, recording per-epoch losses for all training and validation samples. This logging piggybacks on the standard training loop; no additional passes over the training set are required. Consequently, all training samples are scored at the chosen checkpoints. We then compute $\vec{\Delta}_m$ and the *class-specific average validation trajectory* $\vec{\Delta}'_{\mathbf{V},c}$ for each class $c$.

$$\vec{\Delta}'_{\mathbf{V},c} \coloneqq \left( \frac{1}{|\mathbf{V}_c|} \sum_{\vec{q}_j \in \mathbf{V}_c} \left[ \ell(\theta_{\mathbf{S}}^1, \vec{q}_j) - \ell(\theta_{\mathbf{S}}^0, \vec{q}_j) \right], \; \ldots, \; \frac{1}{|\mathbf{V}_c|} \sum_{\vec{q}_j \in \mathbf{V}_c} \left[ \ell(\theta_{\mathbf{S}}^T, \vec{q}_j) - \ell(\theta_{\mathbf{S}}^{T-1}, \vec{q}_j) \right] \right) \in \mathbb{R}^T, \tag{7}$$

$$\text{where} \quad \mathbf{V}_c \coloneqq \{ \vec{q}_j \in \mathbf{V} : y_{q_j} = c \}. \tag{8}$$

In accordance with standard practice for coreset selection, we score samples within each class independently. For a training sample $\vec{z}_m \in \mathbf{S}$ with label $y_m = c$, its CLD score is the Pearson correlation between its trajectory and the corresponding class-specific validation trajectory:

$$\mathtt{CLD}(\vec{z}_m) \;\coloneqq\; \rho\left( \vec{\Delta}(\vec{z}_m), \vec{\Delta}'_{\mathbf{V},c} \right) \quad \forall \vec{z}_m : y_m = c. \tag{9}$$

After computing all scores, we select the top-$k_c$ training samples in each class $c$ to form a class-balanced coreset

$$\mathbf{C} \;=\; \bigcup_{c=1}^C \mathbf{C}_c, \qquad \mathbf{C}_c \;=\; \text{Top-}k_c\left( \{ \vec{z}_m \in \mathbf{S} : y_m = c \}, \mathtt{CLD} \right), \tag{10}$$

with total size fixed in advance as $k = \sum_{c=1}^C k_c$. This per-class selection strategy ensures both label balance and stability of dynamics within semantic categories, improving the robustness and interpretability of the resulting coreset.

A key advantage is architectural flexibility. CLD scores can be computed using a proxy model and transferred to larger or deeper architectures. The full coreset selection procedure is summarized in Appendix A.

## 5 Theoretical Analysis of `CLD`-Coresets

We now provide a theoretical justification for selecting high-`CLD` samples, showing that such coresets yield convergence guarantees close to full-data training under the following assumptions.

**Assumption 1** (L-smoothness)**.** *For every fixed sample $\vec{z}$, define $f(\theta) \coloneqq \ell(\theta, \vec{z})$. Then $f$ is L-smooth in $\theta$, i.e.,*

$$f(y) \leq f(x) + \langle \nabla f(x), y - x \rangle + \frac{L}{2} \|y - x\|_2^2, \qquad \forall x, y \in \mathbb{R}^p. \tag{11}$$

*Consequently, both the population risk $R_{\mathbf{D}}(\cdot)$ and the empirical risk $\hat{R}(\cdot)$ are also L-smooth.*

**Assumption 2** (Bounded Gradient Norm)**.** *There exists $B > 0$ such that, for all $\theta$ and every training sample $\vec{z}_m \in \mathbf{S}$ and validation sample $\vec{q}_j \in \mathbf{V}$, $\|\nabla_\theta \ell(\theta, \vec{z}_m)\|_2 \leq B$, and $\|\nabla_\theta \ell(\theta, \vec{q}_j)\|_2 \leq B$. Consequently, for any index set $\mathcal{C} \subseteq \{1, \ldots, N\}$,*

$$\left\| \frac{1}{|\mathcal{C}|} \sum_{m \in \mathcal{C}} \nabla_\theta \ell(\theta, \vec{z}_m) \right\|_2 \leq B, \qquad \|G_{\mathbf{V}}(\theta)\|_2 \leq B, \tag{12}$$

*where $G_{\mathbf{V}}(\theta) \coloneqq \frac{1}{Q} \sum_{j=1}^{Q} \nabla_\theta \ell(\theta, \vec{q}_j)$ is the validation-average gradient.*

**Assumption 3** (Validation Representativeness)**.** *With probability at least $1 - \delta'$, the validation gradient $G_{\mathbf{V}}(\cdot)$ at every iterate $\theta$ encountered during training satisfies*

$$\|G_{\mathbf{V}}(\theta) - \nabla_\theta R_{\mathbf{D}}(\theta)\|_2 \leq \delta, \quad where \quad \delta = \mathcal{O}(B/\sqrt{Q}). \tag{13}$$

These assumptions mirror those commonly adopted in analyses of training dynamics (Pruthi et al., 2020; Ilyas et al., 2022; Veen et al., 2020; Clemmensen & Kjærsgaard, 2022; Zhang et al., 2022); we merely state Assumption 3 explicitly for transparency, even though it is typically invoked implicitly and is widely regarded as reasonable.

**Remark 1** (Per-Class Validation Trajectories)**.** *In our implementation, we compute `CLD` using class-specific validation trajectories $\vec{\Delta}'_{\mathbf{V},c}$ rather than a single global trajectory. This refinement aligns with standard coreset practices that enforce class balance, which reduces the variance of the correlation estimates by matching each training sample with the validation subset most relevant to its semantic label. The theoretical guarantees stated here continue to hold, as long as the per-class validation subsets satisfy the representativeness condition in Assumption 3 when interpreted class-conditionally.*

**Theorem 1** (Convergence with `CLD`-Coresets)**.** *Consider a gradient descent algorithm trained over $T$ iterations on a training dataset $\mathbf{S}$ with a held-out validation set $\mathbf{V}$. Given Assumptions 1 to 3, let the learning rate satisfy $0 < \eta \leq 1/L$. Let $\theta_{\mathbf{C}}^t$ denote the parameters at iteration $t$ when training on a coreset $\mathbf{C}$.*

*Then, training on the coreset $\mathbf{C}$, consisting of samples with high `CLD` scores:*

$$\mathbf{C} = \left\{ \vec{z}_m \in \mathbf{S} : \text{CLD}(\vec{z}_m) \geq 1 - \epsilon \right\}, \qquad \epsilon > 0, \ \epsilon \to 0, \tag{14}$$

*guarantees that*

$$\min_{0 \leq t < T} \left\| \nabla_\theta R_{\mathbf{D}}(\theta_{\mathbf{C}}^t) \right\|_2^2 \leq \frac{2 \left[ R_{\mathbf{D}}(\theta_{\mathbf{C}}^0) - R_{\inf} \right]}{\eta T} + L\eta B^2 + \left( B\sqrt{2\kappa} + \delta \right)^2, \tag{15}$$

*where $R_{\inf} \coloneqq \inf_\theta R_{\mathbf{D}}(\theta)$, and $\kappa \geq 0$ is an* alignment-gap *term that quantifies the mismatch between the average coreset gradient and the validation proxy gradient $G_{\mathbf{V}}(\theta)$ along training (see Appendix B for the formal definition). Intuitively, $\kappa$ decreases as the selected samples' `CLD` scores increase and as the coreset grows, and $\kappa \to 0$ as $\epsilon \to 0$.*

*Proof Sketch.* The proof is based on the observation that the change in population risk across training steps can be approximated by the inner product between the gradient of the risk and the update direction. A high `CLD` score implies a strong correlation between a sample's loss-change trajectory and that of the validation set, which in turn suggests consistent alignment between the sample's gradient and the validation gradient.

Due to the *L*-smoothness of the loss function, this alignment persists even when training on the coreset, allowing us to bound the cosine similarity between the average coreset gradient and the true risk gradient. This leads to a controlled approximation error in the optimization update. The total error term $(B\sqrt{2\kappa}+\delta)^2$ is governed by three factors: the `CLD` scores of selected samples, the deviation between the coreset and full-data parameter trajectories, and the quality of the validation set as a proxy for the true distribution. Full details and supporting lemmas are provided in Appendix B. □

**Interpreting the Theory**   Under the stated assumptions, training on the full dataset $\mathbf{S}$ yields the convergence bound

$$\min_{0\le t<T}\ \left\|\nabla_\theta R_{\mathbf{D}}(\theta_{\mathbf{S}}^t)\right\|_2^2 \le \frac{2\left[R_{\mathbf{D}}(\theta_{\mathbf{S}}^0)-R_{\inf}\right]}{\eta T}+L\eta B^2. \tag{16}$$

Theorem 1 shows that training on a high-`CLD` coreset achieves a similar bound, up to an additive deviation term $(B\sqrt{2\kappa}+\delta)^2$. This deviation captures the alignment of the coreset with the validation dynamics ($\kappa$), and the representativeness of the validation set ($\delta$), with $\delta=O(B/\sqrt{Q})$ decreasing in the validation size $Q$, and $\kappa$ decreasing as the CLD selection is tightened (smaller $\epsilon$) or as the coreset size $k$ increases (see Remark 3 and Remark 4 in Appendix B).

The alignment term $\kappa$ reflects both the informativeness of selected samples and the size of the coreset. Higher `CLD` scores indicate stronger agreement with validation loss trajectories and thus tighter gradient alignment, reducing $\kappa$. Additionally, larger coresets more faithfully approximate full-data training dynamics, also lowering $\kappa$. We note that making $k$ extremely small can increase trajectory deviation $\|\delta_t\|$, which is reflected inside $\kappa$ (Remark 4). When the coreset size is fixed, the theorem implies that selecting higher-`CLD` samples improves convergence by minimizing this deviation. Thus, `CLD`-based selection emerges as a principled and necessary criterion for preserving the optimization behavior of full-data training.

**Corollary 1** (Necessity of High `CLD` for Good Coresets)**.** *Under the hypotheses of Theorem 1, achieving convergence rates comparable to full-data training* necessarily *requires that the selected samples exhibit near-maximal `CLD` scores and that the validation set provides a reliable proxy for the true risk gradient. Fulfilling these necessary conditions ensures the optimization dynamics induced by the coreset remain well-aligned with those of full-data training.*

## 6   Experimental Evaluation

We evaluate `CLD` empirically, focusing on its effectiveness and transferability.

**Experimental Setup**   We benchmark on CIFAR-100 (Krizhevsky et al., 2009) and ImageNet-1k (Russakovsky et al., 2015). CIFAR-100 has 50,000 training and 10,000 test images across 100 classes; ImageNet-1k has $\sim 1.28$M training images and a 50,000-image validation set across 1,000 classes. For each random seed, we form a *classwise* held-out validation split from the training data (10% for CIFAR-100; 1% for ImageNet-1k), ensuring equal per-class representation; a different split is generated per seed, and the resulting train/validation partitions are reused across all baselines for fairness. Unless otherwise specified, ResNet-18 (He et al., 2016) is the default architecture for `CLD` scoring and for training on selected coresets. Coresets are constructed *per seed* in a class-balanced manner by selecting, within each class, the top-ranked samples under `CLD`. Subset sizes range from 0.2%–100% on CIFAR-100 and 0.1%–100% on ImageNet-1k. We report the mean and standard deviation over 5 independent seeds.

*Baselines.* We compare against representative state-of-the-art methods from three families: *score-based* (`Forgetting` (Toneva et al., 2018), `EL2N` (Paul et al., 2021), and `CCS` (Zheng et al., 2022) using `AUM` (Pleiss et al., 2020)), *optimization-based* (`Glister` (Killamsetty et al., 2021b), $\mathbb{D}^2$-`Pruning` (Maharana et al., 2024)), and *training-property–based* (`TDDS` (Zhang et al., 2024), `SloCurv` (Garg & Roy, 2023), `DUAL` (Cho et al.,

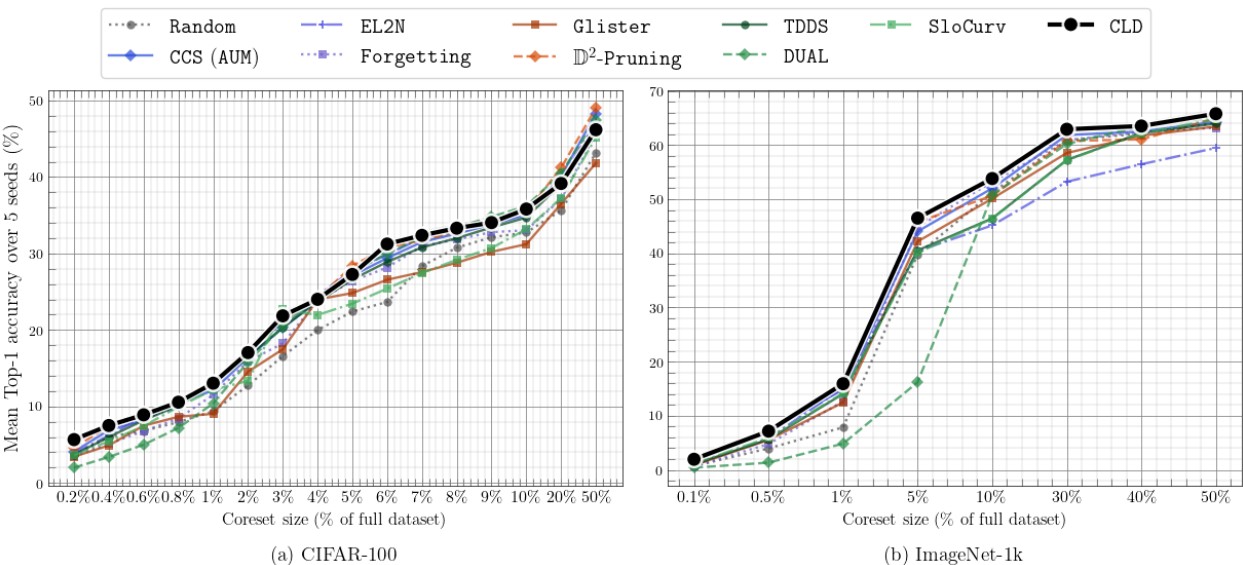

(a) CIFAR-100  (b) ImageNet-1k

Figure 2: Test accuracy (mean over five seeds) for representative coreset selection methods on CIFAR-100 and ImageNet-1k with ResNet-18. CLD consistently matches or outperforms baselines across dataset sizes. *(X-axis uses a non-uniform coreset-size grid.)* Color map: blues (score-based), oranges (optimization-based), greens (training-property-based), black (CLD). For reference, the *full-data mean top-1 accuracy over five seeds* is 70.95 on CIFAR-100 and 69.91 on ImageNet-1k. Complete numerical results are available in Appendix C.

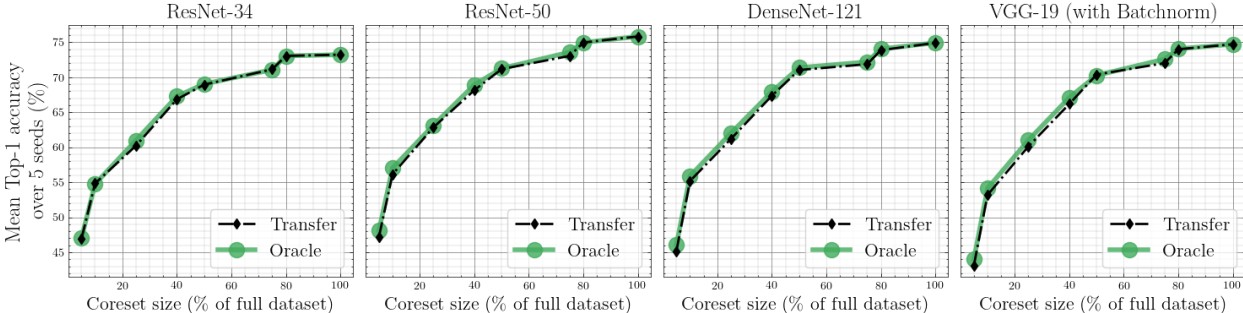

Figure 3: *Transferability of* CLD-*based coresets across architectures on ImageNet-1k.* Each subplot reports test accuracy (mean over five runs) for target models trained on coresets of varying sizes. **Transfer** coresets (dashed black, diamonds) are selected using ResNet-18; **Oracle** coresets (solid green, circles) are computed by the target itself. Transferred coresets are within 1% of oracle coresets across all targets and sizes.

2025)), plus Random. We use implementations from the DeepCore (Guo et al., 2022) library when available, and otherwise rely on official GitHub repositories. All methods are run under a consistent training setup (40 pretraining epochs where required), without any additional fine-tuning or regularization. To ensure fairness, all baselines, including ours, select and train coresets using the same backbone (ResNet-18).

*Transferability protocol.* On ImageNet-1k we additionally test cross-architecture transfer. We compute **Transfer** coresets using ResNet-18 and apply them to ResNet-34, ResNet-50, VGG-19, and DenseNet-121. We compare this to an **Oracle** setting where each target model computes its own CLD scores and coresets from its dynamics.

**Results and Observations**   Figure 2 summarizes performance on CIFAR-100 and ImageNet-1k compared to other methods. Across both datasets, CLD consistently matches or outperforms the strongest baselines from each family. The most competitive alternatives are $\mathbb{D}^2$-Pruning and DUAL, though both degrade at very small coreset sizes. On CIFAR-100, Glister, DUAL, CCS (AUM) can slightly edge out CLD at larger subsets (by $<1\%$), whereas CLD consistently leads on ImageNet-1k. At large subset sizes, CLD converges to full-data

performance with negligible deviation from the strongest baseline. Full numerical tables (and additional methods beyond those plotted) are deferred to Appendix C to avoid clutter. A complementary analysis of the subset fraction required to match full-data accuracy is presented in Appendix D.2.

For cross-architecture transfer on ImageNet-1k, Figure 3 shows that **Transfer** coresets selected with ResNet-18 closely track **Oracle** coresets computed by the target model itself. The gap remains below 1% across ResNet-34, ResNet-50, VGG-19, and DenseNet-121 and across coreset sizes, including transfers across architecture families (ResNet $\rightarrow$ DenseNet/VGG).

**Takeaways** `CLD` achieves near-optimal accuracy across subset sizes while incurring the lowest compute and storage overhead among strong baselines, and its coresets transfer effectively from lightweight proxies to larger targets. Together, these properties make `CLD` a scalable, reliable choice for coreset selection in both single-architecture and cross-architecture regimes.

## 7 Computational and Storage Efficiency

Beyond accuracy, a practical coreset method should keep both *compute* and *storage* costs low. `CLD` does so by relying only on per-sample *loss scalars* that standard training already produces; no per-sample gradients, Hessians, or pairwise similarities are required. Practically, this entails scoring the full training set at a small number of checkpoints, but the scores are exactly the per-sample losses computed during training, i.e., no extra inference sweeps. The only extra work is a forward-only sweep over a small held-out query set each proxy epoch ($Q \ll N$) to record query losses. In contrast, gradient/adversarial methods incur extra backward passes, while similarity/nearest-neighbor methods require feature extraction and large feature caches. We quantify the *end-to-end* compute cost (selection *plus* training on the selected coreset) and the storage overhead in detail in Appendix E and summarize the symbolic complexity below.

**Notation and setup.** We measure compute in floating-point operations (FLOPs) and report storage overheads:

- **Data and epochs.** $N$ training samples, $Q$ query samples; $T$ epochs for the *large* model, $T_{\text{proxy}}$ for the *proxy*; $T_{\text{early}}$ (early scoring), $T_{\text{proxy,early}}$ (early proxy epochs in `DUAL`).
- **Model cost convention.** Large model forward cost $f_{\text{large}}$, proxy forward cost $f$ with $f \ll f_{\text{large}}$. One backward $\approx 2$ forwards $\Rightarrow$ one training step $\approx 3$ forwards per example.
- **Subset/problem.** $k$ coreset size; $d$ input dimension; $c$ classes; $R$ repeats (restarts/probes); $\gamma$ reselection interval.
- **CRAIG embeddings.** $F$ penultimate-feature dimension; $D_{\text{eff}} := F+c$ is the embedding size used by `CRAIG`.
- **Method-specific.** $J$ window length (`Dyn-Unc`/`DUAL`/`TDDS`); $H$ message-passing rounds ($\mathbb{D}^2$-`Pruning`); $\kappa$ $k$NN degree; $U$ unlabeled-pool size (`Cal`); $\gamma_{\text{anc}}$ anchor spacing and $A = T_{\text{proxy}}/\gamma_{\text{anc}}$ anchors (`CRAIG`); $\epsilon$ stochastic-greedy tolerance; $\lambda$ trade-off in `GraphCut`.

**Results and observations (compute).** As summarized in Table 1, methods that score *during early training of the large model* (e.g., `Forgetting`, `EL2N`, `GraNd`) require one or more full sweeps over all $N$ examples with the large network for $T_{\text{early}}$ epochs (and sometimes $R$ repeats), so their selection cost includes terms like $3NT_{\text{early}}R f_{\text{large}}$, making them compute-inefficient even if the coreset used later is small. Optimization-with-reselection methods (e.g., `Glister`) add frequent subset updates every $\gamma$ epochs, driving $\mathcal{O}\big((kQ + N\log(1/\epsilon))\,f_{\text{large}}\,T/\gamma\big)$ on top of $3kT f_{\text{large}}$. Feature/graph–based selectors (`Herding`, `Moderate`, $\mathbb{D}^2$-`Pruning`, `Cal`) pay $3NT_{\text{proxy}}f$ plus at least one $Nf$ encoding pass (sometimes graph/$k$NN work). By contrast, `CLD` uses only proxy training and cheap per-epoch query forwards:

$$\text{Compute}_{\text{CLD}} = 3NT_{\text{proxy}}f + QT_{\text{proxy}}f + 3kT f_{\text{large}},$$

with no gradient/Hessian sweeps, no adversarial steps, and no pairwise similarities.

**Results and observations (storage).** To make storage comparisons transparent, Table 1 reports *selection-stage storage overhead only*, i.e., method-specific extras beyond storing the large model's weights.

Table 1: End-to-end compute and *storage overhead* for CLD and baselines. Storage column lists *method-specific extras during selection only.* Notation in Section 7.

| Method | Computational cost (selection + coreset training) | Storage overhead (method-specific extras) |
|---|---|---|
| Herding | $3NT_{\text{proxy}}f + Nf + \mathcal{O}(Ndk) + 3kTf_{\text{large}}$ | $\mathcal{O}(Nd)$ [features] (+ optional $\mathcal{O}(k^2)$ Gram) |
| Forgetting | $3NT_{\text{early}}f_{\text{large}} + 3kT_{\text{late}}f_{\text{large}}$ | $\mathcal{O}(N)$ [per-sample counter] |
| AUM | $3NT_{\text{proxy}}f + 3kTf_{\text{large}}$ | $\mathcal{O}(N)$ [running sums] |
| Cal | $3kT_{\text{proxy}}f + Uf + \mathcal{O}(U\kappa d) + 3kTf_{\text{large}}$ | $\mathcal{O}(Ud)$ [proxy features] |
| GraNd | $3NT_{\text{early}}Rf_{\text{large}} + 3NT_{\text{early}}Rf_{\text{large}} + 3kT_{\text{late}}f_{\text{large}}$ | $\mathcal{O}(N)$ [scores/logs] |
| EL2N | $3NT_{\text{early}}Rf_{\text{large}} + 3kT_{\text{late}}f_{\text{large}}$ | $\mathcal{O}(N)$ [scores] |
| Moderate | $3NT_{\text{proxy}}f + Nf + \mathcal{O}(Nd + N\log N) + 3kTf_{\text{large}}$ | $\mathcal{O}(Nd)$ [proxy features] |
| $\mathbb{D}^2$-Pruning | $3NT_{\text{proxy}}f + Nf + \mathcal{O}(N\kappa d) + \mathcal{O}(HN\kappa) + 3kTf_{\text{large}}$ | $\mathcal{O}(Nd) + \mathcal{O}(N\kappa)$ [features + $k$NN graph] |
| CRAIG | $3NT_{\text{proxy}}f + \mathcal{O}\big(A\,(Nk + N\log(1/\epsilon))\,D_{\text{eff}}\big) + 3kTf_{\text{large}}$ | $\mathcal{O}(N(F+c))$ [per-anchor embeddings] |
| Glister | $3kTf_{\text{large}} + \mathcal{O}\big((kQ + N\log(1/\epsilon))\,f_{\text{large}}\,T/\gamma\big)$ | $\mathcal{O}(Q)$ [validation cache] |
| GraphCut | $3NT_{\text{proxy}}f + Nf + \mathcal{O}(N^2k) + 3kTf_{\text{large}}$ | $\mathcal{O}(N^2)$ [pairwise similarities] |
| SloCurv | $3NT_{\text{proxy}}f + 3N(R+1)f + 3kTf_{\text{large}}$ | $\mathcal{O}(N) + \mathcal{O}(Rd)$ [running stats + probe dirs] |
| TDDS | $3NT_{\text{proxy}}f + 3NT_{\text{proxy}}f + 3kTf_{\text{large}}$ | $\mathcal{O}(NJ)$ [windowed logs] |
| Dyn-Unc | $3NT_{\text{proxy}}f + 3kTf_{\text{large}}$ | $\mathcal{O}(NJ)$ [windowed logs] |
| DUAL | $3NT_{\text{proxy,early}}f + 3kTf_{\text{large}}$ | $\mathcal{O}(NJ)$ [windowed logs] |
| **CLD (Ours)** | $\mathbf{3NT_{\text{proxy}}f + QT_{\text{proxy}}f + 3kTf_{\text{large}}}$ | $\mathcal{O}\big((\mathbf{N+Q})\mathbf{T}_{\text{proxy}}\big)$ [loss logs] |

Early-training methods (Forgetting, EL2N, GraNd, AUM) need only $\mathcal{O}(N)$ scalars; windowed-uncertainty methods (Dyn-Unc, DUAL, TDDS) add $\mathcal{O}(NJ)$ logs. CRAIG stores $\mathcal{O}(N(F+c))$ embeddings, similarity/feature methods cache $\mathcal{O}(Nd)$ (plus $\mathcal{O}(N\kappa)$ graphs), and GraphCut is $\mathcal{O}(N^2)$. CLD uses only scalar loss logs $\mathcal{O}\big((N+Q)T_{\text{proxy}}\big)$.

**A visual summary.** Figure 4 summarizes the trade-off between accuracy (y-axis) and end-to-end compute (x-axis, log scale), with bubble size proportional to the *selection-stage storage overhead*. Points in the upper-left with small bubbles are closest to the "Pareto-efficient" frontier, combining high accuracy with low compute and storage cost. To provide a concrete, quantitative context for these trade-offs, the plot is generated using an illustrative setup: selecting 10% coresets of ImageNet-1k with a ResNet-18 proxy, then training a ResNet-50 on the chosen coreset (see Appendix E for details). In this setting, methods that score *during early training of the large model* (e.g., GraNd, EL2N) and those with frequent reselection (e.g., Glister) appear far to the right due to large compute costs, while feature/similarity-based selectors (Herding, Moderate, Cal, $\mathbb{D}^2$-Pruning) have large bubbles from $\mathcal{O}(Nd)$ feature caches. CLD lies near the efficient frontier: its selection cost is proxy-only plus lightweight query forwards, and its storage is just scalar loss logs. We also show CLD$_{90}$ (scores derived using loss values from all proxy epochs) and CLD$_{45}$ (first 45 epochs only); the latter cuts selection compute nearly in half while preserving accuracy (discussed in Section 8). This mirrors observations for DUAL, which also achieves minimal accuracy drop by using only early proxy epochs, underscoring that temporal truncation can further improve efficiency without sacrificing performance. For clarity, the figure uses shades of blue for score-based methods, shades of orange for optimization-based methods, shades of green for training-property-based methods, and dark red for CLD.

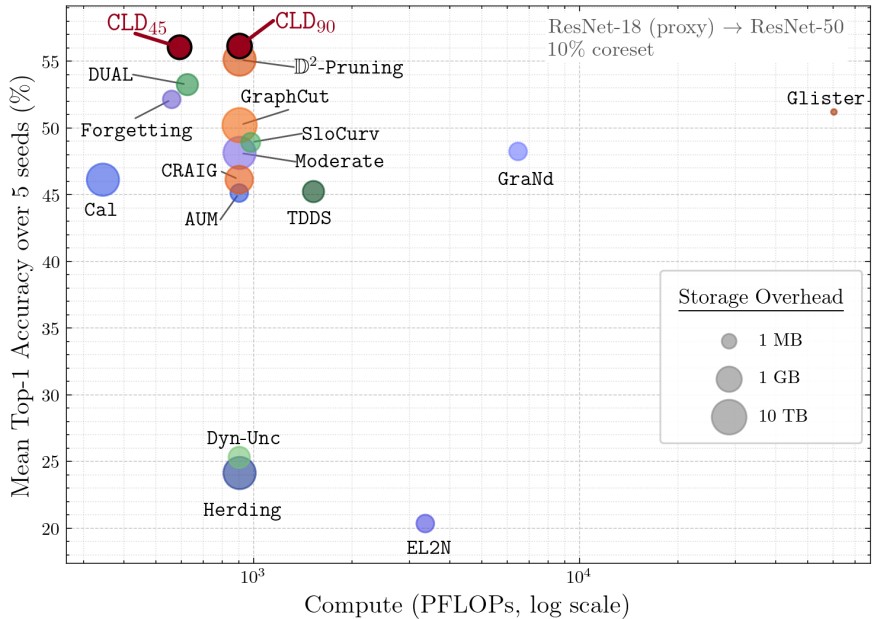

Figure 4: Efficiency summary: Accuracy vs. Compute (x-axis, log scale); bubble size is proportional to the selection-stage storage overhead. The plot uses an illustrative setup for concreteness: selecting 10% coresets of ImageNet-1k with a ResNet-18 proxy and training a ResNet-50 on the coreset (see Appendix E). Both $CLD_{90}$ (all proxy epochs) and $CLD_{45}$ (first 45 proxy epochs) are shown; the latter achieves similar accuracy at roughly half the selection compute. A similar trend is observed for DUAL when restricted to early proxy epochs (which we discuss in Section 8), highlighting that early-epoch scoring can improve efficiency without harming performance.

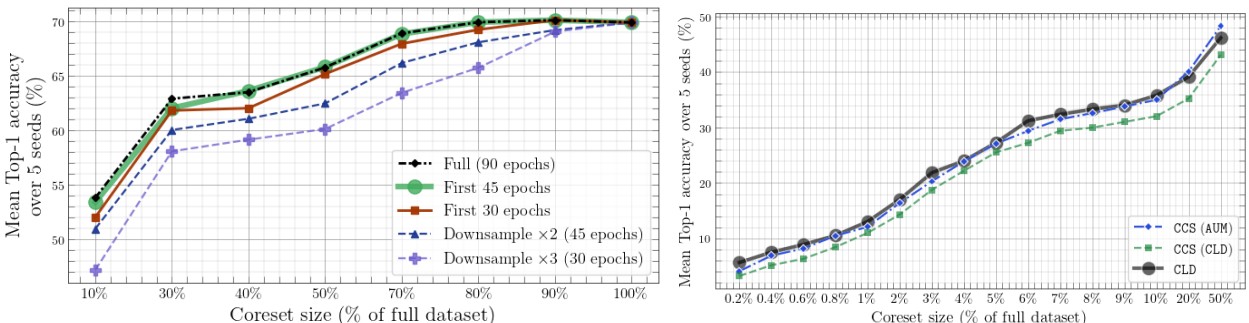

(a) Test accuracy when CLD is computed from early or sub-sampled checkpoints. Stability holds even at coarse resolution. *(X-axis uses a non-uniform coreset-size grid.)*

(b) *Ablation on bias reduction.* CCS-style stratified sampling on top of CLD consistently reduces accuracy across coreset sizes on CIFAR-100.

Figure 5: **Stability and bias.** (a) CLD is stable under reduced temporal resolution; (b) external stratified sampling is unnecessary, and often harmful, for CLD.

## 8 Discussion

We discuss practical considerations and empirical findings that further illustrate the applicability of CLD.

**Stability under temporal subsampling.** We first assess robustness to reduced temporal resolution on ImageNet-1k with ResNet-18. CLD is computed either from the first 30 or 45 checkpoints (out of 90) or from trajectories subsampled at 2× or 3× lower frequency, while training still runs for all 90 epochs. As shown in Figure 5a, using only early checkpoints (30/45) yields accuracy nearly identical to the full-trajectory setting,

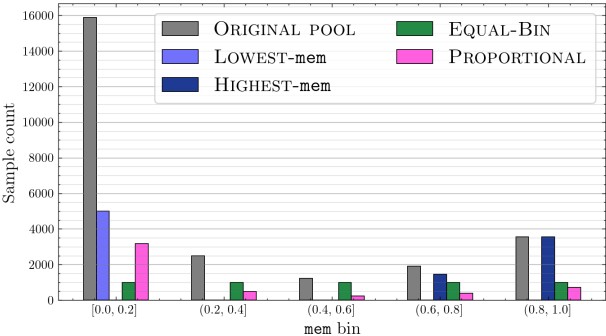
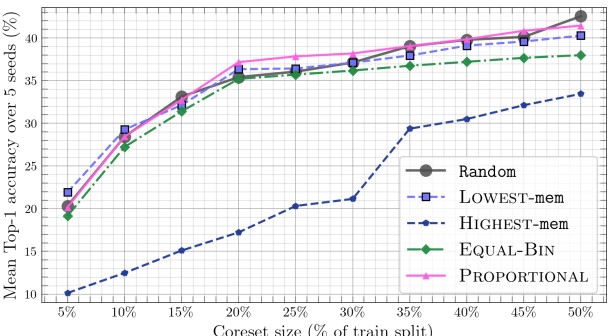

(a) **mem Scores of Validation Sets Tested.** Counts per mem bin for each heuristic (RANDOM, LOWEST-mem, HIGHEST-mem, EQUAL-BIN, PROPORTIONAL); the pool's original distribution is included for reference.

(b) **Effect of Validation Set Composition** CIFAR-100, ResNet-18; mean over five seeds (shaded bands: std). Validation sets are built by the five mem-based heuristics. No label cleaning was applied.

Figure 6: Validation proxy study on CIFAR-100: (a) distributional differences induced by mem-based heuristics; (b) downstream impact on CLD coreset performance.

indicating that the informative signal is captured early in training. 2× subsampling preserves accuracy, and even 3× subsampling incurs only a minor degradation due to reduced temporal resolution.

**Bias reduction and stratified sampling.** Several methods incorporate bias-reduction mechanisms, such as CCS (Zheng et al., 2022), which stratifies selection across score percentiles to promote diversity. In contrast, CLD leverages per-class validation trajectories, yielding a generalization-aware signal that naturally balances classes and downweights noisy or redundant samples. As shown in Figure 5b, applying CCS-style stratified sampling on top of CLD scores consistently *reduces* accuracy across coreset sizes on CIFAR-100. This contrasts with metrics like AUM, where CCS can be beneficial; for CLD, percentile quotas perturb its validation-aligned ranking and reintroduce less informative points.

**Validation proxy: composition and size matter.** Our theoretical guarantees for CLD require that the validation set be a faithful proxy for the test distribution (Assumption 3 in Section 5). To understand this, we ask: *How sensitive is CLD to the composition of the validation set, and does bias in this validation set affect downstream coreset quality?* To probe this, we split CIFAR-100's 50k training set into a classwise 25k/25k train/pool partition. From the pool, we constructed 5000 example validation sets using five heuristics based on publicly available memorization scores (Feldman & Zhang, 2020), denoted mem. Intuitively, low-mem points correspond to canonical, stereotypical examples (often helpful for transfer and exploited by methods such as SloCurv), while high-mem points surface atypical or mislabeled examples. These heuristics, therefore, let us bias the validation set toward "typical" or "atypical" regions of the data. We trained ResNet-18 proxies on the 25k train split, computed CLD with respect to each heuristic-based validation set, built class-balanced coresets, and fine-tuned ResNet-18 across coreset sizes (five seeds; see Figure 6b). As shown in Figure 6a, the non-random heuristics yield validation sets with markedly different mem profiles. Two clear patterns emerge. First, validation sets biased toward HIGHEST-mem examples consistently degrade performance across coreset sizes: overrepresenting atypical or mislabeled samples lowers accuracy and hinders learning. Second, validation sets built from LOWEST-mem examples generally support stronger generalization, but retaining *some* high-mem points is beneficial for capturing long-tail behavior likely present in the test distribution. In our runs, PROPORTIONAL sampling, drawing examples according to the pool's original mem distribution, was the most reliable overall, and would likely improve further if mislabeled points were filtered or downweighted (which we did *not* do in this study). These findings underscore that CLD's effectiveness depends critically on the quality of the validation signal: coresets cannot exceed the fidelity of the validation dynamics they are aligned with. This motivates future work on principled procedures to build clean, representative validation sets (e.g., robust to label noise) and to select validation sizes that balance reliability with efficiency.

Overall, CLD already achieves implicit bias reduction via validation alignment; external stratified sampling is unnecessary and often counterproductive. We provide further results, including seed-wise stability (Ap-

pendix D.1) and connections to influence functions and training data attribution methods (Appendix F), in the appendix.

**Loss Differences as a Gradient-Free Proxy for Influence.** The `CLD` metric is built on correlating per-example *loss differences*, a deliberate choice over simpler signals like raw losses or gradient norms. This choice is theoretically motivated: as we formalize in Lemma 1, a first-order expansion shows that the loss difference $\Delta\ell(\theta^t; z)$ approximates the gradient inner product $\langle\nabla_\theta\ell(\theta^{t-1}; z), \delta\theta^{t-1}\rangle$. This insight positions `CLD` as an efficient proxy for the alignment dynamics tracked by influence methods that compute gradient similarities, such as `TracIn` (Pruthi et al., 2020). The primary advantage of this proxy approach is **computational efficiency**. Methods like `TracIn` are powerful but expensive, requiring the storage and processing of high-dimensional per-sample gradients. `CLD` captures the same underlying alignment dynamics while avoiding this overhead entirely. Simpler signals, such as raw losses or gradient norms, are even cheaper but conceptually flawed, as they are confounded by signal drift or discard crucial directional information. Empirically, `CLD`'s performance is comparable to `TracIn` on key attribution metrics, including the *Linear Datamodeling Score* (a measure of how well scores correlate with true sample influence) and *prediction brittleness* (a measure of how critical top-ranked samples are to model predictions). See Appendix F for the full analysis and a broader discussion of how `CLD` relates to influence methods.

**Beyond Supervised Vision: Scope and Caveats.** Because `CLD` requires only per-example training losses and a small validation proxy, its core criterion can be adapted to other supervised settings (e.g., contrastive learning with a validation loss, or object detection by aggregating per-instance losses to a per-image score). In this work, we intentionally limit our scope to supervised image classification for controlled comparisons and do not claim empirical validation outside this domain. Applying `CLD` to modern language model fine-tuning, however, is non-trivial due to challenges like the "squeezing effect" (Ren & Sutherland, 2025). As fine-tuning progresses, the model concentrates probability mass onto the exact training sequences, which can paradoxically lower the assigned probabilities of similar but non-identical validation samples. This causes a divergence between training and validation loss trajectories, meaning that training samples promoting generalization are not guaranteed to have a high `CLD` score. Corroborating this finding, Xia et al. (2024a) observe that, unlike in vision, minimizing validation loss does not reliably improve model performance in instruction tuning. Adapting a `CLD`-style selector for LLMs will therefore require developing task-appropriate generalization proxies, which is an important direction for future work.

## 9 Limitations

While `CLD` is scalable and effective, it does have limitations. First, it requires access to training loss trajectories across multiple checkpoints, which may not be feasible in settings where models are deployed as black boxes or when fine-tuning from pretrained checkpoints without full retraining. Second, although `CLD` requires a hold-out validation set, this reduces the number of samples for training.

## 10 Conclusion

We introduced *Correlation of Loss Differences* (`CLD`), a simple, scalable metric for identifying data that aligns with generalization. By relying only on per-sample losses across training checkpoints, without gradients, pairwise similarities, or second-order information, `CLD` enables principled coreset selection with low compute and storage cost. Across CIFAR-100 and ImageNet-1k, `CLD` matches or outperforms state-of-the-art methods over a wide range of subset sizes and attains full-data accuracy with substantially smaller subsets. `CLD`-selected coresets also transfer across architectures (ResNet, VGG, DenseNet) with $<1\%$ degradation, remain stable when using only early checkpoints, and inherently reduce bias via per-class validation alignment, obviating additional stratified sampling. Our theory further shows that the convergence gap under coreset training is controlled by sample–validation alignment and the representativeness of the validation set. Taken together, these properties make `CLD` a practical tool for large-scale, budgeted training and a principled foundation for future work on robust validation design and budget-aware data selection.

## Acknowledgements

This work was supported in part by the Center for the Co-Design of Cognitive Systems (CoCoSys), a DARPA-sponsored JUMP 2.0 center, the Semiconductor Research Corporation (SRC), the National Science Foundation, and Collins Aerospace. We are also thankful to Efstathia Soufleri, Akshita Gupta, Utkarsh Saxena, Amitangshu Mukherjee, and Sakshi Choudhary for their helpful discussions and feedback.

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

## A CLD-**Coreset Selection Algorithm**

For completeness, we provide the full pseudocode for the coreset selection procedure described in Section 4.1. This algorithm computes per-class CLD scores by correlating each training sample's loss trajectory with the corresponding class-specific validation trajectory, and selects a fixed number of top-scoring samples per class to form a class-balanced coreset.

---

**Algorithm 1:** Coreset Selection Using CLD (Per-Class)

---

**Input:** Training set $\mathbf{S} = \{\vec{z}_m\}_{m=1}^N$, Validation set $\mathbf{V} = \{\vec{q}_j\}_{j=1}^Q$, Class budgets $\{k_c\}_{c=1}^C$ with $k = \sum_{c=1}^C k_c$,
        Epochs $T$, Initial params $\theta_{\mathbf{S}}^0$, Loss $\ell$, Optimizer $\mathcal{A}$, Hyperparameters $\lambda$

**Output:** Coreset $\mathbf{C}$ of size $k$

1   **for** $t \leftarrow 1$ **to** $T$ **do**
2      $\theta_{\mathbf{S}}^t \leftarrow \mathcal{A}(\theta_{\mathbf{S}}^{t-1}, \mathbf{S}, \lambda)$ // `update model`
3      **for** $m \leftarrow 1$ **to** $N$ **do**
4         Store $\ell(\theta_{\mathbf{S}}^t, \vec{z}_m)$ // `record train loss`
5      **for** $j \leftarrow 1$ **to** $Q$ **do**
6         Store $\ell(\theta_{\mathbf{S}}^t, \vec{q}_j)$ // `record val loss`

7   **for** $m \leftarrow 1$ **to** $N$ **do**
8      $\vec{\Delta}(\vec{z}_m) \leftarrow \left[ \ell(\theta_{\mathbf{S}}^t, \vec{z}_m) - \ell(\theta_{\mathbf{S}}^{t-1}, \vec{z}_m) \right]_{t=1}^T$
9   **for** $j \leftarrow 1$ **to** $Q$ **do**
10     $\vec{\Delta}(\vec{q}_j) \leftarrow \left[ \ell(\theta_{\mathbf{S}}^t, \vec{q}_j) - \ell(\theta_{\mathbf{S}}^{t-1}, \vec{q}_j) \right]_{t=1}^T$
11   **for** $c \leftarrow 1$ **to** $C$ **do**
12     $\mathbf{S}_c \leftarrow \{\vec{z}_m \in \mathbf{S} : y_m = c\}$;
13     $\mathbf{V}_c \leftarrow \{\vec{q}_j \in \mathbf{V} : y_j = c\}$;
14     $\vec{\Delta}'_{\mathbf{V},c} \leftarrow \frac{1}{|\mathbf{V}_c|} \sum_{\vec{q}_j \in \mathbf{V}_c} \vec{\Delta}(\vec{q}_j)$;
15     **foreach** $\vec{z}_m \in \mathbf{S}_c$ **do**
16        $\text{CLD}(\vec{z}_m) \leftarrow \rho\big( \vec{\Delta}(\vec{z}_m), \, \vec{\Delta}'_{\mathbf{V},c} \big)$
17     $\mathbf{C}_c \leftarrow$ top-$k_c$ elements of $\mathbf{S}_c$ by $\text{CLD}(\vec{z}_m)$;
18   $\mathbf{C} \leftarrow \bigcup_{c=1}^C \mathbf{C}_c$;
19   **return C**

---

*Implementation note.* Loss values in Steps 3–6 are logged as part of the normal training loop; no additional passes are introduced.

## B Detailed Theoretical Framework

In this appendix, we provide the complete theoretical framework supporting the results stated in Section 5. We first outline the detailed lemmas establishing gradient alignment and approximation properties of CLD-selected coresets. We then conclude with a full proof of the convergence guarantee presented in Theorem 1.

### B.1 Roadmap and Notation

**Proof Outline.** Our main convergence guarantee, Theorem 1, is built upon three supporting lemmas:

- **Lemma 1**, which establishes that a high CLD score implies strong alignment between a sample's gradient and the validation gradient.

- **Lemma 2**, which shows that this alignment is preserved during coreset training, provided the coreset and full-data parameter trajectories remain close.

- **Lemma 3**, which leverages this stable alignment to bound the approximation error between the average coreset gradient and the true population risk gradient.

**Notation Summary.** We use the following symbols throughout our analysis:

- $L$: The smoothness constant of the loss function.
- $B$: An upper bound on the per-sample gradient norm.
- $k$: The size of the coreset.
- $Q$: The size of the validation set.
- $\epsilon$: A parameter controlling the strictness of CLD-based sample selection.
- $\kappa$: The alignment-gap term, quantifying the gradient mismatch from coreset selection.
- $\delta$: The validation representativeness error, which decays as $O(\sqrt{Q})$.
- $\eta$: The learning rate (step size) of the optimizer.
- $T$: The total number of training iterations.
- $\theta_S^t, \theta_C^t$: Model parameters at iteration $t$ when training on the full dataset and the coreset, respectively.

## B.2 Supporting Lemmas for Theorem 1

**Lemma 1** (High CLD Implies Gradient Alignment). *Consider a training sample $\vec{z}_m \in \mathbf{S}$ with $\mathrm{CLD}(\vec{z}_m) = \rho(\vec{\Delta}(\vec{z}_m), \vec{\Delta}'_{\mathbf{V}}) \geq 1 - \epsilon$ for some small $\epsilon > 0$. Let $\theta_{\mathbf{S}}^t$ be the parameters obtained by running algorithm $\mathcal{A}$ on $\mathbf{S}$ for $t$ iterations. Let $\delta\theta^{t-1} := \theta_{\mathbf{S}}^t - \theta_{\mathbf{S}}^{t-1}$ be the parameter update at step $t$.*

*Suppose the learning algorithm is run for a sufficiently large number of iterations $T$. Assume the sequence of parameter updates $\{\delta\theta^{t-1}\}_{t=1}^T$ is sufficiently varied. This means the updates are not persistently orthogonal to any fixed non-zero vector direction in the relevant parameter subspace.*

*Then, for most training steps $t$ where $G_{\mathbf{V}}(\theta_{\mathbf{S}}^t) \neq 0$, the sample gradient $\nabla_\theta \ell(\theta_{\mathbf{S}}^t, \vec{z}_m)$ and the validation gradient $G_{\mathbf{V}}(\theta_{\mathbf{S}}^t)$ are well-aligned:*

$$\cos\left(\angle\left(\nabla_\theta \ell(\theta_{\mathbf{S}}^t, \vec{z}_m), G_{\mathbf{V}}(\theta_{\mathbf{S}}^t)\right)\right) \geq 1 - \epsilon'_t, \tag{17}$$

*where $\epsilon'_t \to 0$ as $\epsilon \to 0$.*

*Proof Outline.* Use first-order loss changes to relate per-example loss differences and validation loss differences at consecutive steps. High correlation of differences implies an (approximately) positive linear link between their inner products with update directions, forcing the sample gradient to be a positive scalar multiple of the validation gradient under sufficiently varied updates. Continuity then yields $\cos(\angle(\nabla\ell(\cdot, \vec{z}_m), G_{\mathbf{V}})) \geq 1 - \epsilon'_t$ with $\epsilon'_t \to 0$ as $\epsilon \to 0$. $\square$

*Proof.* We first analyze the idealized case where the correlation is perfect ($\epsilon = 0$) and the underlying approximations hold exactly, and then argue by continuity.

Assume the first-order Taylor expansions are exact for the loss changes:

$$\left(\vec{\Delta}(\vec{z}_m)\right)_t = \ell(\theta_{\mathbf{S}}^t, \vec{z}_m) - \ell(\theta_{\mathbf{S}}^{t-1}, \vec{z}_m) \approx \langle \nabla_\theta \ell(\theta_{\mathbf{S}}^{t-1}, \vec{z}_m), \delta\theta^{t-1} \rangle = x_t, \tag{18}$$

$$(\vec{\Delta}'_{\mathbf{V}})_t = \frac{1}{Q}\sum_{j=1}^Q \left(\ell(\theta_{\mathbf{S}}^t, \vec{q}_j) - \ell(\theta_{\mathbf{S}}^{t-1}, \vec{q}_j)\right) \approx \langle G_{\mathbf{V}}(\theta_{\mathbf{S}}^{t-1}), \delta\theta^{t-1} \rangle = y_t. \tag{19}$$

Assume perfect correlation $\rho(\vec{x}, \vec{y}) = 1$.
This implies an exact positive linear relationship $x_t = c\, y_t + K'$ for all $t$, where $c = \sigma_x/\sigma_y > 0$ and $K' = \overline{x} - c\,\overline{y}$. Substituting the definitions of $x_t$ and $y_t$:

$$\langle \nabla_\theta \ell(\theta_{\mathbf{S}}^{t-1}, \vec{z}_m), \delta\theta^{t-1} \rangle = c\, \langle G_{\mathbf{V}}(\theta_{\mathbf{S}}^{t-1}), \delta\theta^{t-1} \rangle + K'. \tag{20}$$

Rearranging yields:

$$\langle \nabla_\theta \ell(\theta_{\mathbf{S}}^{t-1}, \vec{z}_m) - c\, G_{\mathbf{V}}(\theta_{\mathbf{S}}^{t-1}), \delta\theta^{t-1} \rangle = K'. \tag{21}$$

Since the mean of the loss trajectory will be smaller compared to the variance of the terms (losses eventually reduce to 0), it is reasonable to assume $K' \approx 0$.

Thus, for $t = 1, \ldots, T$:

$$\langle \nabla_\theta \ell(\theta_{\mathbf{S}}^{t-1}, \vec{z}_m) - c\, G_{\mathbf{V}}(\theta_{\mathbf{S}}^{t-1}), \delta\theta^{t-1} \rangle = 0. \tag{22}$$

Let $\vec{w}_{t-1} := \nabla_\theta \ell(\theta_{\mathbf{S}}^{t-1}, \vec{z}_m) - c\, G_{\mathbf{V}}(\theta_{\mathbf{S}}^{t-1})$.

The vector $\vec{w}_{t-1}$ is exactly orthogonal to the update direction $\delta\theta^{t-1}$ at each step $t$.

Now, invoke the assumption that the sequence of updates $\{\delta\theta^{t-1}\}_{t=1}^{T}$ is sufficiently varied.

This means the updates are not persistently orthogonal to any fixed non-zero direction $\vec{w}_{t-1}$.

If $\vec{w}_{t-1}$ were non-zero, the variation in updates would eventually yield a $\delta\theta^{t-1}$ such that $\langle \vec{w}_{t-1}, \delta\theta^{t-1} \rangle \neq 0$.

Since the inner product is exactly zero for all $t$ in our idealized case, the only possibility consistent with the sufficient variation assumption is that $\vec{w}_{t-1}$ must be the zero vector. Thus:

$$\vec{w}_{t-1} = \nabla_\theta \ell(\theta_{\mathbf{S}}^{t-1}, \vec{z}_m) - c\, G_{\mathbf{V}}(\theta_{\mathbf{S}}^{t-1}) = \vec{0}. \tag{23}$$

This signifies that the sample gradient is exactly a positive scalar multiple ($c > 0$) of the validation gradient:

$$\nabla_\theta \ell(\theta_{\mathbf{S}}^{t-1}, \vec{z}_m) = c\, G_{\mathbf{V}}(\theta_{\mathbf{S}}^{t-1}). \tag{24}$$

Consequently, the vectors are perfectly collinear and point in the same direction (assuming $G_{\mathbf{V}}(\theta_{\mathbf{S}}^{t-1}) \neq 0$).

The angle $\gamma_{t-1}$ between them is exactly 0. Therefore, in this idealized case:

$$\cos(\gamma_{t-1}) = \cos\left(\angle\left(\nabla_\theta \ell(\theta_{\mathbf{S}}^{t-1}, \vec{z}_m), G_{\mathbf{V}}(\theta_{\mathbf{S}}^{t-1})\right)\right) = 1. \tag{25}$$

This derivation holds under the ideal conditions ($\epsilon = 0$, exact Taylor approx., $K' = 0$).

Since the involved operations are continuous, when the conditions are only approximately met (i.e., $\rho \geq 1 - \epsilon$ with $\epsilon \to 0$, Taylor approx. is good, $K'$ is small), the resulting cosine similarity will be close to 1.

We express this conclusion as

$$\cos(\gamma_{t-1}) \geq 1 - \epsilon'_{t-1}, \tag{26}$$

where the error $\epsilon'_{t-1} \to 0$ as $\epsilon \to 0$.

Assuming this alignment holds for most steps $t$ (implying alignment at step $t$ relies on properties at $t - 1$), the lemma statement follows. $\qquad\square$

**Remark 2** (On Update Sequence Variation). *The assumption regarding the update sequence $\{\delta\theta^{t-1}\}$ is that it exhibits enough variation over the trajectory to ensure that no fixed non-zero vector can remain orthogonal to all updates. This property is weaker than requiring the updates to span the entire parameter space, but it is sufficient for the argument. It essentially prevents the gradient difference vector from hiding in a direction that the optimization process never explores. Stochastic optimization methods accumulating updates over many iterations (large $T$) are often expected to satisfy this sufficient variation condition.*

**Lemma 2** (Stability of Gradient Alignment). *Suppose the conditions in Theorem 1 hold: specifically, $L$-smoothness and bounded gradients ($\|\nabla_\theta \ell(\theta, \vec{z})\|_2 \leq B$ for all $\vec{z}$). Consider a coreset $\mathbf{C}$ constructed by selecting samples with high* CLD *scores:*

$$\mathbf{C} = \{\vec{z}_m \in \mathbf{S} : \mathtt{CLD}(\vec{z}_m) \geq 1 - \epsilon\}, \quad with \quad |\mathbf{C}| = k\,, \epsilon > 0,\, \epsilon \to 0. \tag{27}$$

*Assume that during training, the difference between the parameter trajectories satisfies $\|\theta_{\mathbf{C}}^t - \theta_{\mathbf{S}}^t\|_2 = \|\delta_t\|_2$ at step $t$.*

*Then, for each sample $\vec{z}_m \in \mathbf{C}$, the cosine similarity between its gradient and the average validation gradient at step $t$ is lower bounded by*

$$\cos\left(\angle\left(\nabla_\theta \ell(\theta_{\mathbf{C}}^t, \vec{z}_m), G_{\mathbf{V}}(\theta_{\mathbf{C}}^t)\right)\right) \geq 1 - \kappa, \tag{28}$$

*where $\kappa = \epsilon'_t + \frac{4L}{B}\|\delta_t\|_2 + \frac{3L^2}{B^2}\|\delta_t\|_2^2$, and $\epsilon'_t \to 0$ as $\epsilon \to 0$.*

*Proof outline.* Compare gradients at $\theta_{\mathbf{C}}^t$ and $\theta_{\mathbf{S}}^t$ using $L$-smoothness to bound deviations by $O(\|\delta_t\|)$. Expand the inner product and bound cross-terms via Cauchy–Schwarz and the gradient-norm bound $B$. Plug the base alignment from Lemma 1 at $\theta_{\mathbf{S}}^t$ to transfer alignment to $\theta_{\mathbf{C}}^t$, yielding the stated $1 - \kappa$ lower bound with linear/quadratic dependence on $\|\delta_t\|$. $\qquad\square$

*Proof.* Let $\omega_m = \nabla_\theta \ell(\theta_{\mathbf{C}}^t, \vec{z}_m) - \nabla_\theta \ell(\theta_{\mathbf{S}}^t, \vec{z}_m)$ and $\omega_{\mathbf{V}} = G_{\mathbf{V}}(\theta_{\mathbf{C}}^t) - G_{\mathbf{V}}(\theta_{\mathbf{S}}^t)$ denote the deviations between gradients evaluated on the coreset trajectory and the full dataset trajectory.

By $L$-smoothness of $\ell(\cdot, \vec{z}_m)$ and of the validation-average loss $\hat{R}_{\mathbf{V}}(\theta) \coloneqq \frac{1}{Q} \sum_{j=1}^{Q} \ell(\theta, \vec{q}_j)$, we have:

$$\|\omega_m\|_2 \leq L \|\delta_t\|_2, \tag{29}$$

$$\|\omega_{\mathbf{V}}\|_2 \leq L \|\delta_t\|_2. \tag{30}$$

Expanding the inner product:

$$\langle \nabla_\theta \ell(\theta_{\mathbf{C}}^t, \vec{z}_m), G_{\mathbf{V}}(\theta_{\mathbf{C}}^t) \rangle = \langle \nabla_\theta \ell(\theta_{\mathbf{S}}^t, \vec{z}_m) + \omega_m, G_{\mathbf{V}}(\theta_{\mathbf{S}}^t) + \omega_{\mathbf{V}} \rangle \tag{31}$$

$$= \langle \nabla_\theta \ell(\theta_{\mathbf{S}}^t, \vec{z}_m), G_{\mathbf{V}}(\theta_{\mathbf{S}}^t) \rangle + \langle \omega_m, G_{\mathbf{V}}(\theta_{\mathbf{S}}^t) \rangle + \langle \nabla_\theta \ell(\theta_{\mathbf{S}}^t, \vec{z}_m), \omega_{\mathbf{V}} \rangle + \langle \omega_m, \omega_{\mathbf{V}} \rangle. \tag{32}$$

Applying Cauchy–Schwarz inequality and the bounded gradient norm $\|\nabla_\theta \ell(\theta, \vec{z})\|_2 \leq B$, we have:

$$\langle \omega_m, G_{\mathbf{V}}(\theta_{\mathbf{S}}^t) \rangle \geq -\|\omega_m\|_2 \|G_{\mathbf{V}}(\theta_{\mathbf{S}}^t)\|_2 \geq -LB \|\delta_t\|_2, \tag{33}$$

$$\langle \nabla_\theta \ell(\theta_{\mathbf{S}}^t, \vec{z}_m), \omega_{\mathbf{V}} \rangle \geq -\|\nabla_\theta \ell(\theta_{\mathbf{S}}^t, \vec{z}_m)\|_2 \|\omega_{\mathbf{V}}\|_2 \geq -LB \|\delta_t\|_2, \tag{34}$$

$$\langle \omega_m, \omega_{\mathbf{V}} \rangle \geq -\|\omega_m\|_2 \|\omega_{\mathbf{V}}\|_2 \geq -L^2 \|\delta_t\|_2^2. \tag{35}$$

Thus,

$$\langle \nabla_\theta \ell(\theta_{\mathbf{C}}^t, \vec{z}_m), G_{\mathbf{V}}(\theta_{\mathbf{C}}^t) \rangle \geq \langle \nabla_\theta \ell(\theta_{\mathbf{S}}^t, \vec{z}_m), G_{\mathbf{V}}(\theta_{\mathbf{S}}^t) \rangle - 2LB \|\delta_t\|_2 - L^2 \|\delta_t\|_2^2. \tag{36}$$

The denominator is upper bounded by:

$$\|\nabla_\theta \ell(\theta_{\mathbf{C}}^t, \vec{z}_m)\|_2 \|G_{\mathbf{V}}(\theta_{\mathbf{C}}^t)\|_2 \leq (B + L \|\delta_t\|_2)^2. \tag{37}$$

Combining this result with Lemma 1, we can conclude:

$$\cos\left(\angle\left(\nabla_\theta \ell(\theta_{\mathbf{C}}^t, \vec{z}_m), G_{\mathbf{V}}(\theta_{\mathbf{C}}^t)\right)\right) \geq 1 - \left(\epsilon_t' + \frac{4L}{B} \|\delta_t\|_2 + \frac{3L^2}{B^2} \|\delta_t\|_2^2\right) = 1 - \kappa, \tag{38}$$

where $\epsilon_t'$ captures the initial alignment error when training on $\mathbf{S}$. This completes the proof. $\qquad\square$

**Remark 3.** *Lemma 2 shows that if a sample's gradient is well-aligned with the validation gradient during training on the full dataset (i.e., $\epsilon_t'$ is small), then this alignment is preserved when training on a coreset $\mathbf{C}$, as long as the parameter trajectories $\theta_{\mathbf{S}}^t$ and $\theta_{\mathbf{C}}^t$ remain close. The degradation in alignment is bounded by terms that are linear and quadratic in $\|\delta_t\|_2$. Thus, as long as the coreset trajectory stays near the full dataset trajectory, the generalization-relevant properties captured by* CLD *remain stable. This stability is crucial for ensuring that* CLD*-based coresets maintain the training dynamics of the full dataset.*

**Remark 4** (Influence of Coreset Size on $\kappa$)**.** *It is important to explicitly consider how the coreset size $k = |\mathbf{C}|$ (as specified in Lemma 2) influences the components of $\kappa = \epsilon_t' + \frac{4L}{B} \|\delta_t\|_2 + \frac{3L^2}{B^2} \|\delta_t\|_2^2$.*

- *The term $\epsilon_t'$, representing the initial alignment error derived from Lemma 1, is affected by $k$. A smaller coreset size $k$ allows for a more stringent selection criterion for samples based on their* CLD *scores. Specifically, one can choose only samples with* CLD$(\vec{z}_m)$ *very close to 1, which corresponds to a smaller $\epsilon$ in the selection rule* CLD$(\vec{z}_m) \geq 1 - \epsilon$ *(from Theorem 1 and Lemma 2). A smaller $\epsilon$ naturally leads to a smaller $\epsilon_t'$.*

- *Conversely, the terms in $\kappa$ that depend on $\|\delta_t\|_2 = \|\theta_{\mathbf{C}}^t - \theta_{\mathbf{S}}^t\|_2$ (the deviation between coreset and full-data parameter trajectories) are also influenced by $k$. While a smaller $k$ allows for higher individual sample quality, a very small $k$ might result in a coreset that is less representative of the full dataset $\mathbf{S}$. This reduced representativeness can lead to a larger divergence $\|\delta_t\|_2$ during training, as the optimization trajectory on the small coreset may differ more substantially from that on the full data. An increase in $\|\delta_t\|_2$ would, in turn, increase the overall value of $\kappa$.*

*Therefore, the selection of an appropriate coreset size $k$ involves an inherent trade-off. A smaller $k$ can be beneficial for the $\epsilon'_t$ component of $\kappa$ by enabling the selection of higher-quality samples. However, if $k$ is too small, it could adversely affect the components of $\kappa$ related to $\|\delta_t\|_2$ by making the coreset insufficiently representative. The stability discussed in Remark 3 relies on $\|\delta_t\|_2$ remaining small, highlighting the importance of $k$ being chosen to adequately approximate the full dataset's training dynamics while leveraging the benefits of high `CLD` scores.*

**Lemma 3** (Subset-Gradient Approximation). *Suppose the conditions in Theorem 1 hold, including L-smoothness, bounded gradients, and validation representativeness as described in Section 5.*

*Define the average coreset gradient at step $t$ as*

$$\gamma_{\mathbf{C}}^t = \frac{1}{|\mathbf{C}|} \sum_{m \in \mathbf{C}} \nabla_\theta \ell(\theta_{\mathbf{C}}^t, \vec{z}_m). \tag{39}$$

*Then, for every training step $t$, we have*

$$\left\| \gamma_{\mathbf{C}}^t - \nabla_\theta R_{\mathbf{D}}(\theta_{\mathbf{C}}^t) \right\|_2 \leq B\sqrt{2\kappa} + \delta, \tag{40}$$

*where $\kappa$ captures the alignment error and satisfies $\kappa \to 0$ as $\epsilon \to 0$.*

*Proof outline.* Split the error as $\|\gamma_{\mathbf{C}}^t - G_{\mathbf{V}}\| + \|G_{\mathbf{V}} - \nabla R_{\mathbf{D}}\|$. The second term is $\delta$ by validation representativeness. For the first, average the per-sample deviations and apply Jensen to get a mean of squared distances; combine with Lemma 2 to bound each by $2B^2\kappa$, yielding $\|\gamma_{\mathbf{C}}^t - G_{\mathbf{V}}\| \leq B\sqrt{2\kappa}$. $\square$

*Proof.* We decompose the error using the triangle inequality:

$$\left\| \gamma_{\mathbf{C}}^t - \nabla_\theta R_{\mathbf{D}}(\theta_{\mathbf{C}}^t) \right\|_2 \leq \left\| \gamma_{\mathbf{C}}^t - G_{\mathbf{V}}(\theta_{\mathbf{C}}^t) \right\|_2 + \left\| G_{\mathbf{V}}(\theta_{\mathbf{C}}^t) - \nabla_\theta R_{\mathbf{D}}(\theta_{\mathbf{C}}^t) \right\|_2. \tag{41}$$

The second term $\|G_{\mathbf{V}}(\theta_{\mathbf{C}}^t) - \nabla_\theta R_{\mathbf{D}}(\theta_{\mathbf{C}}^t)\|_2$ is bounded by $\delta$ by the validation representativeness assumption.

To bound the first term $\|\gamma_{\mathbf{C}}^t - G_{\mathbf{V}}(\theta_{\mathbf{C}}^t)\|_2$, we apply Jensen's inequality:

$$\left\| \gamma_{\mathbf{C}}^t - G_{\mathbf{V}}(\theta_{\mathbf{C}}^t) \right\|_2^2 = \left\| \frac{1}{|\mathbf{C}|} \sum_{m \in \mathbf{C}} \left( \nabla_\theta \ell(\theta_{\mathbf{C}}^t, \vec{z}_m) - G_{\mathbf{V}}(\theta_{\mathbf{C}}^t) \right) \right\|^2 \tag{42}$$

$$\leq \frac{1}{|\mathbf{C}|} \sum_{m \in \mathbf{C}} \left\| \nabla_\theta \ell(\theta_{\mathbf{C}}^t, \vec{z}_m) - G_{\mathbf{V}}(\theta_{\mathbf{C}}^t) \right\|_2^2. \tag{43}$$

Define $\varphi_m^t$ as the angle between $\nabla_\theta \ell(\theta_{\mathbf{C}}^t, \vec{z}_m)$ and $G_{\mathbf{V}}(\theta_{\mathbf{C}}^t)$. By Lemma 2, we have $\cos \varphi_m^t \geq 1 - \kappa$ for all $m$. Expanding the squared distance:

$$\left\| \nabla_\theta \ell(\theta_{\mathbf{C}}^t, \vec{z}_m) - G_{\mathbf{V}}(\theta_{\mathbf{C}}^t) \right\|_2^2 = \left\| \nabla_\theta \ell(\theta_{\mathbf{C}}^t, \vec{z}_m) \right\|_2^2 + \left\| G_{\mathbf{V}}(\theta_{\mathbf{C}}^t) \right\|_2^2 - 2\langle \nabla_\theta \ell(\theta_{\mathbf{C}}^t, \vec{z}_m), G_{\mathbf{V}}(\theta_{\mathbf{C}}^t) \rangle \tag{44}$$

$$= \left\| \nabla_\theta \ell(\theta_{\mathbf{C}}^t, \vec{z}_m) \right\|_2^{\ 2} + \left\| G_{\mathbf{V}}(\theta_{\mathbf{C}}^t) \right\|_2^2 - 2 \left\| \nabla_\theta \ell(\theta_{\mathbf{C}}^t, \vec{z}_m) \right\|_2 \left\| G_{\mathbf{V}}(\theta_{\mathbf{C}}^t) \right\|_2 \cos \varphi_m^t. \tag{45}$$

Since $\|\nabla_\theta \ell(\theta_{\mathbf{C}}^t, \vec{z}_m)\|_2, \|G_{\mathbf{V}}(\theta_{\mathbf{C}}^t)\|_2 \leq B$ and $\cos \varphi_m^t \geq 1 - \kappa$, we have

$$\left\| \nabla_\theta \ell(\theta_{\mathbf{C}}^t, \vec{z}_m) - G_{\mathbf{V}}(\theta_{\mathbf{C}}^t) \right\|_2^2 \leq 2B^2\kappa. \tag{46}$$

Substituting back into the Jensen bound,

$$\left\|\gamma_{\mathbf{C}}^t - G_{\mathbf{V}}(\theta_{\mathbf{C}}^t)\right\|_2^2 \le 2B^2\kappa. \tag{47}$$

Taking square roots gives

$$\left\|\gamma_{\mathbf{C}}^t - G_{\mathbf{V}}(\theta_{\mathbf{C}}^t)\right\|_2 \le B\sqrt{2\kappa}. \tag{48}$$

Thus, combining the two bounds,

$$\left\|\gamma_{\mathbf{C}}^t - \nabla_\theta R_{\mathbf{D}}(\theta_{\mathbf{C}}^t)\right\|_2 \le B\sqrt{2\kappa} + \delta, \tag{49}$$

as claimed. $\square$

**Remark 5.** *Lemma 3 shows that under mild conditions, the average gradient computed over a coreset selected based on* CLD *remains close to the true risk gradient throughout training. The deviation is controlled by two sources: the alignment error $\kappa$ arising from the selection of high-*CLD *samples, and the validation approximation error $\delta$ due to finite sample size. Consequently, optimization over* CLD*-coresets closely tracks the gradient flow of the full dataset, ensuring that convergence and generalization properties are preserved. This result is crucial for connecting loss trajectory dynamics with practical coreset construction.*

### B.3 Proof for Theorem 1

*Proof outline.* Apply the $L$-smooth descent lemma to $R_{\mathbf{D}}(\theta)$ with the update $\theta_{\mathbf{C}}^{t+1} = \theta_{\mathbf{C}}^t - \eta\gamma_{\mathbf{C}}^t$ to obtain a one-step inequality involving $\langle\nabla R_{\mathbf{D}}, \gamma_{\mathbf{C}}^t\rangle$. Decompose $\gamma_{\mathbf{C}}^t = \nabla R_{\mathbf{D}} + (\gamma_{\mathbf{C}}^t - \nabla R_{\mathbf{D}})$ and bound the error term using Lemma 3: $E_t = \|\gamma_{\mathbf{C}}^t - \nabla R_{\mathbf{D}}\| \le B\sqrt{2\kappa} + \delta$. Control the mixed term by Cauchy–Schwarz + Young; bound $\|\gamma_{\mathbf{C}}^t\|$ by $B$. Sum over $t$ to telescope $R_t - R_{t+1}$ and isolate $\min_t \|\nabla R_{\mathbf{D}}(\theta_{\mathbf{C}}^t)\|^2$, yielding the stated bound with $(B\sqrt{2\kappa} + \delta)^2$ and an $L\eta B^2$ residual. $\square$

*Proof.* Define

$$R_t := R_{\mathbf{D}}(\theta_{\mathbf{C}}^t), \qquad G_t := \nabla_\theta R_{\mathbf{D}}(\theta_{\mathbf{C}}^t), \qquad \gamma_{\mathbf{C}}^t := \frac{1}{|\mathbf{C}|}\sum_{m\in\mathbf{C}}\nabla_\theta\ell(\theta_{\mathbf{C}}^t, \vec{z}_m). \tag{50}$$

By $L$-smoothness of $\ell(\cdot)$, and in extension $R_{\mathbf{D}}(\cdot)$, and $\eta \le 1/L$,

$$R_{t+1} \le R_t + \langle G_t, \theta_{\mathbf{C}}^{t+1} - \theta_{\mathbf{C}}^t\rangle + \frac{L}{2}\left\|\theta_{\mathbf{C}}^{t+1} - \theta_{\mathbf{C}}^t\right\|_2^2. \tag{51}$$

Substituting the model update $\theta_{\mathbf{C}}^{t+1} - \theta_{\mathbf{C}}^t = -\eta\gamma_{\mathbf{C}}^t$:

$$R_{t+1} \le R_t - \eta\langle G_t, \gamma_{\mathbf{C}}^t\rangle + \frac{L\eta^2}{2}\left\|\gamma_{\mathbf{C}}^t\right\|_2^2. \tag{52}$$

Rearranging,

$$\eta\langle G_t, \gamma_{\mathbf{C}}^t\rangle \le R_t - R_{t+1} + \frac{L\eta^2}{2}\left\|\gamma_{\mathbf{C}}^t\right\|_2^2. \tag{53}$$

Decomposing the inner product using the true gradient $G_t$ we get,

$$\langle G_t, \gamma_{\mathbf{C}}^t\rangle = \langle G_t, G_t + (\gamma_{\mathbf{C}}^t - G_t)\rangle = \|G_t\|_2^2 + \langle G_t, \gamma_{\mathbf{C}}^t - G_t\rangle. \tag{54}$$

Substituting this back,

$$\eta\|G_t\|_2^2 \le R_t - R_{t+1} - \eta\langle G_t, \gamma_{\mathbf{C}}^t - G_t\rangle + \frac{L\eta^2}{2}\left\|\gamma_{\mathbf{C}}^t\right\|_2^2. \tag{55}$$

Let $E_t = \|\gamma_{\mathbf{C}}^t - G_t\|_2$.
Lemma 3 under the stated assumptions gives the bound $E_t \le B\sqrt{2\kappa} + \delta$.
By Cauchy–Schwarz inequality,

$$-\eta\langle G_t, \gamma_{\mathbf{C}}^t - G_t\rangle \le \eta\|G_t\|_2\left\|\gamma_{\mathbf{C}}^t - G_t\right\|_2 = \eta\|G_t\|_2 E_t. \tag{56}$$

Young's inequality states that $ab \leq a^2/(2\gamma) + \gamma b^2/2, \quad \forall a, b \geq 0$ and $\gamma > 0$.
By using this inequality with $\gamma = 1$,

$$-\eta\langle G_t, \gamma_{\mathbf{C}}^t - G_t \rangle \ \leq \ \frac{\eta}{2}\|G_t\|_2^2 + \frac{\eta}{2}E_t^2. \tag{57}$$

Substituting this into the inequality in Equation (55),

$$\eta\|G_t\|_2^2 \leq R_t - R_{t+1} + \tfrac{\eta}{2}\|G_t\|_2^2 + \tfrac{\eta}{2}E_t^2 + \frac{L\eta^2}{2}\|\gamma_{\mathbf{C}}^t\|_2^2. \tag{58}$$

Since all gradients are bounded by $B$, their average $\|\gamma_{\mathbf{C}}^t\|_2$ is also bounded by $B$.

$$\therefore \frac{\eta}{2}\|G_t\|_2^2 \ \leq \ R_t - R_{t+1} + \frac{\eta}{2}E_t^2 + \frac{L\eta^2}{2}B^2. \tag{59}$$

Summing from $t = 0$ to $T - 1$:

$$\frac{\eta}{2}\sum_{t=0}^{T-1}\|G_t\|_2^2 \ \leq \ (R_0 - R_T) + \sum_{t=0}^{T-1}\left(\frac{\eta}{2}E_t^2 + \frac{L\eta^2}{2}B^2\right). \tag{60}$$

Let $R_{\inf} = \inf_\theta R_{\mathbf{D}}(\theta)$. Then $R_0 - R_T \leq R_0 - R_{\inf}$. Substituting the bound $E_t \leq B\sqrt{2\kappa} + \delta$:

$$\frac{\eta}{2}\sum_{t=0}^{T-1}\|G_t\|_2^2 \ \leq \ R_{\mathbf{D}}(\theta_{\mathbf{C}}^0) - R_{\inf} + T\frac{\eta}{2}(B\sqrt{2\kappa} + \delta)^2 + T\frac{L\eta^2}{2}B^2. \tag{61}$$

The sum on the left is lower bounded by $T$ times the minimum term, i.e., $\sum_{t=0}^{T-1}\|G_t\|_2^2 \geq T \cdot \min_{0 \leq t < T}\|G_t\|_2^2$.

$$\therefore \frac{\eta T}{2}\min_{0 \leq t < T}\|G_t\|_2^2 \ \leq \ R_{\mathbf{D}}(\theta_{\mathbf{C}}^0) - R_{\inf} + T\frac{\eta}{2}(B\sqrt{2\kappa} + \delta)^2 + T\frac{L\eta^2}{2}B^2. \tag{62}$$

Since $\eta, T > 0$,

$$\min_{0 \leq t < T}\|G_t\|_2^2 \leq \frac{2\big[R_{\mathbf{D}}(\theta_C^0) - R_{\inf}\big]}{\eta T} + (B\sqrt{2\kappa} + \delta)^2 + L\eta B^2. \tag{63}$$

This proves the theorem. $\qquad\square$

## C  Datasets, Models, and Experimental Details

We evaluate our method on two standard image classification benchmarks: CIFAR-100 (Krizhevsky et al., 2009) and ImageNet-1k (Russakovsky et al., 2015). CIFAR-100 consists of 50,000 training and 10,000 test images across 100 classes and is publicly available without licensing restrictions. For ImageNet-1k, we use the official release from `https://image-net.org/download.php`, which is provided under a standard academic research license and requires user agreement to the terms of access.

**Architectures.**  Our experiments employ the following CNN backbones:

- **ResNet-18, ResNet-34, and ResNet-50** He et al. (2016) from `https://pytorch.org/vision/stable/models/resnet.html` (BSD-3-Clause license).

- **VGG-19 with batch normalization** Simonyan & Zisserman (2014) from `https://pytorch.org/vision/stable/models/vgg.html` (BSD-3-Clause license).

- **DenseNet-121** Huang et al. (2017) from `https://pytorch.org/vision/stable/models/densenet.html` (BSD-3-Clause license).

For baseline comparisons, we use the `DeepCore` library Guo et al. (2022) (`https://github.com/PatrickZH/DeepCore`), which provides standardized implementations of several coreset selection techniques and is licensed under MIT. All training and evaluation code was implemented in PyTorch; dependencies including torchvision are MIT/BSD licensed.

**Training Setup for CIFAR-100.** All networks were trained using SGD Bottou (2010) for 164 epochs with an initial learning rate of 0.1, decayed by a factor of 0.1 at epochs 81 and 121. Nesterov momentum Sutskever et al. (2013) with momentum 0.9 was used, along with weight decay $5 \times 10^{-4}$. Standard augmentations included resizing to $32 \times 32$, random cropping with padding $= 4$, random horizontal flips, and normalization.

**Training Setup for ImageNet-1k.** Training followed standard ImageNet protocols: all models were trained with SGD for 90 epochs, with a learning rate of 0.1 decayed by 0.1 at epochs 30 and 60. Nesterov momentum with coefficient 0.9 and weight decay $10^{-4}$ were used. Data augmentations included random resized cropping to $224 \times 224$, horizontal flipping, and normalization.

**Reproducibility.** No fine-tuning or additional regularization was applied to any method, including ours, ensuring fairness in coreset comparisons. All methods used the same validation split as the validation proxy, and the same train split as the full training set available for each seed when scoring training samples. Each experiment was repeated across 5 independent runs with distinct seeds; reported results reflect the mean and standard deviation.

All experiments were conducted on a private compute cluster with access to NVIDIA A40 GPUs (48 GB memory, 300W TDP). All training and evaluation runs were performed in full precision using PyTorch.

**Results.** Performance results for CIFAR-100 and ImageNet-1k coreset experiments are shown in Tables 2, 3, 4, and 5. Cross-architecture transferability results for CLD coresets, as discussed in Section 6, are shown in Table 6.

Table 2: Performance (top-1 accuracy) of score-based coreset methods on CIFAR100 trainsplit. The coresets were selected and finetuned on ResNet-18. The full trainset performance was $\mathbf{70.95 \pm 0.68}$. The mean accuracy over 5 runs, along with their standard deviation, is reported.

| Coreset Sizes | Random | Herding | Forgetting | Cal | EL2N | Moderate | CCS(AUM) | $\mathbb{D}^2$-Pruning | CLD (Ours) |
|---|---|---|---|---|---|---|---|---|---|
| 0.2% | 3.66 ± 0.41 | 2.57 ± 0.52 | 3.52 ± 0.16 | 5.24 ± 0.41 | 3.9 ± 0.45 | 3.8 ± 0.47 | 4.1 ± 0.5 | 4.6 ± 0.52 | **5.67** ± **0.36** |
| 0.4% | 6.03 ± 0.28 | 3.42 ± 0.49 | 5.12 ± 0.53 | 7.46 ± 0.28 | 6.2 ± 0.41 | 6 ± 0.44 | 6.9 ± 0.42 | 7.4 ± 0.46 | **7.51** ± **0.59** |
| 0.6% | 6.8 ± 0.27 | 4.07 ± 0.41 | 6.8 ± 0.18 | 9.12 ± 0.27 | 8.7 ± 0.4 | 8.4 ± 0.42 | 8.2 ± 0.38 | 8.6 ± 0.43 | **8.91** ± **0.41** |
| 0.8% | 7.96 ± 0.79 | 5.14 ± 0.7 | 8.42 ± 0.38 | 10.19 ± 0.79 | 9.8 ± 0.48 | 10.2 ± 0.49 | 10.5 ± 0.55 | 10.5 ± 0.5 | **10.56** ± **0.24** |
| 1% | 9.38 ± 0.41 | 5.32 ± 0.33 | 11.53 ± 0.44 | 13.73 ± 0.41 | 13.2 ± 0.46 | 12.7 ± 0.45 | 12.1 ± 0.49 | 13.1 ± 0.47 | **13.04** ± **0.54** |
| 2% | 12.74 ± 0.28 | 8.29 ± 0.45 | 15.9 ± 0.21 | 16.45 ± 0.28 | 17 ± 0.43 | 16.3 ± 0.43 | 16.4 ± 0.46 | 17.5 ± 0.45 | **17.05** ± **0.22** |
| 3% | 16.58 ± 0.95 | 9.23 ± 0.52 | 18.24 ± 0.32 | 20.05 ± 0.95 | 21.1 ± 0.5 | 20 ± 0.46 | 20.3 ± 0.52 | 21.3 ± 0.49 | **21.85** ± **0.85** |
| 4% | 19.99 ± 1.04 | 11.52 ± 1.16 | 23.82 ± 0.86 | 22.97 ± 1.04 | 24.5 ± 0.47 | 23.2 ± 0.44 | 23.9 ± 0.5 | 24.1 ± 0.46 | **24.01** ± **0.24** |
| 5% | 22.41 ± 0.54 | 13.66 ± 1.35 | 26.38 ± 0.85 | 24.37 ± 0.54 | 27.8 ± 0.44 | 26.1 ± 0.41 | 27.1 ± 0.44 | 28.4 ± 0.43 | **27.26** ± **0.83** |
| 6% | 23.66 ± 0.49 | 15.49 ± 0.92 | 28.16 ± 0.62 | 26.93 ± 0.49 | 29.9 ± 0.4 | 28.4 ± 0.39 | 29.4 ± 0.4 | 30.8 ± 0.4 | **31.24** ± **0.19** |
| 7% | 28.36 ± 0.85 | 18.52 ± 0.56 | 30.95 ± 0.66 | 27.37 ± 0.85 | 32 ± 0.38 | 30.3 ± 0.36 | 31.5 ± 0.38 | 31.7 ± 0.37 | **32.37** ± **0.71** |
| 8% | 30.75 ± 1.12 | 18.52 ± 0.39 | 31.84 ± 0.23 | 28.32 ± 1.12 | 33.2 ± 0.36 | 31.6 ± 0.34 | 32.6 ± 0.36 | 32.9 ± 0.35 | **33.31** ± **0.68** |
| 9% | 32.12 ± 1.48 | 18.52 ± 0.98 | 32.79 ± 0.94 | 29.19 ± 1.48 | 34.4 ± 0.34 | 33 ± 0.33 | 33.8 ± 0.35 | 34.1 ± 0.34 | **34.04** ± **0.74** |
| 10% | 32.75 ± 1.02 | 19.54 ± 0.85 | 33.04 ± 0.65 | 31.02 ± 1.02 | 35.8 ± 0.32 | 34.2 ± 0.31 | 35 ± 0.33 | 35.4 ± 0.32 | **35.81** ± **0.21** |
| 20% | 35.63 ± 0.99 | 35.14 ± 1.06 | 37.12 ± 0.85 | 35.24 ± 0.81 | 39.6 ± 0.28 | 38.7 ± 0.27 | 40.1 ± 0.29 | 41.2 ± 0.28 | **39.15** ± **0.57** |
| 50% | 43.17 ± 1.02 | 44.14 ± 0.4 | 45.78 ± 0.41 | 42.18 ± 0.69 | 47.2 ± 0.24 | 46 ± 0.23 | 48.3 ± 0.25 | 49 ± 0.24 | **46.18** ± **0.13** |
| 75% | 63.21 ± 0.5 | 66.12 ± 0.46 | 66.12 ± 0.61 | 63.05 ± 0.41 | 65.8 ± 0.2 | 65.1 ± 0.19 | 66.9 ± 0.21 | 67.8 ± 0.2 | **68.01** ± **0.82** |

Table 3: Performance (top-1 accuracy) of optimization and training property-based coreset methods on CIFAR100 trainsplit. The coresets were selected and finetuned on ResNet-18. The full trainset performance was **70.95 ± 0.68**. The mean accuracy over 5 runs, along with their standard deviation, is reported.

| Coreset Sizes | Random | CRAIG | Glister | GraphCut | SloCurv | TDDS | Dyn − Unc | DUAL | CLD (Ours) |
|---|---|---|---|---|---|---|---|---|---|
| 0.2% | 3.66 ± 0.41 | 3.5 ± 0.42 | 3.43 ± 0.32 | 5.8 ± 0.24 | 3.62 ± 0.44 | 3.7 ± 0.44 | 2.1 ± 0.55 | 2 ± 0.56 | **5.67 ± 0.36** |
| 0.4% | 6.03 ± 0.28 | 5.8 ± 0.4 | 4.91 ± 0.37 | 7.07 ± 0.39 | 5.46 ± 0.36 | 6 ± 0.41 | 3.5 ± 0.5 | 3.4 ± 0.52 | **7.51 ± 0.59** |
| 0.6% | 6.8 ± 0.27 | 7.9 ± 0.39 | 7.49 ± 0.61 | 8.96 ± 0.31 | 7.44 ± 0.2 | 8.2 ± 0.4 | 5.1 ± 0.48 | 5 ± 0.49 | **8.91 ± 0.41** |
| 0.8% | 7.96 ± 0.79 | 9.8 ± 0.47 | 8.65 ± 0.47 | 9.39 ± 0.81 | 9.96 ± 0.42 | 10.1 ± 0.48 | 7 ± 0.53 | 7.2 ± 0.54 | **10.56 ± 0.24** |
| 1% | 9.38 ± 0.41 | 12.2 ± 0.44 | 9.04 ± 0.43 | 11.86 ± 0.47 | 12.17 ± 0.3 | 12.8 ± 0.45 | 9.5 ± 0.5 | 10.4 ± 0.51 | **13.04 ± 0.54** |
| 2% | 12.74 ± 0.28 | 15.9 ± 0.42 | 14.54 ± 0.41 | 16.95 ± 0.55 | 13.35 ± 0.42 | 16.5 ± 0.43 | 14.2 ± 0.47 | 15.8 ± 0.48 | **17.05 ± 0.22** |
| 3% | 16.58 ± 0.95 | 19.6 ± 0.46 | 17.47 ± 0.32 | 19.21 ± 0.57 | 22.67 ± 0.8 | 20.3 ± 0.47 | 18.9 ± 0.5 | 20.5 ± 0.5 | **21.85 ± 0.85** |
| 4% | 19.99 ± 1.04 | 23 ± 0.44 | 23.99 ± 0.58 | 21.33 ± 0.71 | 21.97 ± 0.55 | 23.7 ± 0.45 | 22.7 ± 0.48 | 24.3 ± 0.47 | **24.01 ± 0.24** |
| 5% | 22.41 ± 0.54 | 25.8 ± 0.4 | 24.83 ± 0.73 | 26.31 ± 0.58 | 23.44 ± 0.71 | 26.6 ± 0.42 | 25.8 ± 0.44 | 27.5 ± 0.43 | **27.26 ± 0.83** |
| 6% | 23.66 ± 0.49 | 28.1 ± 0.38 | 26.57 ± 0.69 | 30.35 ± 0.68 | 25.41 ± 0.43 | 28.9 ± 0.39 | 28.2 ± 0.41 | 30 ± 0.4 | **31.24 ± 0.19** |
| 7% | 28.36 ± 0.85 | 30 ± 0.36 | 27.57 ± 0.84 | 31.63 ± 0.71 | 27.45 ± 0.6 | 30.8 ± 0.37 | 30.3 ± 0.38 | 32.1 ± 0.38 | **32.37 ± 0.71** |
| 8% | 30.75 ± 1.12 | 31.3 ± 0.34 | 28.79 ± 0.8 | 32.22 ± 0.48 | 29.17 ± 0.47 | 32 ± 0.35 | 31.7 ± 0.36 | 33.4 ± 0.35 | **33.31 ± 0.68** |
| 9% | 32.12 ± 1.48 | 32.7 ± 0.32 | 30.22 ± 0.32 | 33.02 ± 0.83 | 30.71 ± 0.57 | 33.4 ± 0.33 | 33 ± 0.34 | 34.8 ± 0.33 | **34.04 ± 0.74** |
| 10% | 32.75 ± 1.02 | 34.1 ± 0.3 | 31.22 ± 1.33 | 34.41 ± 0.96 | 33.17 ± 1.16 | 34.7 ± 0.31 | 34.4 ± 0.32 | 36.2 ± 0.31 | **35.81 ± 0.21** |
| 20% | 35.63 ± 0.99 | 38.2 ± 0.26 | 36.49 ± 0.47 | 38.02 ± 0.52 | 37.23 ± 0.81 | 39.1 ± 0.27 | 38.8 ± 0.28 | 40.5 ± 0.27 | **39.15 ± 0.57** |
| 50% | 43.17 ± 1.02 | 45.6 ± 0.22 | 41.81 ± 0.88 | 45.23 ± 0.42 | 45.19 ± 0.42 | 46.5 ± 0.23 | 45.8 ± 0.24 | 47.5 ± 0.23 | **46.18 ± 0.13** |
| 75% | 63.21 ± 0.5 | 64.8 ± 0.19 | 63.85 ± 1.02 | 65.18 ± 0.32 | 66.53 ± 0.61 | 65.9 ± 0.2 | 64.5 ± 0.2 | 66.2 ± 0.19 | **68.01 ± 0.82** |

Table 4: Performance (top-1 accuracy) of score-based coreset methods on ImageNet-1k trainsplit. The coresets were selected and finetuned on ResNet-18. The full trainset performance was $\mathbf{69.91 \pm 0.01}$. The mean accuracy over 5 runs, along with their standard deviation, is reported.

| Coreset Sizes | Random | Herding | Forgetting | Cal | EL2N | Moderate | CCS(AUM) | $\mathbb{D}^2$-Pruning | **CLD** |
|---|---|---|---|---|---|---|---|---|---|
| 0.1% | 0.7 | 0.31 | 0.64 | 1.13 | 0.88 | 0.85 | 1.52 | 1.95 | **1.96** |
| | $\pm 0.03$ | $\pm 0.01$ | $\pm 0.01$ | $\pm 0.12$ | $\pm 0.25$ | $\pm 0.2$ | $\pm 0.5$ | $\pm 0.4$ | $\pm \mathbf{0.7}$ |
| 0.5% | 3.98 | 1.39 | 4.78 | 6.84 | 5.83 | 4.75 | 7.04 | 7.2 | **7.16** |
| | $\pm 0.19$ | $\pm 0.17$ | $\pm 1.01$ | $\pm 0.13$ | $\pm 0.1$ | $\pm 0.5$ | $\pm 0.1$ | $\pm 0.25$ | $\pm \mathbf{0.63}$ |
| 1% | 7.86 | 4.32 | 12.67 | 13.17 | 15.2 | 14.32 | 14.86 | 16.01 | **15.92** |
| | $\pm 0.43$ | $\pm 0.62$ | $\pm 0.51$ | $\pm 0.22$ | $\pm 0.5$ | $\pm 1.03$ | $\pm 0.25$ | $\pm 0.4$ | $\pm \mathbf{0.41}$ |
| 5% | 39.78 | 15.36 | 44.86 | 37.65 | 40.43 | 38.95 | 44.04 | 45.75 | **46.5** |
| | $\pm 0.23$ | $\pm 0.18$ | $\pm 0.74$ | $\pm 1.3$ | $\pm 0.03$ | $\pm 0.25$ | $\pm 0.1$ | $\pm 0.5$ | $\pm \mathbf{0.19}$ |
| 10% | 51.24 | 26.84 | 53.19 | 44.16 | 45.16 | 44.56 | 52.01 | 50.65 | **53.81** |
| | $\pm 0.04$ | $\pm 0.05$ | $\pm 0.06$ | $\pm 0.78$ | $\pm 0.4$ | $\pm 0.1$ | $\pm 0.2$ | $\pm 0.3$ | $\pm \mathbf{0.23}$ |
| 30% | 60.87 | 46.61 | 60.9 | 54.41 | 53.22 | 55.29 | 61.84 | 60.75 | **62.91** |
| | $\pm 0.13$ | $\pm 0.87$ | $\pm 0.05$ | $\pm 0.45$ | $\pm 0.25$ | $\pm 0.1$ | $\pm 0.5$ | $\pm 0.1$ | $\pm \mathbf{0.51}$ |
| 40% | 62.13 | 53.88 | 62.39 | 58.45 | 56.45 | 60.08 | 62.48 | 61.04 | **63.51** |
| | $\pm 0.38$ | $\pm 0.23$ | $\pm 0.91$ | $\pm 0.92$ | $\pm 0.1$ | $\pm 0.15$ | $\pm 0.02$ | $\pm 0.5$ | $\pm \mathbf{0.75}$ |
| 50% | 64.11 | 59.14 | 63.18 | 60.11 | 59.46 | 62.58 | 64.31 | 64.92 | **65.78** |
| | $\pm 0.12$ | $\pm 0.41$ | $\pm 0.05$ | $\pm 0.6$ | $\pm 0.45$ | $\pm 0.25$ | $\pm 0.04$ | $\pm 0.65$ | $\pm \mathbf{1.03}$ |
| 65% | 65.21 | 64.23 | 65.24 | 64.42 | 61.28 | 64.98 | 65.17 | 67.01 | **68.02** |
| | $\pm 0.03$ | $\pm 0.2$ | $\pm 0.02$ | $\pm 0.01$ | $\pm 0.25$ | $\pm 0.1$ | $\pm 0.1$ | $\pm 0.1$ | $\pm \mathbf{0.38}$ |
| 70% | 68.81 | 65.22 | 67.85 | 66.57 | 64.23 | 65.18 | 68.81 | 68.91 | **68.91** |
| | $\pm 0.04$ | $\pm 0.02$ | $\pm 0.9$ | $\pm 0.03$ | $\pm 0.89$ | $\pm 0.03$ | $\pm 0.1$ | $\pm 0.01$ | $\pm \mathbf{0.01}$ |
| 75% | 68.41 | 67.42 | 68.01 | 67.12 | 65.45 | 67.13 | 69.01 | 69.42 | **69.42** |
| | $\pm 0.02$ | $\pm 0.01$ | $\pm 0.1$ | $\pm 0.02$ | $\pm 0.2$ | $\pm 0.03$ | $\pm 0.03$ | $\pm 0.05$ | $\pm \mathbf{0.05}$ |
| 80% | 68.12 | 68.02 | 68.81 | 68.15 | 66.95 | 68.9 | 69.93 | 69.93 | **69.93** |
| | $\pm 0.03$ | $\pm 0.02$ | $\pm 0.5$ | $\pm 0.03$ | $\pm 0.25$ | $\pm 0.01$ | $\pm 0.02$ | $\pm 0.02$ | $\pm \mathbf{0.02}$ |
| 85% | 68.75 | 68.01 | 68.81 | 68.91 | 67.17 | 68.9 | 69.91 | 69.93 | **69.93** |
| | $\pm 0.05$ | $\pm 0.03$ | $\pm 0.5$ | $\pm 0.04$ | $\pm 0.1$ | $\pm 0.2$ | $\pm 0.25$ | $\pm 0.02$ | $\pm \mathbf{0.02}$ |
| 90% | 69.1 | 69.92 | 70.04 | 69.23 | 68.81 | 69.01 | 70.12 | 70.12 | **70.12** |
| | $\pm 0.78$ | $\pm 0.4$ | $\pm 0.52$ | $\pm 0.5$ | $\pm 0.3$ | $\pm 0.2$ | $\pm 0.03$ | $\pm 0.03$ | $\pm \mathbf{0.03}$ |
| 95% | 69.91 | 69.91 | 69.91 | 70.12 | 69.91 | 70.12 | 69.91 | 70.12 | **70.12** |
| | $\pm 0.06$ | $\pm 0.04$ | $\pm 0.04$ | $\pm 0.01$ | $\pm 0.04$ | $\pm 0.03$ | $\pm 0.1$ | $\pm 0.03$ | $\pm \mathbf{0.03}$ |

Table 5: Performance (top-1 accuracy) of optimization and training property-based coreset methods on ImageNet-1k trainsplit. The coresets were selected and finetuned on ResNet-18. The full trainset performance was **69.91 ± 0.01**. The mean accuracy over 5 runs, along with their standard deviation, is reported.

| Coreset Sizes | Random | CRAIG | Glister | GraphCut | SloCurv | TDDS | Dyn − Unc | DUAL | **CLD** |
|---|---|---|---|---|---|---|---|---|---|
| 0.1% | 0.7 ± 0.03 | 0.94 ± 0.1 | 0.86 ± 0.01 | 1.09 ± 0.09 | 1.23 ± 0.06 | 1.04 ± 0.2 | 0.45 ± 0.9 | 0.41 ± 1.06 | **1.96 ± 0.7** |
| 0.5% | 3.98 ± 0.19 | 6.41 ± 0.25 | 5.55 ± 0.05 | 7.27 ± 0.03 | 5.89 ± 0.07 | 5.64 ± 0.03 | 1.45 ± 1.05 | 1.34 ± 0.87 | **7.16 ± 0.63** |
| 1% | 7.86 ± 0.43 | 15.56 ± 0.05 | 12.45 ± 0.01 | 14.27 ± 0.31 | 14.17 ± 0.02 | 14.05 ± 0.1 | 3.92 ± 0.8 | 4.85 ± 0.75 | **15.92 ± 0.41** |
| 5% | 39.78 ± 0.23 | 39.95 ± 0.01 | 42.19 ± 0.03 | 39.8 ± 0.6 | 40.1 ± 0.14 | 40.54 ± 0.05 | 15.98 ± 0.5 | 16.2 ± 0.4 | **46.5 ± 0.19** |
| 10% | 51.24 ± 0.04 | 46.76 ± 0.1 | 50.1 ± 0.01 | 48.27 ± 1.02 | 46.39 ± 0.5 | 46.45 ± 0.25 | 29.78 ± 0.6 | 50.75 ± 0.5 | **53.81 ± 0.23** |
| 30% | 60.87 ± 0.13 | 55.41 ± 0.05 | 58.53 ± 0.05 | 61.23 ± 0.01 | 57.19 ± 0.01 | 57.24 ± 0.1 | 50.16 ± 0.5 | 60.19 ± 0.03 | **62.91 ± 0.51** |
| 40% | 62.13 ± 0.38 | 56.55 ± 0.02 | 61.72 ± 0.02 | 63.23 ± 0.08 | 62.11 ± 0.67 | 62.23 ± 0.5 | 60.25 ± 0.2 | 63.45 ± 0.25 | **63.51 ± 0.75** |
| 50% | 64.11 ± 0.12 | 58.56 ± 0.5 | 63.41 ± 0.51 | 65.17 ± 0.05 | 64.78 ± 0.13 | 63.96 ± 0.81 | 63.17 ± 0.1 | 65.21 ± 0.04 | **65.78 ± 1.03** |
| 65% | 65.21 ± 0.03 | 63.41 ± 0.2 | 65.83 ± 0.02 | 67.91 ± 0.54 | 65.81 ± 0.02 | 65.19 ± 0.1 | 65.98 ± 0.25 | 68.31 ± 0.01 | **68.02 ± 0.38** |
| 70% | 68.81 ± 0.04 | 65.21 ± 0.01 | 67.91 ± 0.03 | 68.52 ± 0.04 | 67.18 ± 0.03 | 67.25 ± 0.01 | 66.32 ± 0.1 | 68.76 ± 0.5 | **68.91 ± 0.01** |
| 75% | 68.41 ± 0.02 | 68.01 ± 0.5 | 68.54 ± 0.46 | 68.81 ± 0.02 | 68.03 ± 0.15 | 68.14 ± 0.25 | 68.19 ± 0.01 | 69.92 ± 0.01 | **69.42 ± 0.05** |
| 80% | 68.12 ± 0.03 | 68.01 ± 0.5 | 68.78 ± 0.02 | 69.12 ± 0.05 | 69.1 ± 0.14 | 68.91 ± 0.2 | 68.19 ± 0.01 | 69.92 ± 0.01 | **69.93 ± 0.02** |
| 85% | 68.75 ± 0.05 | 68.78 ± 0.02 | 68.78 ± 0.02 | 70.01 ± 0.28 | 69.1 ± 0.02 | 68.91 ± 0.25 | 69.93 ± 0.2 | 69.93 ± 0.2 | **69.93 ± 0.02** |
| 90% | 69.1 ± 0.78 | 68.48 ± 0.25 | 69.18 ± 0.34 | 70.12 ± 0.43 | 69.68 ± 0.03 | 69.68 ± 0.03 | 70.12 ± 0.03 | 70.12 ± 0.03 | **70.12 ± 0.03** |
| 95% | 69.91 ± 0.06 | 69.91 ± 0.04 | 69.91 ± 0.04 | 70.12 ± 0.43 | 69.91 ± 0.04 | 69.68 ± 0.03 | 70.12 ± 0.03 | 70.12 ± 0.03 | **70.12 ± 0.03** |

Table 6: Cross-architecture performance of coresets of different sizes of ImageNet-1k identified by `CLD`. Each cell reports mean test accuracy (top) and standard deviation (bottom) over 5 runs. Minimal accuracy drop ($< 1\%$) is observed when transferring coresets from a smaller ResNet-18 model to larger or different architectures.

| Target Model | Source Model | Coreset size (% of dataset) | | | | | | | |
|---|---|---|---|---|---|---|---|---|---|
| | | 5% | 10% | 25% | 40% | 50% | 75% | 80% | 100% |
| ResNet-34 | ResNet-18 | 46.91 ± 0.03 | 54.83 ± 0.05 | 60.21 ± 0.12 | 66.83 ± 0.48 | 68.93 ± 0.01 | 71.12 ± 0.01 | 73.04 ± 0.05 | 73.21 ± 0.01 |
| ResNet-34 | ResNet-34 | 47.01 ± 0.02 | 54.75 ± 0.35 | 60.98 ± 0.71 | 67.26 ± 0.10 | 69.03 ± 0.04 | 71.06 ± 0.02 | 73.04 ± 0.01 | 73.21 ± 0.01 |
| ResNet-50 | ResNet-18 | 47.19 ± 0.01 | 56.14 ± 0.05 | 62.83 ± 0.13 | 68.14 ± 0.01 | 71.15 ± 0.03 | 73.04 ± 0.07 | 74.95 ± 0.02 | 75.81 ± 0.05 |
| ResNet-50 | ResNet-50 | 48.10 ± 0.05 | 57.03 ± 0.03 | 63.14 ± 0.02 | 68.91 ± 0.05 | 71.25 ± 0.04 | 73.57 ± 0.01 | 74.95 ± 0.02 | 75.81 ± 0.05 |
| DenseNet-121 | ResNet-18 | 45.18 ± 0.37 | 55.18 ± 0.61 | 61.19 ± 0.01 | 67.34 ± 0.04 | 71.02 ± 0.83 | 71.85 ± 0.04 | 73.85 ± 0.02 | 74.90 ± 0.37 |
| DenseNet-121 | DenseNet-121 | 46.07 ± 0.02 | 55.88 ± 0.03 | 62.01 ± 0.72 | 67.91 ± 0.61 | 71.36 ± 0.10 | 72.18 ± 0.07 | 74.04 ± 0.04 | 74.90 ± 0.37 |
| VGG-19(bn) | ResNet-18 | 43.12 ± 0.04 | 53.18 ± 0.03 | 60.12 ± 0.02 | 66.17 ± 0.07 | 70.37 ± 0.01 | 72.01 ± 0.04 | 73.98 ± 0.37 | 74.70 ± 0.71 |
| VGG-19(bn) | VGG-19(bn) | 44.01 ± 0.50 | 54.12 ± 0.47 | 61.01 ± 0.31 | 67.09 ± 0.16 | 70.25 ± 0.25 | 72.58 ± 0.02 | 74.05 ± 0.04 | 74.70 ± 0.71 |

# D  Additional Ablations

## D.1  Stability across random seeds

We measure sensitivity to random initialization by computing per-example `CLD` scores across five independent seeds on ImageNet-1k (ResNet-18). The pairwise mean absolute error (MAE) between score vectors is consistently below $10^{-5}$, indicating negligible variance and high reproducibility; see Figure 7a.

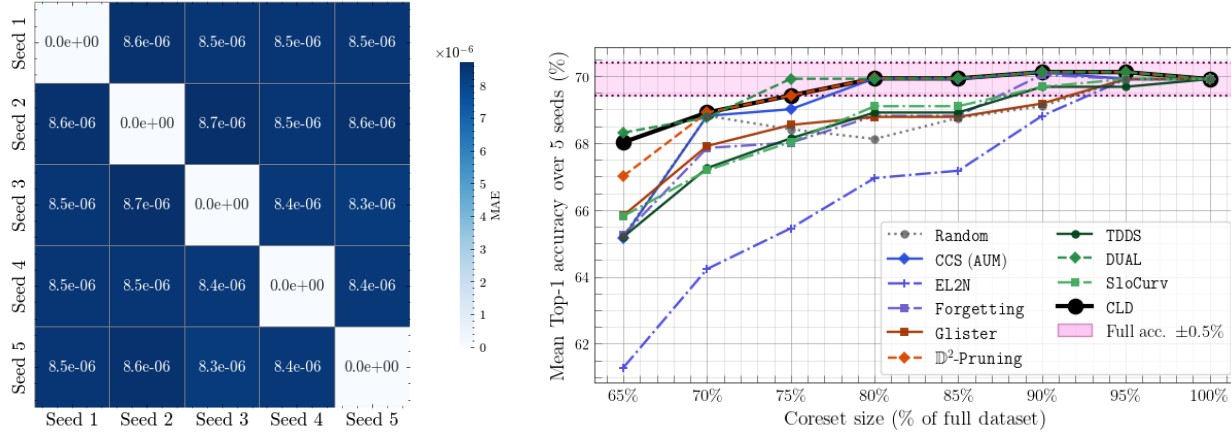

(a) **Seed reproducibility.** Pairwise MAE of `CLD` scores across 5 seeds (lower is better).

(b) **Subset size vs. accuracy.** ImageNet-1k, ResNet-18. The shaded band marks the 0.5% tolerance below full-data accuracy.

Figure 7: Seed-wise stability and subset-size trade-offs on ImageNet-1k.

## D.2 Minimum subset size for full-data accuracy

We quantify the subset fraction required to recover near full-data performance on ImageNet-1k (ResNet-18). As shown in Figure 7b, `CLD` attains test accuracy within 0.5% of the full-data model using only 75% of the training set, on par with $\mathbb{D}^2$-`Pruning` and `DUAL`, and superior to other baselines we evaluated.

# E    Detailed Explanation of Compute and Storage Cost of Coreset Methodologies

**Recap of Notation**   We denote the number of training samples by $N$ ($\mathbf{S} \sim \mathbf{D}^N$) and the number of query (held-out validation) samples by $Q$ ($\mathbf{V} \sim \mathbf{D}^Q$). The model is trained for $T$ epochs. Certain TDA metrics have hyperparameters (denoted $\lambda_\tau$) used to compute the TDA metric $\tau$, which may influence computational cost.

We measure computation in floating-point operations (FLOPs). Let $f_{\text{large}}$ be the cost of a *single-example* forward pass for the large model with $p_{\text{large}}$ parameters, and approximate the backward-pass cost as $2f_{\text{large}}$. When a proxy (smaller) model is used, we write its per-example forward cost and parameter count as $f$ and $p$, with $f \ll f_{\text{large}}$ and $p \ll p_{\text{large}}$.

$R$ is the number of model retrainings (when applicable). $k$ is the coreset size; $d$ is the feature dimensionality; and $B$ is the minibatch size (used during training but not appearing in per-example FLOP counts). Some methods perform subset *reselection* during training; when used, we denote the reselection interval (in epochs) by $\gamma$.

**Reference: full-data training (large model)**   Training the large model on all $N$ points for $T$ epochs costs $3NT f_{\text{large}}$ FLOPs and stores $p_{\text{large}}$ parameters (ignoring optimizer state). Totals below are the end-to-end cost to (i) select a coreset of size $k$ and (ii) train the large model on that coreset.

**Example scenario (used for all plug-in estimates).**   To further illustrate the computational efficiency of `CLD`, we provide approximate cost values by substituting the values of the parameters for finding and training a 10% coreset on the ImageNet-1k dataset.

- $N$=1,268,355 (99% of train), $Q$=12,812 (remaining 1%), $d$=224×224×3, $c$=1000.
- Size of coreset $k$=126,836.
- Proxy encoder: ResNet-18 with $p$=11,689,128 and per-example forward FLOPs $f$=1,818,228,160.
- Large model: ResNet-50 with $p_{\text{large}}$=25,557,032 and $f_{\text{large}} \approx$8,178,000,000.
- We use the standard ImageNet recipe of $T$=$T_{\text{proxy}}$=90 epochs (Yang, 2017).
- When a method has additional parameters, we use the paper's choice (e.g., in `Glister`, $\gamma$=20).

## E.1    Score-based Methods

### E.1.1    Kernel Herding (`Herding`)

`Herding` (Chen et al., 2010) iteratively constructs a representative subset by approximating the data distribution in an RKHS. At iteration $t$, it selects

$$\vec{z}^{*t} \;=\; \underset{\vec{z} \in \mathbf{S}}{\arg\max}\; \langle w_{t-1},\, \phi(\vec{z}) \rangle,$$

where $\phi(\cdot)$ is the kernel feature map and $w_{t-1}$ is an RKHS weight vector. Repeating for $k$ iterations yields a size-$k$ coreset.

**Execution.** One-time selection prior to training the large model (a proxy encoder is used to obtain features):

i) *Train proxy encoder:* train a proxy model for $T_{\text{proxy}}$ epochs (forward cost $f$, parameters $p$).
ii) *Encode full dataset:* extract features for all $N$ points using the trained proxy ($Nf$ FLOPs).
iii) *Herding selection:* for $t$=1:$k$, update candidate scores and select $\vec{z}^{*t}$ using inner products in feature space (explicit features of dim. $d$ give $\mathcal{O}(Nd)$ per iteration, i.e., $\mathcal{O}(Ndk)$ total).
iv) *Train on coreset:* train the large model on the selected $k$ points for $T_{\text{late}}$ epochs (here $T_{\text{late}}$=$T$).

**End-to-end compute (selection + coreset training).**

$$\text{Compute}_{\texttt{Herding}} = \underbrace{3N\,T_{\text{proxy}}\,f}_{\text{train proxy}} + \underbrace{Nf}_{\text{feature extraction}} + \underbrace{\mathcal{O}(Ndk)}_{\text{iterative herding updates}} + \underbrace{3k\,T_{\text{late}}\,f_{\text{large}}}_{\text{train large on coreset}}$$

**Selection-stage storage overhead.** Storing explicit features dominates; caching a $k\times k$ Gram among selected points is optional:

$$\text{StorageOverhead}_{\texttt{Herding}} = \mathcal{O}(Nd) \ \ (+\ \mathcal{O}(k^2) \text{ optional Gram})$$

**Example scenario values (ImageNet-1k; $T_{\textbf{late}}$=90, $T_{\textbf{proxy}}$=90).**

$$\text{Compute}_{\texttt{Herding}} \text{ (PFLOPs only)} \approx \underbrace{622.663}_{3NT_{\text{proxy}}f} + \underbrace{2.306}_{Nf} + \underbrace{280.061}_{3kT_{\text{late}}f_{\text{large}}} = \textbf{905.031} \text{ PFLOPs,}$$

(negligible herding updates)

$$\text{StorageOverhead}_{\texttt{Herding}}(\text{float32}) \approx Nd \times 4 = \textbf{763}.\textbf{692} \text{ GB (features only).}$$

### E.1.2 Example Forgetting (`Forgetting`)

`Forgetting` (Toneva et al., 2018) measures, for each training sample, how many times it transitions from being correctly classified to incorrectly classified during training ("forgetting events"). Examples with higher forgetting counts are ranked as more informative.

**Execution.** One-time selection prior to training the large model on the coreset. Let $T_{\text{early}}$ be the number of early epochs run on the full dataset to collect forgetting statistics, and $T_{\text{late}} = T - T_{\text{early}}$ the remaining epochs used to train on the selected coreset:

   i) Train on all $N$ points for $T_{\text{early}}$ epochs while tracking, per example, the previous correctness bit and a forgetting counter (constant-time update per visit).

   ii) Select the top $k$ examples by forgetting count; train the large model on this coreset for $T_{\text{late}}$ epochs.

**End-to-end compute (selection + coreset training).**

$$\text{Compute}_{\texttt{Forgetting}} = \underbrace{3N\,T_{\text{early}}\,f_{\text{large}}}_{\text{collect forgetting on full data}} + \underbrace{3k\,T_{\text{late}}\,f_{\text{large}}}_{\text{train large on coreset}}$$

**Selection-stage storage overhead.** Streaming the metric requires only one scalar counter (and one correctness bit) per training example:

$$\text{StorageOverhead}_{\texttt{Forgetting}} = \mathcal{O}(N) \ \ (\text{per-sample counter/bit})$$

**Example scenario values (ImageNet-1k; $T_{\textbf{early}}$=10, $T_{\textbf{late}}$=80).**

$$\text{Compute}_{\texttt{Forgetting}} \text{ (PFLOPs only)} \approx \underbrace{311.178}_{3NT_{\text{early}}f_{\text{large}}} + \underbrace{248.943}_{3kT_{\text{late}}f_{\text{large}}} = \textbf{560.122} \text{ PFLOPs,}$$

$$\text{StorageOverhead}_{\texttt{Forgetting}}(\text{float32}) \approx N \times 4 = 5{,}073{,}420 \text{ B} = \textbf{0}.\textbf{005} \text{ GB.}$$

### E.1.3 Area Under Margin (`AUM`)

`AUM` (Pleiss et al., 2020) scores each training sample by aggregating its *margin* over training (e.g., logit of the true class minus the max non-true logit), producing the *area under the margin* across epochs/updates. Higher absolute AUM indicates more consistently confident predictions; lower AUM can flag ambiguous or noisy samples. We compute AUM with a *proxy* model and then train the large model on the selected coreset.

**Execution.** One-time selection with a proxy; the large model then trains on the coreset for all $T$ epochs:

    i) *Train proxy & log margins:* train a proxy for $T_{\text{proxy}}$ epochs on all $N$ samples (per-example forward cost $f$, parameters $p$), recording each sample's margin as it appears in training (no extra forward/backward beyond training).

    ii) *Compute AUM & select:* for each sample, aggregate (e.g., sum/average) its logged margins to obtain AUM and select a size-$k$ coreset according to the desired criterion (e.g., highest AUM, or filter low-AUM points).

    iii) *Train large on coreset:* train the large model on the selected $k$ samples for $T$ epochs.

**End-to-end compute (selection + coreset training).**

$$\text{Compute}_{\texttt{AUM}} = \underbrace{3N\,T_{\text{proxy}}\,f}_{\text{train proxy (margin logging piggybacks)}} + \underbrace{3k\,T\,f_{\text{large}}}_{\text{train large on coreset}}$$

**Selection-stage storage overhead.** AUM can be streamed with a running sum/count per sample:

$$\text{StorageOverhead}_{\texttt{AUM}} = \mathcal{O}(N) \quad \text{(running sums)}$$

**Example scenario values (ImageNet-1k; $T_{\textbf{proxy}}$=90, $T$=90).**

$$\text{Compute}_{\texttt{AUM}} \text{ (PFLOPs only)} \approx \underbrace{622.663}_{3NT_{\text{proxy}}f} + \underbrace{280.061}_{3kT\,f_{\text{large}}} = \textbf{902.724} \text{ PFLOPs,}$$

$$\text{StorageOverhead}_{\texttt{AUM}} \text{ (float32)} \approx N \times 4 = 5{,}073{,}420 \text{ B} = \textbf{0.005} \text{ GB.}$$

### E.1.4 Contrastive Active Learning (`Cal`)

`Cal` (Margatina et al., 2021) acquires unlabeled examples that are near labeled ones in feature space yet differ in predictive probabilities (contrastive pairs), using nearest neighbors over encoder features and a simple divergence-based ranking.

**Execution.** A one-time selection stage is performed prior to training the large model:

    i) train a *proxy* model from scratch on the (growing) labeled set up to size $k$ (per-example forward cost $f \ll f_{\text{large}}$);

    ii) encode all $U$ unlabeled points with the trained proxy (here $U$=$N$);

    iii) run $k$NN-style neighbor search between the unlabeled pool and the labeled set (size $k$), and select a coreset of size $k$.

*(The divergence computation is much cheaper than (ii)–(iii) and is absorbed into big-$\mathcal{O}$.)*

**Overall compute (select once, then train-on-coreset).**

$$\text{Compute}_{\texttt{Cal}} = \underbrace{3k\,T_{\text{proxy}}\,f}_{\text{train proxy}} + \underbrace{U\,f}_{\text{encode unlabeled pool}} + \underbrace{\mathcal{O}(U\,k\,d)}_{k\text{NN over features}} + \underbrace{3kT\,f_{\text{large}}}_{\text{train large on coreset}}$$

**Selection-stage storage overhead.** Storage overhead is due to the cached features for the unlabeled pool

$$\text{StorageOverhead}_{\texttt{Cal}} = \mathcal{O}(Ud) \quad \text{(proxy features)}$$

**Example scenario values: ImageNet-1k; $U$=$N$).**

$$\text{Compute}_{\texttt{Cal}} \text{ (PFLOPs only)} \approx \underbrace{62.267}_{3kT_{\text{proxy}}f} + \underbrace{2.306}_{Uf\ (U=N)} + \underbrace{280.061}_{3kT\,f_{\text{large}}}$$
$$= \textbf{344.634} \text{ PFLOPs,}$$

$$\text{StorageOverhead}_{\texttt{Cal}} \text{ (float32)} \approx Ud \times 4 = \textbf{763.739} \text{ GB.}$$

### E.1.5 Gradient and Error L2 Norm-based Data Pruning (GraNd, EL2N)

GraNd (Paul et al., 2021) ranks training examples by the (expected) per-example gradient norm early in training:

$$\mathrm{GraNd}_t(\vec{z}_m) = \mathbb{E}_{\theta_t} \left\| \nabla_{\theta_t} \ell(\theta_t, \vec{z}_m) \right\|_2^2,$$

averaged over multiple random initializations and early epochs, then retains the top-$k$ examples.

EL2N (Paul et al., 2021) ranks examples by the (expected) L2 error of predictions early in training:

$$\mathrm{EL2N}_t(\vec{z}_m) = \mathbb{E}_{\theta_t} \left\| \mathbf{p}_\theta(\vec{x}_m) - \mathbf{y}_m \right\|_2,$$

where $\mathbf{p}_\theta$ are class probabilities and $\mathbf{y}_m$ is the one-hot label. Scores are computed at small $t$ and optionally averaged over $R$ runs.

**Execution.** One-time selection prior to training the large model. Let $T_{\mathrm{early}}$ be the number of early epochs used for scoring and $T_{\mathrm{late}} = T - T_{\mathrm{early}}$ the remaining epochs used to train on the coreset:

  i) For each of $R$ initializations, train on all $N$ points for $T_{\mathrm{early}}$ epochs (costing $3NT_{\mathrm{early}}f_{\mathrm{large}}$) while logging per-example predictions.
   (a) EL2N: compute scores directly from the logged predictions (no extra passes).
   (b) GraNd: run an *additional scoring sweep* to obtain per-sample gradients (one forward + one backward pass per example per early epoch).
  ii) Average scores across runs; keep the top $k$; train the large model on the coreset for $T_{\mathrm{late}}$ epochs.

**End-to-end compute (selection + train-on-coreset).**

$$\mathrm{Compute}_{\mathtt{EL2N}} = \underbrace{3N\,T_{\mathrm{early}}\,R\,f_{\mathrm{large}}}_{\text{early training (errors reused)}} + \underbrace{3k\,T_{\mathrm{late}}\,f_{\mathrm{large}}}_{\text{train large on coreset}},$$

$$\mathrm{Compute}_{\mathtt{GraNd}} = \underbrace{3N\,T_{\mathrm{early}}\,R\,f_{\mathrm{large}}}_{\text{early training}} + \underbrace{3N\,T_{\mathrm{early}}\,R\,f_{\mathrm{large}}}_{\text{extra per-sample gradient sweeps}} + \underbrace{3k\,T_{\mathrm{late}}\,f_{\mathrm{large}}}_{\text{train large on coreset}}.$$

**Selection-stage storage overhead.** During scoring, a running vector of $N$ scalar scores need to be stored:

$$\mathrm{StorageOverhead}_{\mathtt{EL2N}} = \mathcal{O}(N),$$
$$\mathrm{StorageOverhead}_{\mathtt{GraNd}} = \mathcal{O}(N).$$

**Example scenario values (ImageNet-1k; $R$=10, $T_{\mathbf{early}}$=10, $T_{\mathbf{late}}$=80).**

$$\mathrm{Compute}_{\mathtt{EL2N}} \text{ (PFLOPs only)} \approx \underbrace{3{,}111.782}_{3NT_{\mathrm{early}}R\,f_{\mathrm{large}}} + \underbrace{248.943}_{3kT_{\mathrm{late}}f_{\mathrm{large}}} = \mathbf{3{,}360.725} \text{ PFLOPs,}$$

$$\mathrm{Compute}_{\mathtt{GraNd}} \text{ (PFLOPs only)} \approx \underbrace{3{,}111.782}_{\text{early training}} + \underbrace{3{,}111.782}_{\text{extra gradient sweeps}} + \underbrace{248.943}_{3kT_{\mathrm{late}}\,f_{\mathrm{large}}} = \mathbf{6{,}472.507} \text{ PFLOPs,}$$

$$\mathrm{StorageOverhead} \text{ (float32)} \approx N \times 4 = \mathbf{0.005} \text{ GB for each.}$$

### E.1.6 Using Class Feature Medians (Moderate)

Moderate (Xia et al., 2022) builds a representative coreset by selecting, *within each class*, the samples whose feature-to-center distances are closest to that class's *median* distance (thus avoiding both easy near-center redundancies and far-out outliers). We compute class centers and distances in a proxy feature space.

**Execution.** One-time selection with a proxy, then train the large model on the coreset for all $T$ epochs:

i) *Train proxy:* train a proxy encoder on all $N$ samples for $T_{\text{proxy}}$ epochs (per-example forward cost $f$, parameters $p$).

ii) *Encode dataset:* extract proxy features for all $N$ samples (cost $Nf$ FLOPs).

iii) *Class-median selection:* for each class, compute the class center and all sample distances, then select the per-class quota of samples whose distances are closest to the class-wise median (distance computation $\mathcal{O}(Nd)$; median/quantile selection $\mathcal{O}(N \log N)$ or linear-time selection).

iv) *Train large on coreset:* train the large model on the selected $k$ samples for $T$ epochs.

**End-to-end compute (selection + coreset training).**

$$\text{Compute}_{\text{Moderate}} = \underbrace{3N\,T_{\text{proxy}}\,f}_{\text{train proxy}} + \underbrace{Nf}_{\text{feature extraction}} + \underbrace{\mathcal{O}\big(Nd + N \log N\big)}_{\text{class centers, distances, medians}} + \underbrace{3k\,T\,f_{\text{large}}}_{\text{train large on coreset}}$$

**Selection-stage storage overhead.** The main storage overhead is from caching the features during selection

$$\text{StorageOverhead}_{\text{Moderate}} = \mathcal{O}(Nd) \quad \text{(proxy features)}$$

**Example scenario values (ImageNet-1k; $T_{\textbf{proxy}}$=90, $T$=90).**

$$\text{Compute}_{\text{Moderate}} \text{ (PFLOPs only)} \approx \underbrace{622.663}_{3NT_{\text{proxy}}f} + \underbrace{2.306}_{Nf} + \underbrace{280.061}_{3kT\,f_{\text{large}}} = \textbf{905.031} \text{ PFLOPs,}$$

$$\text{StorageOverhead}_{\text{Moderate}}(\text{float32}) \approx Nd \times 4 = \textbf{763.739} \text{ GB.}$$

### E.1.7 Message Passing ($\mathbb{D}^2 - \texttt{Pruning}$)

$\mathbb{D}^2\texttt{Pruning}$ (Maharana et al., 2024) selects a coreset by *balancing difficulty and diversity* via message passing on a dataset graph built from proxy features. Initial per-sample difficulty scores (from the proxy) are diffused over a $k$NN graph so that each example's score incorporates information from its neighbors; a graph-based sampler then selects a subset that covers diverse yet difficult regions.

**Execution.** One-time selection with a proxy, then train the large model on the coreset for all $T$ epochs:

i) *Train proxy:* train a proxy encoder on all $N$ samples for $T_{\text{proxy}}$ epochs (per-example forward cost $f$, parameters $p$).

ii) *Encode dataset:* extract proxy features for all $N$ samples (cost $Nf$ FLOPs).

iii) *Build graph:* construct a $k$NN graph over the features (e.g., ANN); cost $\mathcal{O}(N\,k\,d)$ (or $\mathcal{O}(N\,d \log N)$).

iv) *Message passing & sampling:* run $H$ rounds of (forward/reverse) message passing on the $N \times k$ edges to update difficulty-aware scores, then sample a size-$k$ coreset (cost $\mathcal{O}(H\,N\,k)$ plus linear-time sampling).

v) *Train large on coreset:* train the large model on the selected $k$ samples for $T$ epochs.

**End-to-end compute (selection + coreset training).**

$$\text{Compute}_{\mathbb{D}^2\texttt{Pruning}} = \underbrace{3N\,T_{\text{proxy}}\,f}_{\text{train proxy}} + \underbrace{Nf}_{\text{feature extraction}} + \underbrace{\mathcal{O}(N\,k\,d) + \mathcal{O}(H\,N\,k)}_{\text{graph build \& message passing}} + \underbrace{3k\,T\,f_{\text{large}}}_{\text{train large on coreset}}$$

**Selection-stage storage overhead.** Caching proxy features dominates; the $k$NN adjacency is linear in $N$ and smaller in practice:

$$\text{StorageOverhead}_{\mathbb{D}^2\texttt{Pruning}} = \mathcal{O}(Nd) + \mathcal{O}(N\kappa) \quad \text{(features + $k$NN graph)}$$

**Example scenario values (ImageNet-1k; $T_{\textbf{proxy}}$=90, $T$=90).**

$$\text{Compute}_{\mathbb{D}^2\texttt{Pruning}} \text{ (PFLOPs only)} \approx \underbrace{622.663}_{3NT_{\text{proxy}}f} + \underbrace{2.306}_{Nf} + \underbrace{280.061}_{3kT\,f_{\text{large}}} = \textbf{905.031} \text{ PFLOPs,} \quad \text{(excluding graph terms)}$$

$$\text{StorageOverhead}_{\mathbb{D}^2\texttt{Pruning}} \text{ (float32)} \approx Nd \times 4 = \textbf{763.692} \text{ GB.}$$

### E.2 Optimization-based Methods

#### E.2.1 Gradient Matching Optimization (`CRAIG`)

`CRAIG` (Mirzasoleiman et al., 2020) selects a subset whose (aggregated) gradients closely match those of the full dataset across training, typically using a submodular (stochastic-greedy) objective over *per-sample gradient embeddings* at selected *anchor* epochs. We compute embeddings with a proxy model and then train the large model on the coreset for all $T$ epochs.

**Execution.** One-time selection with a proxy, then train the large model:

  i) *Train proxy:* train a proxy network on all $N$ samples for $T_{\text{proxy}}$ epochs (per-example forward cost $f$, parameters $p$).
  ii) *Per-sample gradient embeddings at anchors:* every $\gamma_{\text{anc}}$ epochs (anchors $A{=}T_{\text{proxy}}/\gamma_{\text{anc}}$), compute for each sample the *last-layer* gradient embedding using only forward-pass outputs:

$$g_i^{(a)} \;=\; \left(\mathbf{p}_\theta^{(a)}(\vec{x}_i) - \mathbf{y}_i\right) \;\oplus\; h_\theta^{(a)}(\vec{x}_i),$$

  where $h_\theta$ is the penultimate representation (dim. $F$) and $\mathbf{p}_\theta - \mathbf{y}_i \in \mathbb{R}^C$ is the class-probability error; this avoids backward passes (piggybacks on training). Selection then runs stochastic-greedy on these embeddings per anchor.
  iii) *Select & train large:* union the anchor-wise selections to a size-$k$ coreset and train the large model on it for all $T$ epochs.

**End-to-end compute (selection + coreset training).**

$$\text{Compute}_{\texttt{CRAIG}} = \underbrace{3N\,T_{\text{proxy}}\,f}_{\text{train proxy (embeddings piggyback)}} + \underbrace{\mathcal{O}\big(A\,(Nk + N\log(1/\epsilon))\,D_{\text{eff}}\big)}_{\text{stochastic-greedy over anchors}} + \underbrace{3k\,T\,f_{\text{large}}}_{\text{train large on coreset}}$$

Here $D_{\text{eff}} \approx F{+}C$ is the embedding dimensionality (penultimate features and class-probability error); the submodular arithmetic is negligible in FLOPs relative to training and is kept in big-$\mathcal{O}$.

**Selection-stage storage overhead.** We stream anchor processing so only a single anchor's embeddings need be cached at once:

$$\boxed{\text{StorageOverhead}_{\texttt{CRAIG}} = \; \mathcal{O}\big(N(F{+}C)\big) \;\; \text{(per-anchor embeddings, streamed per anchor)}}$$

**Example scenario values (ImageNet-1k; $T_{\mathbf{proxy}}{=}$90, $T{=}$90, $F{=}$512, $C{=}$1000).**

$$\text{Compute}_{\texttt{CRAIG}} \text{ (PFLOPs only; excluding big-}\mathcal{O}\text{ selection)} \approx \underbrace{622.663}_{3NT_{\text{proxy}}f} + \underbrace{280.061}_{3kT\,f_{\text{large}}} = \mathbf{902.724} \text{ PFLOPs,}$$

$$\text{Storage}_{\texttt{CRAIG}} \text{ (float32)} \approx N(F{+}C) \times 4 \;=\; \mathbf{7.671} \text{ GB.}$$

#### E.2.2 Generalization-based Data Subset Selection for Efficient and Robust Learning (`Glister`)

`Glister` (Killamsetty et al., 2021b) selects $\mathbf{S}_j$ of size $k$ via a mixed discrete–continuous bi-level objective:

$$\underset{\mathbf{S}_j \subseteq \mathbf{S},\ |\mathbf{S}_j| \leq k}{\arg\max} \ \mathcal{L}_{\mathbf{V}}\big(\theta^\star(\mathbf{S}_j)\big) \quad \text{where} \quad \theta^\star(\mathbf{S}_j) = \underset{\theta}{\arg\max} \ \mathcal{L}_{\mathbf{S}}(\theta; \mathbf{S}_j).$$

**Execution.** `Glister` *replaces the training loop*: it interleaves training on the current subset with periodic re-selection every $\gamma$ epochs (using stochastic-greedy with a Taylor approximation).

**Overall compute (train-on-coreset).** Training on the coreset over $T$ epochs costs $3kT\,f_{\text{large}}$. Selection across $T$ epochs at frequency $\gamma$ costs $\mathcal{O}\left(\frac{\left(kQ + N\log(1/\epsilon)\right)f_{\text{large}}\,T}{\gamma}\right)$. Hence

$$\boxed{\text{Compute}_{\texttt{Glister}} = 3kT\,f_{\text{large}} + \mathcal{O}\left(\frac{\left(kQ + N\log(1/\epsilon)\right)f_{\text{large}}\,T}{\gamma}\right)}$$

**Selection-stage storage overhead.** Storage overhead is from validation caches:

$$\boxed{\text{StorageOverhead}_{\texttt{Glister}} = \mathcal{O}(Q) \quad \text{(validation cache)}}$$

**Example scenario values:**

$$\text{Compute}_{\texttt{Glister}} \approx \underbrace{280.061}_{3kT\,f_{\text{large}}\ \text{(PFLOPs)}} + \underbrace{60{,}017.420}_{\frac{T}{\gamma}(kQ+N\log(1/\epsilon))\,f_{\text{large}}\ \text{(PFLOPs;}\ \epsilon=0.01,\ \gamma=20)}$$

$$= \mathbf{60{,}297.481}\ \text{PFLOPs},$$

$$\text{StorageOverhead}_{\texttt{Glister}} \approx Q \times 4 = 51{,}248\ \text{B} \approx \mathbf{5.1 \times 10^{-5}}\ \text{GB}\ (\approx 0.05\ \text{MB}).$$

### E.2.3 GraphCut-based Data Subset Selection (`GraphCut`)

`GraphCut` (Iyer et al., 2021) selects $\mathbf{S}_j$ via the generalized graph-cut function

$$f(\mathbf{S}_j) = \lambda \sum_{m \in \mathbf{S}} \sum_{j \in \mathbf{S}_j} s(m,j) - \sum_{j_1,j_2 \in \mathbf{S}_j} s(j_1,j_2), \quad \lambda \geq 2.$$

**Execution.** `GraphCut` *adds* a one-time selection stage prior to training (it does not replace training). The procedure is:

   i) train a proxy model;
   ii) extract features for all $N$ training points using the trained proxy (per-example cost $f \ll f_{\text{large}}$);
   iii) run (stochastic-)greedy selection to build a size-$k$ subset.

*(Similarity operations are typically much cheaper than (ii)–(iii), so we absorb them into big-$\mathcal{O}$.)*

**Overall compute (train-on-coreset).** One-time selection (including proxy training) + large-model training:

$$\boxed{\text{Compute}_{\texttt{GraphCut}} = \underbrace{3N\,T_{\text{proxy}}\,f}_{\text{train proxy}} + \underbrace{Nf}_{\text{feature extraction}} + \underbrace{\mathcal{O}(N^2 k)}_{\text{greedy selection}} + \underbrace{3kT\,f_{\text{large}}}_{\text{train large on coreset}}}$$

**Selection-stage storage overhead.** Storage overhead is from storing pairwise similarities

$$\boxed{\text{StorageOverhead}_{\texttt{GraphCut}} = \mathcal{O}(N^2) \quad \text{(pairwise similarities / kernel)}}$$

**Example scenario values:**

$$\text{Compute}_{\texttt{GraphCut}}\ (\text{PFLOPs only}) \approx \underbrace{622.663}_{3NT_{\text{proxy}}f} + \underbrace{2.306}_{Nf} + \underbrace{280.061}_{3kT\,f_{\text{large}}}$$

$$= \mathbf{905.031}\ \text{PFLOPs},$$

$$\text{StorageOverhead}_{\texttt{GraphCut}}\ (\text{float32}):\ \text{full kernel peak} \approx \mathbf{6{,}434.944}\ \text{GB},$$

$$\text{half kernel peak} \approx \mathbf{3{,}217.493}\ \text{GB}.$$

### E.2.4 Reconstructing the Decision Boundary (`BoundarySet-CCS`)

`BoundarySet-CCS` (Yang et al., 2024) selects samples *near the model's decision boundary* and then enforces *coverage* across distance bands. Distance-to-boundary is approximated per sample by the minimum number of PGD steps required to flip its prediction; `CCS` (coverage-centric sampling) then allocates the coreset budget across bands to preserve distribution coverage.

**Execution.** One-time selection with a proxy, then train the large model on the coreset for all $T$ epochs:

i) *Train proxy:* train a proxy network on all $N$ samples for $T_{\text{proxy}}$ epochs (per-example forward cost $f$, parameters $p$).

ii) *Distance-to-boundary (PGD):* for each sample, run projected gradient steps until misclassification (cap at $K_{\max}$ steps). If the stopping step is $k$, define $d(x) = k$. Each PGD step requires one forward & one backward; we use $3f$ per step in our convention. Let $\bar{K} \leq K_{\max}$ be the average steps per sample.

iii) *CCS selection:* partition samples by $d(x) \in \{0, \ldots, K_{\max}\}$ and allocate the size-$k$ budget across bands (linear-time bucketting and sampling).

iv) *Train large on coreset:* train the large model on the selected $k$ points for *all $T$* epochs.

**End-to-end compute (selection + coreset training).**

$$\text{Compute}_{\texttt{BoundarySet-CCS}} = \underbrace{3N\,T_{\text{proxy}}\,f}_{\text{train proxy}} + \underbrace{3N\,\bar{K}\,f}_{\text{PGD distance-to-boundary sweeps}} + \underbrace{3k\,T\,f_{\text{large}}}_{\text{train large on coreset}}$$

**Selection-stage storage overhead.** Storage is mainly through the scalar distance per sample during selection:

$$\text{StorageOverhead}_{\texttt{BoundarySet-CCS}} = \mathcal{O}(N) \quad (\text{per-sample distance})$$

**Example scenario values (ImageNet-1k; $T_{\textbf{proxy}}$=90, $K_{\max}$=50 so $\bar{K} \approx$50, $T$=90).**

$$\text{Compute}_{\texttt{BoundarySet-CCS}} \text{ (PFLOPs only)} \approx \underbrace{622.663}_{3NT_{\text{proxy}}f} + \underbrace{345.924}_{3N\bar{K}f} + \underbrace{280.061}_{3kT\,f_{\text{large}}} = \textbf{1,248.648} \text{ PFLOPs,}$$

$$\text{StorageOverhead}_{\texttt{BoundarySet-CCS}} \text{ (float32)} \approx N \times 4 = \textbf{0.005} \text{ GB.}$$

### E.3 Training Property-based Methods

#### E.3.1 Samples with Low Loss Curvature (`SloCurv`)

`SloCurv` (Garg & Roy, 2023) scores each training sample by an *input-loss curvature* proxy computed at the end of (proxy) training. For a sample $\vec{z}_m$, with model parameters $\theta^T$ and random Rademacher directions $v_r$ scaled by $h$, the score is

$$\text{Curv}(\vec{z}_m; \theta^T) = \frac{1}{R} \sum_{r=1}^{R} \left\| \nabla_{\vec{z}} \left[ \ell(\theta^T, \vec{z}_m + h v_r) - \ell(\theta^T, \vec{z}_m) \right] \right\|_2^2.$$

Samples with the lowest curvature are retained to form a size-$k$ coreset.

**Execution.** One-time selection prior to training the large model; *a proxy model is used for scoring*:

i) *Train proxy:* train a proxy encoder with per-example forward FLOPs $f$ and parameters $p$ for $T_{\text{proxy}}$ epochs on all $N$ points.

ii) *Curvature scoring:* at the end of proxy training, for each sample compute $\text{Curv}(\vec{z}_m; \theta^T)$ using $R$ Hutchinson repeats. This requires $(R+1)$ gradient evaluations per sample (one at $\vec{z}_m$ and one for each $\vec{z}_m + h v_r$), each costing $\approx (1 \text{ fwd} + 1 \text{ bwd}) \approx 3f$ in our convention.

iii) *Train on coreset:* select the $k$ lowest-curvature samples and train the large model on this coreset for $T_{\text{late}}$=$T$ epochs.

**End-to-end compute (selection + coreset training).**

$$\text{Compute}_{\texttt{SloCurv}} = \underbrace{3N\,T_{\text{proxy}}\,f}_{\text{train proxy}} + \underbrace{3N\,(R+1)\,f}_{\text{curvature scoring at end of proxy training}} + \underbrace{3k\,T_{\text{late}}\,f_{\text{large}}}_{\text{train large on coreset}}$$

**Selection-stage storage overhead.** Storage overhead is from keeping a track of the running curvature values and the directions probed.

$$\text{StorageOverhead}_{\texttt{SloCurv}} = \mathcal{O}(N) + \mathcal{O}(Rd) \quad (\text{running stats} + R \text{ probe dirs})$$

**Example scenario values (ImageNet-1k; $T_{\mathbf{proxy}}$=90, $T_{\mathbf{late}}$=90, $R$=10).**

$$\text{Compute}_{\texttt{SloCurv}} \text{ (PFLOPs only)} \approx \underbrace{622.663}_{3NT_{\text{proxy}}f} + \underbrace{76.103}_{3N(R+1)f} + \underbrace{280.061}_{3kT_{\text{late}}f_{\text{large}}} = \mathbf{978.827} \text{ PFLOPs},$$

$$\text{StorageOverhead}_{\texttt{SloCurv}} \text{ (float32)} \approx N \times 4 + Rd \times 4 = 11{,}094{,}540 \text{ B} = \mathbf{0.011} \text{ GB}.$$

### E.3.2 Temporal Dual-Depth Scoring (`TDDS`)

`TDDS` (Zhang et al., 2024) builds a coreset by combining two temporal depths of signal from training with a *proxy* model. Depth 1 computes, for each epoch, the projection of each sample's *per-sample gradient* onto the epoch's accumulated gradient direction. Depth 2 then aggregates these per-epoch contributions over a sliding window of length $J$ and emphasizes their *temporal variability* (e.g., windowed variance). We maintain windowed statistics in a streaming manner (constant-time updates), so full trajectories need not be stored.

**Execution.** One-time selection with a proxy; the large model then trains on the coreset for the full $T$ epochs:

i) *Train proxy:* train a proxy for $T_{\text{proxy}}$ epochs on all $N$ samples (per-example forward FLOPs $f$, parameters $p$); accumulate the epoch gradient direction.
ii) *Per-sample gradients:* after each proxy epoch, run a scoring sweep to compute per-sample gradients and their projections onto the epoch direction (costing one forward+backward per sample); update the $J$-length windowed statistics and `TDDS` score (streaming).
iii) *Select & train large:* rank by `TDDS` and keep the top-$k$; train the large model on these $k$ samples for *all* $T$ epochs.

**End-to-end compute (selection + coreset training).**

$$\boxed{\text{Compute}_{\texttt{TDDS}} = \underbrace{3N\,T_{\text{proxy}}\,f}_{\text{train proxy}} + \underbrace{3N\,T_{\text{proxy}}\,f}_{\text{per-sample gradient sweeps for TDDS}} + \underbrace{3k\,T\,f_{\text{large}}}_{\text{train large on coreset}}}$$

**Selection-stage storage overhead.** Streaming `TDDS` requires a $J$-length buffer of scalar contributions per example (and a temporary epoch-direction vector):

$$\boxed{\text{StorageOverhead}_{\texttt{TDDS}} = \mathcal{O}(NJ) \text{ (windowed logs)}}$$

**Example scenario values (ImageNet-1k; $T$=90, $T_{\mathbf{proxy}}$=90, $J$=10).**

$$\text{Compute}_{\texttt{TDDS}} \text{ (PFLOPs only)} \approx \underbrace{622.663}_{3NT_{\text{proxy}}f} + \underbrace{622.663}_{\text{per-sample gradient sweeps}} + \underbrace{280.061}_{3kT\,f_{\text{large}}} = \mathbf{1{,}525.387} \text{ PFLOPs},$$

$$\text{StorageOverhead}_{\texttt{TDDS}} \text{ (float32)} \approx NJ \times 4 = 50{,}734{,}200 \text{ B} = \mathbf{0.051} \text{ GB}.$$

### E.3.3 Using Prediction Uncertainty with a Proxy (`Dyn-Unc`, `DUAL`)

`Dyn-Unc` (He et al., 2024) measures prediction uncertainty via a sliding window of length $J$ over per-example target-class probabilities and averages the windowed uncertainty across proxy training.

`DUAL` (Cho et al., 2025) combines *uncertainty* with *difficulty* (window-mean prediction) and computes scores from an *early* stage of proxy training.

**Execution.** One-time selection with a proxy; the large model then trains on the coreset for the full $T$ epochs:

i) *Train proxy:* train a proxy with per-example forward FLOPs $f$ and parameters $p$ for $T_{\text{proxy}}$ epochs; maintain sliding-window statistics (length $J$) via $O(1)$ updates per visit.
ii) *Score & select:*

- `Dyn-Unc`: use windowed uncertainty (variance over the last $J$ predictions), averaged over all proxy epochs $T_{\text{proxy}}$.
- `DUAL`: use the product of windowed uncertainty and difficulty (window mean), averaged over the *early* proxy epochs $T_{\text{proxy,early}} \leq T_{\text{proxy}}$.
- *Beta sampling* (`DUAL`): apply pruning-ratio–adaptive sampling based on a Beta distribution to stabilize extreme pruning. This adds negligible compute and storage.
  iii) *Train on coreset (large model):* train for *all* $T$ epochs on the top-$k$ points.

**End-to-end compute (selection + coreset training).**

$$
\begin{aligned}
\text{Compute}_{\text{Dyn-Unc}} &= \underbrace{3N\,T_{\text{proxy}}\,f}_{\text{train proxy + logging}} + \underbrace{3k\,T\,f_{\text{large}}}_{\text{train large on coreset}}, \\
\text{Compute}_{\text{DUAL}} &= \underbrace{3N\,T_{\text{proxy,early}}\,f}_{\text{early proxy scoring}} + \underbrace{3k\,T\,f_{\text{large}}}_{\text{train large on coreset}}.
\end{aligned}
$$

**Selection-stage storage overhead.** Only require a scalar window of values per example.

$$
\begin{aligned}
\text{StorageOverhead}_{\text{Dyn-Unc}} &= \mathcal{O}(NJ), \\
\text{StorageOverhead}_{\text{DUAL}} &= \mathcal{O}(NJ).
\end{aligned}
$$

**Example scenario values (ImageNet-1k; $T$=90, $T_{\text{proxy}}$=90, $T_{\text{proxy,early}}$=50, $J$=10).**

$$
\text{Compute}_{\text{Dyn-Unc}} \text{ (PFLOPs only)} \approx \underbrace{622.663}_{3NT_{\text{proxy}}f} + \underbrace{280.061}_{3kT\,f_{\text{large}}} = \mathbf{902.724} \text{ PFLOPs,}
$$

$$
\text{Compute}_{\text{DUAL}} \text{(PFLOPs only)} \approx \underbrace{345.924}_{3NT_{\text{proxy,early}}f} + \underbrace{280.061}_{3kT\,f_{\text{large}}} = \mathbf{625.985} \text{ PFLOPs,}
$$

$$
\text{StorageOverhead (float32)} \approx NJ \times 4 = \mathbf{0.051} \text{ GB for each.}
$$

### E.4 Our Method - Correlation of Loss Differences (`CLD`)

`CLD` builds a coreset by leveraging only *loss values* over training: it records the per-epoch losses of all training points and a small held-out query set, then ranks training examples using the correlation of loss *differences* across epochs between train and query. No gradients or Hessians are required.

**Execution.** One-time selection with a proxy, followed by large-model training:

i) *Train proxy:* train a proxy model for $T_{\text{proxy}}$ epochs on all $N$ samples (per-example forward cost $f$, parameters $p$).
ii) *Collect losses:* during proxy training, record per-epoch losses for all $N$ training samples (no extra compute beyond the training pass), and run a forward pass on all $Q$ query samples each epoch to record their losses ($Qf$ FLOPs per epoch).
iii) *Score & select:* compute `CLD` scores (correlations of loss differences over epochs) and select a size-$k$ coreset. The arithmetic for correlations/ranking is linear-time in the number of stored losses and is negligible compared to FLOPs above.
iv) *Train on coreset:* train the large model on the selected $k$ points for $T$ epochs.

**End-to-end compute (selection + coreset training).**

$$
\text{Compute}_{\text{CLD}} = \underbrace{3N\,T_{\text{proxy}}\,f}_{\text{train proxy}} + \underbrace{Q\,T_{\text{proxy}}\,f}_{\text{query loss collection}} + \underbrace{3k\,T\,f_{\text{large}}}_{\text{train large on coreset}} \quad \text{(FLOPs)}
$$

**Selection-stage storage overhead.** We store loss scalars for all $N$ training and $Q$ query samples across $T_{\text{proxy}}$ epochs:

$$\boxed{\text{StorageOverhead}_{\texttt{CLD}} = \mathcal{O}\big((N{+}Q)T_{\text{proxy}}\big) \quad (\text{loss logs})}$$

**Example scenario values (ImageNet-1k; $T_{\mathbf{proxy}}{=}90$, $T{=}90$).**

$$\text{Compute}_{\texttt{CLD}} \ (\text{PFLOPs only}) \approx \underbrace{622.663}_{3NT_{\text{proxy}}f} + \underbrace{2.097}_{QT_{\text{proxy}}f} + \underbrace{280.061}_{3kT_{\text{late}}f_{\text{large}}} = \mathbf{904.821} \ \text{PFLOPs},$$

$$\text{StorageOverhead}_{\texttt{CLD}} \ (\text{float32}) \approx (N{+}Q)T_{\text{proxy}} \times 4 = 461{,}220{,}120 \ \text{B} = \mathbf{0.461} \ \text{GB}.$$

# F   Comparison of `CLD` with Influence

The impact measured by `CLD` closely aligns with the "*influence*" of individual training samples on a model's predictions that are measured by *Training Data Attribution* (TDA) methods. TDA methods have been widely employed for tasks such as debugging datasets, interpreting models, and optimizing training efficiency Koh & Liang (2017); Yeh et al. (2018); Feldman & Zhang (2020).

The earliest TDA methods utilized *Leave-One-Out* (LOO) training, which involves retraining the model after removing specific data points and observing the changes in performance. While straightforward, LOO retraining is computationally prohibitive for modern deep learning models due to the need for multiple retraining cycles Koh & Liang (2017). Recent TDA metrics, such as FZ-Influence (`Infl`) Feldman & Zhang (2020) and `Datamodels` Ilyas et al. (2022), have gained popularity owing to precomputed scores for widely-used datasets in computer vision. These methods, however, face scalability challenges.

A prominent alternative that arose was *Influence Functions*, which estimated the effect of downweighting individual samples using first-order (gradient) and second-order (Hessian) computations Koh & Liang (2017); Basu et al. (2021) performed at the end of training. Methods like `RandSelect` Wojnowicz et al. (2016) and `Arnoldi` iterations Schioppa et al. (2022) improved computational efficiency by approximating the Hessian. Similarly, `TRAK` Park et al. (2023) combined random projections, gradient-based methods, and ensembling to estimate the influence of training samples. However, these approaches often rely on strong assumptions, such as convergence to a unique optimal solution, which limits their applicability to neural networks. Additionally, Hessian computations introduce significant computational overhead. To address these challenges, *unrolling-based* methods that observe the learning process across training iterations have been proposed. These techniques approximate the impact of samples by differentiating through the optimization trajectory Hara et al. (2019). Among these, `TracIn` Pruthi et al. (2020) is a highly efficient method that estimates influence using gradients tracked throughout training. Its practical implementation, `TracInCP`, uses intermediate checkpoints to alleviate computational burdens. While effective, unrolling methods require storing intermediate training states, leading to high storage and computational costs.

In contrast, `CLD` solely relies on loss trajectories rather than first- or second-order quantities (e.g., gradients and Hessians).

In order to measure the "influence" of a training sample $\vec{z}_m$ on an individual unseen (or query) sample $\vec{z}_q$, we modified Definition 1 slightly to be

$$\texttt{CLD}_{\texttt{infl}}(\vec{z}_m, \vec{z}_q) \coloneqq \rho\left(\vec{\Delta}_m, \vec{\Delta}_q\right) \tag{64}$$

We will now compare the impact measured by this metric ($\texttt{CLD}_{\texttt{infl}}$) to the influence measured by TDA metrics, by utilizing the *linear datamodeling score* (`LDS`) introduced by Park et al. (2023). `LDS` measures the correlation between group-level attribution scores ($\texttt{CLD}_{\texttt{infl}}$ or influence) and their observed impact on model predictions when subsets of training data are used.

**LDS definition** For a query data point $z_q$, random subsets $\{\mathcal{S}_j\}_{j=1}^C$ are sampled from the training dataset, where each subset $\mathcal{S}_j$ contains $\lceil \alpha N \rceil$ points, with $\alpha \in (0,1)$ as the sampling ratio. Each subset $\mathcal{S}_j$ is used to retrain the model $R$ times with different initializations $\{\xi_r\}_{r=1}^R$ and training parameters $\lambda$, resulting in

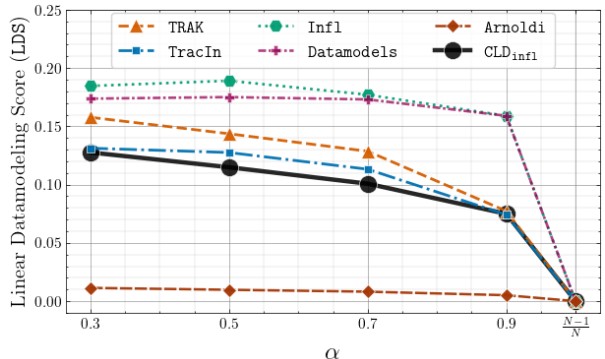 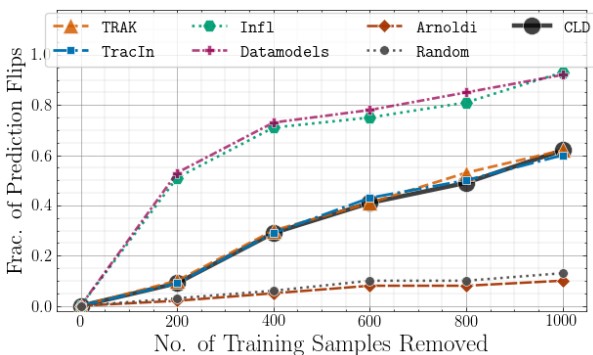

(a) Linear datamodeling scores (LDS) of existing TDA metrics compared to $\mathtt{CLD_{infl}}$. The scores were evaluated on CIFAR-10, ResNet-9 with 200 (randomly selected) query samples evaluated over 100 subsets.

(b) Prediction brittleness of $\mathtt{CLD}$ and TDA metrics on CIFAR-10 with ResNet-9. The top-$k$ influential training samples were removed, and the average prediction flips for 200 query samples over 5 seeds are shown.

Figure 8: Comparison of $\mathtt{CLD}$ to TDA metrics. (Best viewed in color.)

the model $\theta^T_{\mathcal{S}_j, \xi_r}$. This trained model is then used to compute a measurable quantity $f(\vec{z}_q, \theta^T_{\mathcal{S}_j, \xi_r})$. A *group attribution score*, $g_\tau(\vec{z}_q, \mathcal{S}_j, \mathcal{S})$, is calculated as $g_\tau(\vec{z}_q, \mathcal{S}_j, \mathcal{S}) \coloneqq \sum_{\vec{z} \in \mathcal{S}_j} \tau(\vec{z}_q, \vec{z}, \mathcal{S})$, where $\tau(\vec{z}_q, \vec{z}, \mathcal{S})$ is the attribution score for a training point $\vec{z}$ with respect to $\vec{z}_q$. The $\mathtt{LDS}$ is then obtained using Spearman's rank Spearman (1904) correlation ($\rho_s$):

$$\mathtt{LDS}(\vec{z}_q, \alpha) \coloneqq \rho_s \left( \left\{ \frac{1}{R} \sum_{r=1}^{R} f\left(\vec{z}_q, \theta^T_{\mathcal{S}_j, \xi_r}\right) \right\}_{j=1}^{C}, \{g_\tau\left(\vec{z}_q, \mathcal{S}_j, \mathcal{S}\right)\}_{j=1}^{C} \right) \tag{65}$$

**Experimental Setup:** We compared the $\mathtt{LDS}$ scores of $\mathtt{CLD_{infl}}$ against those of $\mathtt{TRAK}$, $\mathtt{Arnoldi}$, $\mathtt{TracIn}$, $\mathtt{Infl}$, and $\mathtt{Datamodels}$. Precomputed scores for $\mathtt{Infl}$ and $\mathtt{Datamodels}$ were used for the CIFAR-10 dataset Krizhevsky et al. (2009) with ResNet-9 He et al. (2016), while 10 models were trained for $\mathtt{TRAK}$, $\mathtt{Arnoldi}$, $\mathtt{TracIn}$, and $\mathtt{CLD_{infl}}$. The evaluation employed $C = 100$ random subsets, sampling ratios $\alpha$ ranging from 0.3 to $\frac{N-1}{N}$, a query set of 200 samples, and $R = 10$ seeds. The measurable quantity was the accuracy of query samples.

**Results and Observations:** The results presented in Figure 8a reveal that while the impact captured by $\mathtt{CLD_{infl}}$ is distinct from the influence measured by traditional TDA metrics, it aligns closely with methods such as $\mathtt{TracIn}$ and $\mathtt{TRAK}$ in terms of behavior while being resource-efficient. Notably, the performance gap between these computationally intensive methods and $\mathtt{CLD_{infl}}$ narrows as $\alpha$ increases. The drop in $\mathtt{LDS}$ scores at $\alpha = \frac{N-1}{N}$ is due to the stochastic nature of model retraining[2].

**Takeaways:** Although $\mathtt{CLD_{infl}}$ (and in essence $\mathtt{CLD}$) fundamentally differs from influence-based TDA metrics, it mirrors their trends at higher sampling ratios while maintaining superior computational efficiency, solidifying its utility as a practical tool for analyzing training dynamics.

## F.1 Importance of the Top-k Samples

We demonstrate that the samples identified by $\mathtt{CLD}$ are indeed pivotal for generalization, addressing the question: *"Are the training samples with the top-k scores truly the most critical for forming a coreset?"* This is evaluated using the *prediction brittleness* metric. This is also mentioned briefly in Section 8.

**Experimental Setup:** To quantify the influence of top-$k$ samples, we systematically removed the most impactful data points identified by their $\mathtt{CLD}$ scores, from the training set and retrained the model. The metric of interest was the fraction of prediction flips observed in a held-out query set after retraining. If these samples are truly critical for generalization, their removal should cause substantial prediction changes. This experiment also included a comparative analysis with the top-$k$ influential samples identified by TDA

---

[2]This observation is consistent with the findings of previous research Karthikeyan & Søgaard (2021); Bae et al. (2024).

scores, discussed in this section. Experiments were performed on the CIFAR-10 dataset using a ResNet-9 architecture, with a randomly selected query set of 200 samples. For each configuration, once the top-$k$ samples were excluded, the model was retrained 5 times to account for randomness, and the average fraction of prediction flips was recorded.

**Results and Observations:** The results, summarized in Figure 8b, illustrate that the top-$k$ samples identified by `CLD` have a comparable influence on prediction outcomes to those identified by TDA-based metrics such as `TracIn` and `TRAK`. Notably, removing the top-800 samples of CIFAR-10, which constitutes just 1.6% of the dataset, results in prediction flips for over half of the query set. This highlights the significant role of the samples identified by `CLD` in supporting model generalization. While metrics like `Datamodels` and `Infl` exhibit greater impact, they are computationally prohibitive, rendering them unsuitable for large-scale coreset generation.

**Takeaways:** `CLD` emerges as an effective and computationally efficient approach for identifying training samples critical to generalization, making it a practical tool for coreset selection in large-scale machine learning pipelines.

