LDS scores of CLD$_{\text{infl}}$ against those of TRAK, Arnoldi, TracIn, Infl, and Datamodels. Precomputed scores for Infl and Datamodels were used for the CIFAR-10 dataset Krizhevsky et al. (2009) with ResNet-9 He et al. (2016), while 10 models were trained for TRAK, Arnoldi, TracIn, and CLD$_{\text{infl}}$. The evaluation employed $C = 100$ random subsets, sampling ratios $\alpha$ ranging from 0.3 to $\frac{N-1}{N}$, a query set of 200 samples, and $R = 10$ seeds. The measurable quantity was the accuracy of query samples.

**Results and Observations:** The results presented in Figure 8a reveal that while the impact captured by CLD$_{\text{infl}}$ is distinct from the influence measured by traditional TDA metrics, it aligns closely with methods such as TracIn and TRAK in terms of behavior while being resource-efficient. Notably, the performance gap between these computationally intensive methods and CLD$_{\text{infl}}$ narrows as $\alpha$ increases. The drop in LDS scores at $\alpha = \frac{N-1}{N}$ is due to the stochastic nature of model retraining[1].

**Takeaways:** Although CLD$_{\text{infl}}$ (and in essence CLD) fundamentally differs from influence-based TDA metrics, it mirrors their trends at higher sampling ratios while maintaining superior computational efficiency, solidifying its utility as a practical tool for analyzing training dynamics.

### F.1 Importance of the Top-k Samples

We demonstrate that the samples identified by CLD are indeed pivotal for generalization, addressing the question: *"Are the training samples with the top-k scores truly the most critical for forming a coreset?"* This is evaluated using the *prediction brittleness* metric. This is also mentioned briefly in Section 8.

**Experimental Setup:** To quantify the influence of top-$k$ samples, we systematically removed the most impactful data points identified by their CLD scores, from the training set and retrained the model. The metric of interest was the fraction of prediction flips observed in a held-out query set after retraining. If these samples are truly critical for generalization, their removal should cause substantial prediction changes. This experiment also included a comparative analysis with the top-$k$ influential samples identified by TDA

---

[1]This observation is consistent with the findings of previous research Karthikeyan & Søgaard (2021); Bae et al. (2024).

scores, discussed in this section. Experiments were performed on the CIFAR-10 dataset using a ResNet-9 architecture, with a randomly selected query set of 200 samples. For each configuration, once the top-$k$ samples were excluded, the model was retrained 5 times to account for randomness, and the average fraction of prediction flips was recorded.

**Results and Observations:** The results, summarized in Figure 8b, illustrate that the top-$k$ samples identified by `CLD` have a comparable influence on prediction outcomes to those identified by TDA-based metrics such as `TracIn` and `TRAK`. Notably, removing the top-800 samples of CIFAR-10, which constitutes just 1.6% of the dataset, results in prediction flips for over half of the query set. This highlights the significant role of the samples identified by `CLD` in supporting model generalization. While metrics like `Datamodels` and `Infl` exhibit greater impact, they are computationally prohibitive, rendering them unsuitable for large-scale coreset generation.

**Takeaways:** `CLD` emerges as an effective and computationally efficient approach for identifying training samples critical to generalization, making it a practical tool for coreset selection in large-scale machine learning pipelines.