# OpenReview forum: "Coresets from Trajectories: Selecting Data via Correlation of Loss Differences"
_TMLR — Accepted by TMLR_

### Review · Reviewer_iHX7 · 2025-09-10

**Summary Of Contributions:**

The paper studies the problem of coreset selection, a subset of the dataset that preserves full-data performance while avoiding the heavy computation many prior methods need (e.g., gradients, Hessians, pairwise similarities). The paper proposes CLD (Correlation of Loss Differences), a simple coreset-selection metric that ranks each training example by how closely its loss-change trajectory over training correlates with a class-wise average loss trajectory computed on a held-out validation set; it needs only per-sample loss logs at a few checkpoints (no gradients) and then picks top-scoring examples per class. The authors provide a convergence analysis showing that training on high-CLD coresets approaches full-data population risk, with the gap controlled by an alignment term $\omega$ and the representativeness $\epsilon$ of the validation set. Empirically on CIFAR-100 and ImageNet-1k, CLD typically matches or outperforms strong baselines across subset sizes, transfers across architectures with minimal accuracy drop, remains stable when computed from early or subsampled checkpoints, and achieves favorable compute/storage trade-offs. Main caveats are the need to log per-sample training losses and to reserve a representative validation set.

**Audience:**

Yes

**Audience Explanation:**

I am not an expert in coreset selection but the paper studies a well defined and interesting problems and proposes a clean and scalable solution to the problem with well done experiments and meaningful results. Data subset selection can have various applications beyond the multi-class classification results discussed in the paper, so I think the findings would be of interest to the ML community,

**Broader Impact Concerns:**

The paper does not have a broader impact section and I don't think it needs one. Although it would be good to mention that procedures like coreset selection can potentially adversely affect performance on minority groups in the data.

**Claims And Evidence:**

Yes

**Claims Explanation:**

> We introduce CLD, a simple and scalable metric for coreset construction based on the correlation between a training sample’s loss differences and the average validation loss trajectory, serving as a proxy for generalization

The authors discuss the metric in Section 4. The metric is described in sufficient detail with clear notation. The metric captures the correlation between the training _dynamics_ of the loss per sample and the _dynamics_ of the average validation loss. By capturing the correlation between the trajectory of loss evolution rather than say just the final loss, the metric captures the effect of the learning algorithm in addition to the informativeness of the sample. The metric is simple and scalable as it does not rely on additional quantities besides those already computed during training process.

> We develop a general convergence framework showing that training on high-CLD samples yields population risk close to full-data training, with the suboptimality explicitly governed by sample alignment and validation representativeness

In Section 4, the authors present the central theoretical results showing that high CLD samples approaches the full-data population risk. The assumptions are stated clearly, and fairly reasonable.The proof provided in the Appendix appears to be sound and correct. The authors also explain the implications of the result in sufficient detail.

> We show that on CIFAR-100 and ImageNet-1k, CLD-selected coresets typically outperform or closely match state-of-the-art methods across a wide range of subset sizes, and remain within 1% of more expensive baselines when not leading.

The experimental results in Section 6 cover CIFAR-100, ImageNet-1k using ResNet-18 as the scoring model. CLD is consistently the top or nearly top-performing approach on both datasets compared to other baselines (some of which use more gradients, etc). The coresets selected with CLD also transfer to other architectures matching the performance of the oracle specific architecture. The ablations in the Appendix further support the consistent results for CLD.

> CLD avoids gradient and curvature computations, incurs minimal compute and storage cost, transfers across architectures via proxy models, and remains stable under checkpoint subsampling and random seeds, making it highly practical for large-scale settings.

The approach does not require expensive computations. It does require maintaining per sample loss over the course of training, which is discussed in the paper (along with ablations). The authors also provide a very well written summary of the computational and memory cost of the baselines in the Appendix.

Overall, I found the paper to be quite well written and thorough. The claims are generally very specific and well supported.

**Requested Changes:**

* (Minor) It would be good to have the full data performance for reference in Figure 2. The numbers are mentioned in the appendix but it would be good to have them in the main result.
* (Minor) The paper is focused on the multi-class classification task. While that is good to have a focused scope, I think it would interesting to discuss the implications to other settings which are prevalent in the current research landscape (unsupervised learning - e.g. next token prediction, masked prediction, etc).
* (Minor) The current experiments while thorough focus solely on vision datasets. The paper would be stronger if CLD's effectiveness could be demonstrated on other modalities.

---

> ### Author Response · Authors · 2025-09-19
> **Response to Reviewer iHX7**
>
> We thank **Reviewer iHX7** for their thorough review and positive assessment of our paper. We are especially grateful for the insightful suggestions for improvement. For convenience, all additions in the revised manuscript are highlighted in red, and deletions are struck out.
>
> > **Requested Change:** It would be good to have the full data performance for reference in Figure 2.
>
> Thank you for the suggestion. We agree this improves the figure's readability and have updated the `caption for Figure 2` to include the mean top-1 accuracy for the full dataset for both CIFAR-100 and ImageNet-1k.
>
> > **Requested Change:** The paper is focused on the multi-class classification task. While that is good to have a focused scope, I think it would be interesting to discuss the implications to other settings which are prevalent in the current research landscape (unsupervised learning - e.g., next token prediction, masked prediction, etc).
> >                                  and
> > **Requested Change:** The current experiments, while thorough, focus solely on vision datasets. The paper would be stronger if CLD's effectiveness could be demonstrated on other modalities.
>
> These are excellent points. To address the scope of our work and its potential in other domains, we have added a new paragraph to the `Discussion (Section 8) page 13` titled **"Beyond Supervised Vision: Scope and Caveats"**.
>
> In this new section, we first discuss how $\mathtt{CLD}$'s core mechanism could be adapted to other supervised tasks like contrastive learning or segmentation. More importantly, we directly address the challenges of applying it to modern language model fine-tuning. We explain that a direct application is non-trivial due to the *"squeezing effect"* described in recent work (Ren & Sutherland, ICLR 2025). As fine-tuning progresses, the model concentrates probability mass on the exact training sequences, which can paradoxically lower the assigned probabilities of similar but non-identical validation samples. This causes a divergence between training and validation loss trajectories, meaning that samples promoting generalization are not guaranteed to have a high $\mathtt{CLD}$ score.
>
> We corroborate this by citing \mathtt{LESS} (Xia et al., ICML 2024), which observes that, unlike in vision, minimizing validation loss does not reliably improve model performance in instruction tuning. We conclude that adapting a $\mathtt{CLD}$-style selector for LLMs will require developing task-appropriate generalization proxies, which we mark as an important direction for future work.
>
> While we believe demonstrating $\mathtt{CLD}$ on other modalities is a valuable future contribution, we have kept the scope of this work focused on vision to allow for controlled and thorough comparisons. We hope this new, more detailed discussion provides the insight the reviewer was looking for.
>
> We appreciate the reviewer's valuable feedback, which we believe has made our paper stronger and its contributions clearer.
>
> ---
>
> ### References
>
> [1] Ren, Yi, and Danica J. Sutherland. "Learning Dynamics of LLM Finetuning." _ICLR_, 2025.
>
> [2] Xia, Mengzhou, et al. "LESS: selecting influential data for targeted instruction tuning." _ICML_. 2024.

---

> > ### Comment · Reviewer_iHX7 · 2025-09-23
> > **Response to rebuttal**
> >
> > Thanks for the response and adding the section about scope!
> >
> > I also read through the other reviews as well as the discussion with reviewer rkSn, who pointed out an important implication in the case of training with minibatches that I missed in my read of the paper. I think the author responses were clear and helpful in clearing up the details. I don't have any further questions.

---

### Review · Reviewer_nKmJ · 2025-09-12

**Summary Of Contributions:**

The paper proposes correlation of loss differences (CLD) a metric to assess the representativeness of a training sample. CLD works by comparing the difference in losses observed on a target point over the training trajectory relative to the difference in losses observed on a validation set. Higher similarity indicates a more representative sample. The authors use CLD as a coreset selection metric. Theoretically, they show that coresets with high CLD will also generalize better. Empirically, the CLD-based coreset selection scheme outperforms all baselines in terms of accuracy as well as many baselines in terms of efficiency.

**Audience:**

Yes

**Audience Explanation:**

Yes; this paper would definitely be of interest to the coreset community, particularly given the strong empirical results.

**Broader Impact Concerns:**

No broader impact concerns.

**Claims And Evidence:**

Yes

**Claims Explanation:**

Overall, this is a well-written, solid paper with strong theoretical backing and compelling experimental results. Theorectically, the authors seem to make reasonable assumptions (even the strongest one, assumption 3, appears to be typical in the literature). The authors evaluate on Imagenet-1k which indicates applicability to large-scale settings.

My primary concern is the lack of a more thorough ablation study beyond the one in Figure 5 (unless I missed it). CLD is based on correlations of loss-differences; some questions that readers may have is how this compares to 1) the parameter gradient similarities between a point and the validation set along the training trajectory, 2) the loss values themselves over the trajectory (perhaps normalized in some way), 3) differences in the gradient norm instead of differences in the losses. Some of these do not have the same advantages of CLD: they are not architecture-agnostic and are less efficient, but comparing with these will provide some more insight into why CLD works so well.

**Requested Changes:**

**Would strengthen**
- Additional ablations as outlined above
- Text in Figure 6 is a bit small; enlarging the text would help readability

---

> ### Author Response · Authors · 2025-09-19
> **Response to Reviewer nKmj**
>
> We thank **Reviewer nKmj** for their positive assessment of our work and for recognizing its strong theoretical backing and compelling experimental results. We are grateful for their constructive suggestions, which we have addressed below. For convenience, all additions in the revised manuscript are highlighted in red, and deletions are struck out.
>
> > **Requested Change:** The lack of a more thorough ablation study comparing $\mathtt{CLD}$ (based on loss-differences) to 1) parameter gradient similarities, 2) raw loss values, and 3) differences in gradient norms.
>
> This is an excellent suggestion that gets to the heart of our method's design. In response to your feedback, we have added a new paragraph to the `Discussion (Section 8) pages 12-13`, "**Loss Differences as a Gradient-Free Proxy for Influence**", to better surface the conceptual comparisons you suggested.
>
> * **1. Comparison to Gradient Similarities:** The first point compares $\mathtt{CLD}$ to methods using gradient similarities, which fall under the umbrella of *influence estimation*. A popular method here is $\mathtt{TracIn}$ ([1]), which tracks gradient similarities to estimate a sample's impact.
>
>     Our work is theoretically founded on this same principle of alignment. `Lemma 1` proves that a first-order expansion shows that a sample's loss difference ($\Delta\ell$) approximates its gradient inner product with the model update ($\langle\nabla\ell, \delta\theta\rangle$). This allows $\mathtt{CLD}$ to serve as a highly efficient, gradient-free proxy for the influence signal that $\mathtt{TracIn}$ measures directly.
>
>     We agree with the reviewer that this is a crucial comparison. Our original submission already included a detailed empirical analysis comparing $\mathtt{CLD}$ to a $\mathtt{TracIn}$ and other influence metrics in `Appendix F`, and the comment highlighted that we needed to make this connection more explicit. The new paragraph in the Discussion now clearly explains this theoretical link and points the reader to our full empirical analysis in `Appendix F`, which shows that $\mathtt{CLD}$ is comparable to $\mathtt{TracIn}$ (gradient similarities) on key attribution metrics.
>
> * **2. & 3. Comparison to Raw Losses & Gradient Norms:** The new discussion paragraph also provides the conceptual ablation you suggested for these alternatives. We clarify that we did not run empirical comparisons on these signals because they are conceptually flawed correlating *raw losses* is confounded by signal drift and scale, while using *gradient norms* discards the crucial directional information that makes alignment meaningful.
>
> We believe this new, more detailed discussion provides the insight the reviewer was looking for and strengthens the motivation for our method.
>
> > **Requested Change:** Text in Figure 6 is a bit small; enlarging the text would help readability.
>
> Thank you for pointing this out. We have updated `Figure 6` in the manuscript and enlarged the text to improve readability.
>
> We hope these changes have fully addressed the reviewer's concerns. We thank the reviewer again for their thoughtful and constructive feedback.
>
> ---
>
> ### References
>
> [1] Pruthi, et. al. _"Estimating training data influence by tracing gradient descent."_, NeurIPS, 2020

---

### Review · Reviewer_rkSn · 2025-09-16

**Summary Of Contributions:**

This paper proposes a new way to construct coresets when there is access to a validation set which is close to the distribution of the dataset. They construct the coreset by considering samples which have a higher correlation to the loss difference between the training and validation trajectory.

The strengths of this contribution is enough ablations and empirical evaluation, an attempt to bridge the theory for the proposed method. The primary weakness is that the loss needs to be computed for all samples in the dataset for each epoch which might be an expensive excercise on large models.  Assumption 3 which assumes representativeess of the validation set is also hard to verify in practice.

**Audience:**

Yes

**Audience Explanation:**

Coreset selection is a relevant problem both in academia and industry and is especially very useful as we scale model sizes. Therefore I am sure a subset of TMLR audience will be interested in this paper.

**Claims And Evidence:**

Yes

**Claims Explanation:**

I like the paper overall especially the empirical evidence provided and the different analyses that the authors have done.

I am a bit iffy about the theoretical results but I don't think the authors overstate their claims. I would request the authors to make the "general"ity of the theoretical results more exact - they are general in what sense?

Let me highlight my reservations about the theoretical framework (I might be wrong and please clarify if possible):
1. How does the size of the coreset, the size of the validation dataset affect the convergence? (Are these effects captured in the quantities $\kappa$ and $\delta$) I know the authors have some discussion in the appendix but a small remark might improve interpretability of the result.

2. Further I don't know how reasonable the assumption 3 is -- is there a simpler way to verify it's existence.

3.  Finally, is there a way to come up with a method which evaluates the loss on some fraction of the dataset and then computes an approximation of the metric? Is there a way to quantify it's error?

**Requested Changes:**

1. Kindly add proof outlines for the proofs in the appendix to make going through them easier. The techniques are standard but still it helps a lot.

2. I'd recommend putting table 1 in the introduction and referencing the papers for each - it greatly improves the readability (btw the table is super nice and helpful)

3. I like the experiments a lot - but can you combine Section 6, 7, and 8 into a single nice numerical section and add an intro paragraph which explains your methodology and what you evaluate on exactly. This is optional but I feel can improve the readability.

4. In Section 9 I would add the limitation that the loss needs to be computed on all samples which might not be possible with very large datasets.

5. Add a remark on tradeoff in size of validation dataset and epsilon?

6. Highlight in introduction the key motivation for which you provide evidence which is computing the coreset on smaller model and transfering it to a large one. (You mention this in passing but i think this should be highlighted more)

---

> ### Author Response · Authors · 2025-09-19
> **Response to Reviewer rkSn**
>
> We sincerely thank **Reviewer rkSn** for their detailed feedback and insightful questions, which have helped us improve the clarity and presentation of our work. We are glad they found the empirical evidence convincing and the table helpful. For convenience, all additions in the revised manuscript are highlighted in red, and deletions are struck out.
>
> We have addressed all requested changes and offer the following point-by-point responses.
>
> > **Requested Change 1 & 5; Theoretical Reservation 1:** Add proof outlines and a remark on the trade-offs involving alignment term ($\kappa$), validation size ($Q$), and selection strictness ($\epsilon$).
>
> Thank you for these suggestions to improve readability. We have now added:
> * **Proof outlines** before each proof in `Appendix B` to make the logical flow easier to follow.
> * An explicit discussion in the main paper under **"Interpreting the Theory"** (`page 7`) that clarifies how the convergence bound is affected by these parameters. We state that the validation error $\delta$ decreases with validation size $Q$, while the alignment term $\kappa$ is improved by a larger coreset size $k$ or stricter selection $\epsilon$.
>
> > **Requested Change 6:** Highlight the key motivation of computing the coreset on a smaller model and transferring it to a large one.
>
> This is an excellent point. The proxy-to-target transfer is a central motivation for our work. We have revised the Introduction (`page 2, paragraph above the summary of contributions`) to more strongly emphasize this advantage, highlighting that coresets can be computed efficiently on a small proxy model (e.g., ResNet-18) and then used to train larger, more complex models (e.g., ResNet-50, VGG, DenseNet) with minimal performance loss.
>
> > **Requested Change 4; Theoretical Reservation 3:** Add a limitation about the need to compute the loss on all samples, which might be expensive.
>
> Thank you for raising this important point. We realize our initial manuscript was not clear enough on this front. A key feature of $\mathtt{CLD}$ is that this is **not a limitation**, because the per-sample training losses are the same ones computed during the standard training loop. **No extra forward passes over the training set are required.** We only require extra forward passes over the validation set, which is typically much smaller than the training set.
>
> To make this explicit, we have added clarifications in three places:
> 1.  `Section 4.1 (Coreset Selection Procedure):` We now state that loss logging "piggybacks" on the training loop.
> 2.  `Section 7 (Computational and Storage Efficiency):` We clarify that scoring uses losses already computed during training.
> 3.  `Appendix A (Algorithm 1):` We added an "Implementation note" to the pseudocode.
>
> > **Theoretical Reservation 2:** How reasonable is Assumption 3, and is there a simpler way to verify it?
>
> Thank you for this question. Assumption 3 is a standard premise in the literature for validation-driven data selection. We have clarified in the manuscript (`page 6`) that this assumption is implicitly or explicitly used by many coreset metrics, including $\mathtt{TDDS}$ ([1]), and $\mathtt{GLISTER}$ ([2]), which also rely on a held-out set as a proxy for the true data distribution. Our theoretical guarantee gracefully handles any potential mismatch between the validation and true distributions via the $\delta$ term in our error bound.
>
> > **Requested Change 2 & 3:** Consider putting Table 1 in the introduction and combining Sections 6, 7, and 8.
>
> We appreciate these suggestions on improving the paper's structure and readability.
> * Regarding **Table 1**, we agree it's very helpful, but feel its density and the many technical terms (e.g., window length $J$, anchor spacing $\gamma_{\text{anc}}$) are better suited for the experimental section after these concepts have been defined. As a compromise, we have added a concise summary of our method's efficiency to the Introduction (`second paragraph on page 2`) with a clear forward reference to the full table.
> * Regarding **merging Sections 6-8**, we opted to keep them separate to distinguish the paper's contributions clearly: Section 6 focuses on accuracy, Section 7 on efficiency (a primary contribution), and Section 8 on broader discussion and analysis. To improve flow, we have added an "Experimental Setup" overview at the beginning of Section 6, which serves as a roadmap for the sections that follow.
>
> We hope these revisions and clarifications have fully addressed the reviewer's concerns. We thank the reviewer again for their constructive feedback.
>
> ---
>
> ### References
> [1] Zhang, et. al. _"Spanning training progress: Temporal dual-depth scoring (TDDS) for enhanced dataset pruning." CVPR_,2024
>
> [2] Killamsetty, et. al. _"Glister: Generalization-based data subset selection for efficient and robust learning". AAAI_, 2021.

---

> > ### Comment · Reviewer_rkSn · 2025-09-19
> >
> > Thanks for resolving most of my comments - I understand your decision in not merging the sections and putting the table upfront. Those were merely cosmetic suggestions. However I want to discuss the following a bit more and understand/make it clear:
> >
> > > Thank you for raising this important point. We realize our initial manuscript was not clear enough on this front. A key feature of $\mathtt{CLD}$ is that this is not a limitation, because the per-sample training losses are the same ones computed during the standard training loop. No extra forward passes over the training set are required. We only require extra forward passes over the validation set, which is typically much smaller than the training set.
> >
> > But then what about a batched setting where the loss is computed over a subset of the training dataset? For a dataset with a million datapoints and a batch/mini-batch size of 10k - it's a 10x expensive to compute it?
> > Or do you only compute after every epoch? In which case your setup is understandable but then your theoretical claim is greatly simplyfing the problem setup.
> > Please clarify this.

---

> ### Author Response · Authors · 2025-09-19
> **Clarification on the loss collection**
>
> Thank you for this excellent follow-up question. This is a crucial practical detail, and it highlights a fundamental advantage of forward-pass methods like $\mathtt{CLD}$ over gradient-based approaches, which we are happy to clarify.
>
> **1. Why $\mathtt{CLD}$ is Efficient in a Batched Setting:**
>
> The reviewer is exactly right that modern training is performed in mini-batches. Our claim that *"no extra forward passes over the training set are required"* is based on the standard procedure for how mini-batch losses are computed.
> * In a forward pass on a mini-batch of size B, the loss is first computed **per-sample**, yielding B individual loss values.
> * These B values are then *reduced* (e.g., averaged) to a single scalar, which is used for the single backward pass.
>
> Our method simply logs the per-sample losses **before** this reduction step, incurring no computational overhead.
>
> **2. The Key Advantage Over Gradient-Based Metrics:**
>
> This brings up a crucial efficiency distinction. Methods that require **per-sample gradients**, such as **$\mathtt{GraNd}$**  and **$\mathtt{TDDS}$**, cannot use the standard "batch-wise" gradient. As shown in our `Table 1`, they must incur a significant overhead to compute these per-sample gradients on the train samples as well, on the order of an extra $3NTf$ FLOPs (for $N$ training samples over $T$ epochs, where a gradient costs $\approx 3f$).
>
> In sharp contrast, $\mathtt{CLD}$'s only additional compute cost is for the forward passes over the small validation set, which amounts to just $QTf$ FLOPs. By leveraging the "free" per-sample losses from the standard training loop, $\mathtt{CLD}$ completely avoids the expensive, per-sample backward passes required by these other methods.
>
> **3. On Per-Epoch Trajectories vs. Per-Iteration Theory:**
>
> Regarding the point on the theoretical setup, our implementation is fully consistent with the per-iteration analysis. We log per-sample losses at every training *step* (i.e., for every mini-batch). Since a standard dataloader processes each sample once per *epoch*, this iterative logging naturally yields a trajectory for each sample that is composed of one loss value per epoch. This per-epoch trajectory is precisely what our theory models and our experiments use. This approach of analyzing dynamics based on per-epoch checkpoints is a *standard practice* in the literature, employed by numerous training-dynamics-based methods including $\mathtt{GraNd}$, $\mathtt{TracIn}$, and $\mathtt{TDDS}$.
>
> Furthermore, our stability analysis in `Figure 5a` shows this is a robust signal, as $\mathtt{CLD}$'s performance is maintained even when these per-epoch trajectories are further subsampled.
>
> To ensure this is perfectly clear for all readers, we will add these notes to `Section 4.1` (**Coreset Selection Procedure**) and `Section 7` (**Computational and Storage Efficiency**) to explicitly describe this mini-batch logging procedure.
>
> We hope this detailed clarification fully resolves the reviewer's concern. We thank the reviewer again for pushing us to make this important detail more explicit.

---

> > ### Comment · Reviewer_rkSn · 2025-09-19
> >
> > Okay I am sorry I am now even more confused - help me understand this. (I am coming from good faith - if there is some logical leap i am making please correct me, likewise acknowledge if there is a mistake in the method) Let us try to break it step by step:
> >
> > Classically in an epoch one has minibatches and for each minibatch one computes the loss and its gradient at a checkpoint t and then updates the parameters using the gradient and logs the loss on the checkpoint of the minibatch (which is an approximation to the loss on the entire dataset). One does this iteratively till the entire dataset is covered and then the next epoch starts.
> >
> > I thought you required computing the loss on ALL training samples? Indicated in Section 4.1 "We first train a source model θS on the full dataset S, recording per-epoch losses for all training and validation
> > samples."
> >
> > But then classically these losses are logged for different checkpoints (the model parameters are updated after every mini-batch) so how is the trajectory consistent across all examples? (As your Theorem 1 assumes)
> > I would have understood if all the loss values are computed at the same checkpoint then your statement holds - but traditionally one only computes the loss values used for backpropagation at that learning step.
> > And so yes for each checkpoint t some subset of data has its loss computed.
> >
> > But your equation (4) assumes that the loss is available for all the data for all the checkpoints.

---

> > > ### Author Response · Authors · 2025-09-19
> > > **Clarifying the checkpoint loss collection (part 1)**
> > >
> > > We are posting our response in two parts due to the character limit on OpenReview.
> > >
> > > We thank the reviewer for their extremely helpful and detailed follow-up. Their core confusion, which is completely understandable, stems from our wording in Section 4.1. The reviewer correctly points to our statement: *"We first train a source model $\theta_\mathbf{S}$ on the full dataset $\mathbf{S}$, recording per-epoch losses for all training and validation samples."* and asks how this is possible without a massive computational cost.
> > >
> > > The reviewer is absolutely correct; computing the loss on all training samples at every single checkpoint would be prohibitively expensive. The key is that our method **does not** do this. The confusion arises from a combination of a notational simplification and the term "per-epoch losses."
> > >
> > > We break down our efficient "piggyback" implementation below.
> > >
> > > #### **1. How Per-Sample Training Losses are Collected Efficiently**
> > >
> > > Instead of extra forward passes, we "piggyback" on the standard training process:
> > > * In a standard forward pass on a mini-batch of size B, the model computes B individual, per-sample loss values.
> > > * These B values are then typically averaged to a single scalar loss, which is used for the single backward pass.
> > > * Our method simply logs these B individual per-sample loss values **before** the averaging step, incurring no extra computational overhead.
> > >
> > > This means we collect the loss for each sample exactly once per epoch, at the moment it appears in its mini-batch. This is what we intended by the phrase "recording per-epoch losses." We recognize this wording was ambiguous and will clarify it in the manuscript. As a result, this process requires **no extra forward passes** over the training set.

---

> ### Author Response · Authors · 2025-09-19
> **Clarifying the checkpoint loss collection (part 2)**
>
> (Part 2/2)
>
> #### **2. Clarifying Trajectory Consistency and Theoretical Grounding**
>
> This brings us to the reviewer's excellent core question about trajectory consistency and the notation in Equation (4).
>
> #### **Notational Simplification**
> The reviewer correctly points out that Equation (4) is a notational simplification. The index $t$ in that equation should be interpreted as the **epoch index**. The term $l(\theta_\mathbf{S}^t, \vec{z})$ represents the loss for a sample $\vec{z}$ during epoch $t$. More precisely:
> * For a **training sample** $\vec{z}_m$, this is the loss recorded when its mini-batch was processed *during* the epoch.
> * For a **validation (query) sample** $\vec{q}_j$, this is the loss recorded during the single forward pass at the *end* of the epoch.
>
> ##### **Theoretical Justification: From Gradient Alignment to CLD**
> The reviewer's question about the soundness of correlating trajectories measured at different moments is the critical point. Our justification is grounded in how practical influence approximation methods work.
>
> 1.  **The Goal: Approximate Gradient Alignment.** The core objective is to approximate a sample's influence, which methods like $\mathtt{TracIn}$ have shown can be measured by its gradient alignment with a target (validation) set [1].
>
> 2.  **Practical Implementation in Stochastic Training.** Computing gradients for all samples at all checkpoints is infeasible. Crucially, the practical implementation of $\mathtt{TracIn}$ addresses this by approximating the influence by summing up the gradient dot products **only in the iterations where the sample $\vec{z}_m$ was used to update the parameters** [1]. Their method, like ours, relies on the checkpoints dictated by the stochastic mini-batch processing.
>
> 3.  **CLD's Parallel Approach and its Theoretical Soundness.** Our "staggered" logging procedure for training samples operates on the exact same principle as methods like $\mathtt{TracIn}$. We collect the loss for a sample $z_m$ **only** at the mini-batch checkpoint where it appears in each epoch. Our key innovation, justified in **Lemma 1**, is using the computationally cheap **loss difference** as a proxy for the expensive gradient inner product used by methods like $\mathtt{TracIn}$.
>
>     The final step is to justify using these staggered mini-batch checkpoints for training losses, especially when correlating them against end-of-epoch validation losses. This approximation holds due to two key assumptions made in our analysis:
>     * **L-smoothness (Assumption 1):** This ensures that the loss function's landscape doesn't change drastically in a small neighborhood. If two parameter vectors $\theta_a$ and $\theta_b$ are close, their corresponding loss values and gradients are also close.
>     * **Small Learning Rate (required by Theorem 1):** Training with a sufficiently small learning rate $\eta$ ensures that the model parameters $\theta$ evolve slowly. The parameters at a mini-batch checkpoint within an epoch are therefore not far from the parameters at the end of that epoch.
>
>     Because the model state evolves smoothly and gradually, the loss computed at the "staggered" checkpoint is a high-fidelity approximation of the loss at the "end-of-epoch" checkpoint, making the correlation a robust and valid measure.
>
> #### **Proposed Revisions**
>
> We recognize this was a major point of confusion. We will revise the manuscript to make this procedure and its justification explicit. We will:
> 1.  Clarify that the index $t$ in trajectory equations refers to the epoch number.
> 2.  Detail the "piggyback" mechanism for logging training losses and the end-of-epoch pass for validation losses.
> 3.  Add a concise theoretical justification that explicitly connects our "staggered" logging to the practical implementation of $\mathtt{TracIn}$, and justifies the correlation with the end-of-epoch validation trajectory via our **L-smoothness** and **learning rate** assumptions.
>
> We thank the reviewer again for their diligence and for pushing us to make this important aspect of our method more explicit. We are confident these revisions will significantly improve the paper's clarity.
>
> ---
>
> #### **References**
> [1] Pruthi, G., et. al. "Estimating training data influence by tracing gradient descent." NeurIPS, 2020.

---

> > ### Comment · Reviewer_rkSn · 2025-09-19
> >
> > Dear Authors,
> > Thanks for the clarification - I'll go through it in detail and get back to you. I'll let you know if I have any more questions.
> > I see your point and I think the concern I raised was valid. Your response has made my understanding better and made the distinction more clear. However I might request do a couple more changes regarding making the assumption around the "staggering" being a high-fidelity approximation more explicit especially upfront in the abstract. I'll let you know once I go through the response and paper again.
> >
> > Overall I am happy with the changes your propose and your engagement - the paper is a nice work and would definitely benefit the community!

---

> > > ### Author Response · Authors · 2025-09-19
> > > **Acknowledgment of Reviewer's Feedback**
> > >
> > > We sincerely thank the reviewer for their positive and thoughtful feedback. We are very glad to hear that our detailed clarification was helpful.
> > >
> > > We appreciate the excellent suggestion regarding the abstract. Making the nature of our "staggered" approximation explicit upfront would certainly improve the paper's clarity for all readers. We are fully prepared to add a concise note about this to the abstract in the final revision, should the reviewer recommend it after their second read.
> > >
> > > Thank you once again for the highly constructive and encouraging dialogue. We look forward to any final recommendations.

---

### Decision · Action_Editor_54Yi · 2025-10-20

**Recommendation:** Accept as is

**Audience:**

Yes

**Audience Explanation:**

Coreset selection is an important topic in machine learning, and this paper provides a novel approach to perform coreset selection. I believe this is of interest for TMLR's audience.

**Claims And Evidence:**

Yes

**Claims Explanation:**

Yes, the paper make clear claims, which are supported by theorems and experiments.